# Optimal prediction of Markov chains with and without spectral gap conditions

**Yanjun Han**
Simons Institute for the Theory of Computing
University of California, Berkeley
Berkeley, CA 94720
`yjhan@berkeley.edu`

**Soham Jana**
Department of Statistics and Data Science
Yale University
New Haven, CT 06520
`soham.jana@yale.edu`

**Yihong Wu**
Department of Statistics and Data Science
Yale University
New Haven, CT 06520
`yihong.wu@yale.edu`

## Abstract

We study the following learning problem with dependent data: Observing a trajectory of length $n$ from a stationary Markov chain with $k$ states, the goal is to predict the next state. For $3 \leq k \leq O(\sqrt{n})$, using techniques from universal compression, the optimal prediction risk in Kullback-Leibler divergence is shown to be $\Theta(\frac{k^2}{n} \log \frac{n}{k^2})$, in contrast to the optimal rate of $\Theta(\frac{\log \log n}{n})$ for $k = 2$ previously shown in [FOPS16]. These rates, slower than the parametric rate of $O(\frac{k^2}{n})$, can be attributed to the memory in the data, as the spectral gap of the Markov chain can be arbitrarily small. To quantify the memory effect, we study irreducible reversible chains with a prescribed spectral gap. In addition to characterizing the optimal prediction risk for two states, we show that, as long as the spectral gap is not excessively small, the prediction risk in the Markov model is $O(\frac{k^2}{n})$, which coincides with that of an iid model with the same number of parameters.

## 1 Introduction

Learning distributions from samples is a central question in statistics and machine learning. While significant progress has been achieved in property testing and estimation based on independent and identically distributed (iid) data, for many applications, most notably natural language processing, two new challenges arise: (a) Modeling data as independent observations fails to capture their temporal dependency; (b) Distributions are commonly supported on a large domain whose cardinality is comparable to or even exceeds the sample size. Continuing the progress made in [FOPS16, HOP18], in this paper we study the following prediction problem with dependent data modeled as Markov chains.

Suppose $X_1, X_2, \ldots$ is a stationary first-order Markov chain on state space $[k] \triangleq \{1, \ldots, k\}$ with unknown statistics. Observing a trajectory $X^n \triangleq (X_1, \ldots, X_n)$, the goal is to predict the next state $X_{n+1}$ by estimating its distribution conditioned on the present data. We use the Kullback-Leibler (KL) divergence as the loss function: For distributions $P = [p_1, \ldots, p_k], Q = [q_1, \ldots, q_k]$, $D(P\|Q) = \sum_{i=1}^k p_i \log \frac{p_i}{q_i}$ if $p_i = 0$ whenever $q_i = 0$ and $D(P\|Q) = \infty$ otherwise. The minimax

35th Conference on Neural Information Processing Systems (NeurIPS 2021).

prediction risk is given by

$$\mathsf{Risk}_{k,n} \triangleq \inf_{\widehat{M}} \sup_{\pi,M} \mathbb{E}[D(M(\cdot|X_n)\|\widehat{M}(\cdot|X_n))] = \inf_{\widehat{M}} \sup_{\pi,M} \sum_{i=1}^{k} \mathbb{E}[D(M(\cdot|i)\|\widehat{M}(\cdot|i))\mathbf{1}_{\{X_n=i\}}] \quad (1)$$

where the supremum is taken over all stationary distributions $\pi$ and transition matrices $M$ (row-stochastic) such that $\pi M = \pi$, the infimum is taken over all estimators $\widehat{M} = \widehat{M}(X_1,\ldots,X_n)$ that are proper Markov kernels (i.e. rows sum to 1), and $M(\cdot|i)$ denotes the $i$th row of $M$. Our main objective is to characterize this minimax risk within universal constant factors as a function of $n$ and $k$.

The prediction problem (1) is distinct from the parameter estimation problem such as estimating the transition matrix [Bar51, AG57, Bil61, WK19] or its properties [CS00, KV16, HJL$^+$18, HKL$^+$19] in that the quantity to be estimated (conditional distribution of the next state) depends on the sample path itself. This is precisely what renders the prediction problem closely relevant to natural applications such as autocomplete and text generation. In addition, this formulation allows more flexibility with far less assumptions compared to the estimation framework. For example, if certain state has very small probability under the stationary distribution, consistent estimation of the transition matrix with respect to usual loss function, e.g. squared risk, may not be possible, whereas the prediction problem is unencumbered by such rare states.

In the special case of iid data, the prediction problem reduces to estimating the distribution in KL divergence. In this setting the optimal risk is well understood, which is known to be $\frac{k-1}{2n}(1 + o(1))$ when $k$ is fixed and $n \to \infty$ [BFSS02] and $\Theta(\frac{k}{n})$ for $k = O(n)$ [Pan04, KOPS15].[1] Typical in parametric models, this rate $\frac{k}{n}$ is commonly referred to the "parametric rate", which leads to a sample complexity that scales proportionally to the number of parameters and inverse proportionally to the desired accuracy.

In the setting of Markov chains, however, the prediction problem is much less understood especially for large state space. Recently the seminal work [FOPS16] showed the surprising result that for stationary Markov chains on two states, the optimal prediction risk satisfies

$$\mathsf{Risk}_{2,n} = \Theta\left(\frac{\log\log n}{n}\right), \quad (2)$$

which has a nonparametric rate even when the problem has only two parameters. The follow-up work [HOP18] studied general $k$-state chains and showed a lower bound of $\Omega(\frac{k\log\log n}{n})$ for uniform (not necessarily stationary) initial distribution; however, the upper bound $O(\frac{k^2 \log\log n}{n})$ in [HOP18] relies on implicit assumptions on mixing time such as spectral gap conditions: the proof of the upper bound for prediction (Lemma 7 in the supplement) and for estimation (Lemma 17 of the supplement) is based on Berstein-type concentration results of the empirical transition counts, which depend on spectral gap. The following theorem resolves the optimal risk for $k$-state Markov chains:

**Theorem 1** (Optimal rates without spectral gap). *There exists a universal constant $C > 0$ such that for all $3 \le k \le \sqrt{n}/C$,*

$$\frac{k^2}{Cn}\log\left(\frac{n}{k^2}\right) \le \mathsf{Risk}_{k,n} \le \frac{Ck^2}{n}\log\left(\frac{n}{k^2}\right). \quad (3)$$

*Furthermore, the lower bound continues to hold even if the Markov chain is restricted to be irreducible and reversible.*

**Remark 1.** The optimal prediction risk of $O(\frac{k^2}{n}\log\frac{n}{k^2})$ can be achieved by an average version of the *add-one estimator* (i.e. Laplace's rule of succession). Given a trajectory $x^n = (x_1,\ldots,x_n)$ of length $n$, denote the transition counts (with the convention $N_i \equiv N_{ij} \equiv 0$ if $n = 0, 1$)

$$N_i = \sum_{\ell=1}^{n-1}\mathbf{1}_{\{x_\ell=i\}}, \quad N_{ij} = \sum_{\ell=1}^{n-1}\mathbf{1}_{\{x_\ell=i, x_{\ell+1}=j\}}. \quad (4)$$

---

[1]Here and below $\asymp, \lesssim, \gtrsim$ or $\Theta(\cdot), O(\cdot), \Omega(\cdot)$ denote equality and inequalities up to universal multiplicative constants.

The add-one estimator for the transition probability $M(j|i)$ is given by

$$\widehat{M}_{x^n}^{+1}(j|i) \triangleq \frac{N_{ij}+1}{N_i+k},\tag{5}$$

which is an additively smoothed version of the empirical frequency. Finally, the optimal rate in (3) can be achieved by the following estimator $\widehat{M}$ defined as an average of add-one estimators over different sample sizes:

$$\widehat{M}_{x^n}(x_{n+1}|x_n) \triangleq \frac{1}{n}\sum_{t=1}^{n}\widehat{M}_{x_{n-t+1}^n}^{+1}(x_{n+1}|x_n).\tag{6}$$

In other words, we apply the add-one estimator to the most recent $t$ observations $(X_{n-t+1},\ldots,X_n)$ to predict the next $X_{n+1}$, then average over $t=1,\ldots,n$. Such Cesàro-mean-type estimators have been introduced before in the density estimation literature (see, e.g., [YB99]). It remains open whether the usual add-one estimator (namely, the last term in (6) which uses all the data) or any add-$c$ estimator for constant $c$ achieves the optimal rate. In contrast, for two-state chains the optimal risk (2) is attained by a hybrid strategy [FOPS16], applying add-$c$ estimator for $c = \frac{1}{\log n}$ for trajectories with at most one transition and $c=1$ otherwise. Also note that the estimator in (6) can be computed in $O(nk)$ time. To derive this first note that given any $j \in [k]$ calculating $\widehat{M}_{x_1^{n-1}}^{+1}(j|x_{n-1})$ takes $O(n)$ time and given any $M_{x_{n-t+1}^{n-1}}^{+1}(j|x_{n-1})$ we need $O(1)$ time to calculate $\widehat{M}_{x_{n-t+2}^{n-1}}^{+1}(j|x_{n-1})$. Summing over all $j$ we get the algorithmic complexity upper bound.

Theorem 1 shows that the departure from the parametric rate of $\frac{k^2}{n}$, first discovered in [FOPS16, HOP18] for binary chains, is even more pronounced for larger state space. As will become clear in the proof, there is some fundamental difference between two-state and three-state chains, resulting in $\mathsf{Risk}_{3,n} = \Theta(\frac{\log n}{n}) \gg \mathsf{Risk}_{2,n} = \Theta(\frac{\log\log n}{n})$. It is instructive to compare the sample complexity for prediction in the iid and Markov model. Denote by $d$ the number of parameters, which is $k-1$ for the iid case and $k(k-1)$ for Markov chains. Define the sample complexity $n^*(d,\epsilon)$ as the smallest sample size $n$ in order to achieve a prescribed prediction risk $\epsilon$. For $\epsilon = O(1)$, we have

$$n^*(d,\epsilon) \asymp \begin{cases} \frac{d}{\epsilon} & \text{iid} \\ \frac{d}{\epsilon}\log\log\frac{1}{\epsilon} & \text{Markov with 2 states} \\ \frac{d}{\epsilon}\log\frac{1}{\epsilon} & \text{Markov with } k \geq 3 \text{ states.} \end{cases}\tag{7}$$

At a high level, the nonparametric rates in the Markov model can be attributed to the memory in the data. On the one hand, Theorem 1 as well as (2) affirm that one can obtain meaningful prediction without imposing any mixing conditions;[2] such decoupling between learning and mixing has also been observed in other problems such as learning linear dynamics [SMT$^+$18, DMM$^+$19]. On the other hand, the dependency in the data does lead to a strictly higher sample complexity than that of the iid case; in fact, the lower bound in Theorem 1 is proved by constructing chains with spectral gap as small as $O(\frac{1}{n})$ (see Section 3). Thus, it is conceivable that with sufficiently favorable mixing conditions, the prediction risk improves over that of the worst case and, at some point, reaches the parametric rate. To make this precise, we focus on Markov chains with a prescribed spectral gap.

It is well-known that for an irreducible and reversible chain, the transition matrix $M$ has $k$ real eigenvalues satisfying $1 = \lambda_1 \geq \lambda_2 \geq \ldots \lambda_k \geq -1$. The *absolute spectral gap* of $M$, defined as

$$\gamma_* \triangleq 1 - \max\left\{|\lambda_i| : i \neq 1\right\},\tag{8}$$

quantifies the memory of the Markov chain. For example, the mixing time is determined by $1/\gamma^*$ (relaxation time) up to logarithmic factors. As extreme cases, the chain which does not move ($M$ is identity) and which is iid ($M$ is rank-one) have spectral gap equal to 0 and 1, respectively. We refer the reader to [LP17] for more background. Note that the definition of absolute spectral gap requires irreducibility and reversibility, thus we restrict ourselves to this class of Markov chains (it is possible to use more general notions such as pseudo spectral gap to quantify the memory of the

---

[2]To see this, it is helpful to consider the extreme case where the chain does not move at all or is periodic, in which case predicting the next state is in fact easy.

process, which is beyond the scope of the current paper). Given $\gamma_0 \in (0, 1)$, define $\mathcal{M}_k(\gamma_0)$ as the set of transition matrices corresponding to irreducible and reversible chains whose absolute spectral gap exceeds $\gamma_0$. Restricting (1) to this subcollection and noticing the stationary distribution here is uniquely determined by $M$, we define the corresponding minimax risk:

$$\mathsf{Risk}_{k,n}(\gamma_0) \triangleq \inf_{\widehat{M}} \sup_{M \in \mathcal{M}_k(\gamma_0)} \mathbb{E}\left[ D(M(\cdot|X_n) \| \widehat{M}(\cdot|X_n)) \right] \tag{9}$$

Extending the result (2) of [FOPS16], the following theorem characterizes the optimal prediction risk for two-state chains with prescribed spectral gaps (the case $\gamma_0 = 0$ correspond to the minimax rate in [FOPS16] over all binary Markov chains):

**Theorem 2** (Spectral gap dependent rates for binary chain). *For any $\gamma_0 \in (0, 1)$*

$$\mathsf{Risk}_{2,n}(\gamma_0) \asymp \frac{1}{n} \max\left\{ 1, \log\log\left( \min\left\{ n, \frac{1}{\gamma_0} \right\} \right) \right\}.$$

Theorem 2 shows that for binary chains, parametric rate $O(\frac{1}{n})$ is achievable if and only if the spectral gap is nonvanishing. While this holds for bounded state space (see Corollary 4 below), for large state space, it turns out that much weaker conditions on the absolute spectral gap suffice to guarantee the parametric rate $O(\frac{k^2}{n})$, achieved by the add-one estimator applied to the entire trajectory. In other words, as long as the spectral gap is not excessively small, the prediction risk in the Markov model behaves in the same way as that of an iid model with equal number of parameters. Similar conclusion has been established previously for the sample complexity of estimating the entropy rate of Markov chains in [HJL$^+$18, Theorem 1].

**Theorem 3.** *The add-one estimator in* (5) *achieves the following risk bound.*

  *(i) For any $k \geq 2$, $\mathsf{Risk}_{k,n}(\gamma_0) \lesssim \frac{k^2}{n}$ provided that $\gamma_0 \gtrsim (\frac{\log k}{k})^{1/4}$.*

  *(ii) In addition, for $k \gtrsim (\log n)^6$, $\mathsf{Risk}_{k,n}(\gamma_0) \lesssim \frac{k^2}{n}$ provided that $\gamma_0 \gtrsim \frac{(\log(n+k))^2}{k}$.*

**Corollary 4.** *For any fixed $k \geq 2$, $\mathsf{Risk}_{k,n}(\gamma_0) = O(\frac{1}{n})$ if and only if $\gamma_0 = \Omega(1)$.*

## 1.1 Proof techniques

The proof of Theorem 1 deviates from existing approaches based on concentration inequalities for Markov chains. For instance, the standard program for analyzing the add-one estimator (5) involves proving concentration of the empirical counts on their population version, namely, $N_i \approx n\pi_i$ and $N_{ij} \approx n\pi_i M(j|i)$, and bounding the risk in the atypical case by concentration inequalities, such as the Chernoff-type bounds in [Lez98, Pau15], which have been widely used in recent work on statistical inference with Markov chains [KV16, HJL$^+$18, HOP18, HKL$^+$19, WK19]. However, these concentration inequalities inevitably depends on the spectral gap of the Markov chain, leading to results which deteriorate as the spectral gap becomes smaller. For two-state chains, results free of the spectral gap are obtained in [FOPS16] using explicit joint distribution of the transition counts; this refined analysis, however, is difficult to extend to larger state space as the probability mass function of $(N_{ij})$ is given by Whittle's formula [Whi55] which takes an unwieldy determinantal form.

Eschewing concentration-based arguments, the crux of our proof of Theorem 1, for both the upper and lower bound, revolves around the following quantity known as *redundancy*:

$$\mathsf{Red}_{k,n} \triangleq \inf_{Q_{X^n}} \sup_{P_{X^n}} D(P_{X^n} \| Q_{X^n}) = \inf_{Q_{X^n}} \sup_{P_{X^n}} \sum_{x^n} P_{X^n}(x^n) \log \frac{P_{X^n}(x^n)}{Q_{X^n}(x^n)}. \tag{10}$$

Here the supremum is taken over all joint distributions of stationary Markov chains $X^n$ on $k$ states, and the infimum is over all joint distributions $Q_{X^n}$. A central quantity which measures the minimax regret in universal compression, the redundancy (10) corresponds to minimax cumulative risk (namely, the total prediction risk when the sample size ranges from 1 to $n$), while (1) is the individual minimax risk at sample size $n$ – see Section 2 for a detailed discussion. We prove the following reduction between prediction risk and redundancy:

$$\frac{1}{n} \mathsf{Red}_{k-1,n}^{\mathsf{sym}} - \frac{\log k}{n} \lesssim \mathsf{Risk}_{k,n} \leq \frac{1}{n-1} \mathsf{Red}_{k,n} \tag{11}$$

where Red$^{\mathsf{sym}}$ denotes the redundancy for *symmetric* Markov chains. The upper bound is standard: thanks to the convexity of the loss function and stationarity of the Markov chain, the risk of the Cesàro-mean estimator (6) can be upper bounded using the cumulative risk and, in turn, the redundancy. The proof of the lower bound is more involved. Given a $(k-1)$-state chain, we embed it into a larger state space by introducing a new state, such that with constant probability, the chain starts from and gets stuck at this state for a period time that is approximately uniform in $[n]$, then enters the original chain. Effectively, this scenario is equivalent to a prediction problem on $k-1$ states with a *random* (approximately uniform) sample size, whose prediction risk can then be related to the cumulative risk and redundancy. This intuition can be made precise by considering a Bayesian setting, in which the $(k-1)$-state chain is randomized according to the least favorable prior for (10), and representing the Bayes risk as conditional mutual information and applying the chain rule.

Given the above reduction in (11), it suffices to show both redundancies therein are on the order of $\frac{k^2}{n}\log\frac{n}{k^2}$. The redundancy is upper bounded by *pointwise redundancy*, which replaces the average in (10) by the maximum over all trajectories. Following [DMPW81, CS04], we consider an explicit probability assignment defined by add-one smoothing and using combinatorial arguments to bound the pointwise redundancy, shown optimal by information-theoretic arguments.

The optimal spectral gap-dependent rate in Theorem 2 relies on the key observation in [FOPS16] that, for binary chains, the dominating contribution to the prediction risk comes from trajectories with a single transition, for which we may apply an add-$c$ estimator with $c$ depending appropriately on the spectral gap. The lower bound is shown using a Bayesian argument similar to that of [HOP18, Theorem 1]. The proof of Theorem 3 relies on more delicate concentration arguments as the spectral gap is allowed to be vanishingly small. Notably, for small $k$, direct application of existing Bernstein inequalities for Markov chains in [Lez98, Pau15] falls short of establishing the parametric rate of $O(\frac{k^2}{n})$ (see Remark 4 in Section 7.2 for details); instead, we use a fourth moment bound which turns out to be well suited for analyzing concentration of empirical counts conditional on the terminal state.

For large $k$, we further improve the spectral gap condition using a simulation argument for Markov chains using independent samples [Bil61, HJL$^+$18]. A key step is a new concentration inequality for $D(P\|\widehat{P}_{n,k}^{+1})$, where $\widehat{P}_{n,k}^{+1}$ is the add-one estimator based on $n$ iid observations of $P$ supported on $[k]$:

$$\mathbb{P}\left(D(P\|\widehat{P}_{n,k}^{+1}) \geq c\cdot\frac{k}{n} + \frac{\mathsf{polylog}(n)\cdot\sqrt{k}}{n}\right) \leq \frac{1}{\mathsf{poly}(n)}, \tag{12}$$

for some absolute constant $c > 0$. Note that an application of the classical concentration inequality of McDiarmid would result in the second term being $\mathsf{polylog}(n)/\sqrt{n}$, and (12) crucially improves this to $\mathsf{polylog}(n)\cdot\sqrt{k}/n$. Such an improvement has been recently observed by [MJT$^+$20, Agr20, GR20] in studying the similar quantity $D(\widehat{P}_n\|P)$ for the (unsmoothed) empirical distribution $\widehat{P}_n$; however, these results, based on either the method of types or an explicit upper bound of the moment generating function, are not directly applicable to (12) in which the true distribution $P$ appears as the first argument in the KL divergence.

## 1.2 Related work

While the exact prediction problem studied in this paper has recently been in focus since [FOPS16, HOP18], there exists a large body of literature on relate works. As mentioned before some of our proof strategies draws inspiration and results from the study of redundancy in universal compression, its connection to mutual information, as well as the perspective of sequential probability assignment as prediction, dating back to [Dav73, DMPW81, Ris84, Sht87, Rya88]. Asymptotic characterization of the minimax redundancy for Markov sources, both average and pointwise, were obtained in [Dav83, Att99, JS02], in the regime of fixed alphabet size $k$ and large sample size $n$. Non-asymptotic characterization was obtained in [Dav83] for $n \gg k\log^2 k$ and recently extended to $n \asymp k^2$ in [TJW18], which further showed that the behavior of the redundancy remains unchanged even if the Markov chain is very close to being iid in terms of spectral gap $\gamma^* = 1 - o(1)$.

The current paper adds to a growing body of literature devoted to statistical learning with dependent data, in particular those dealing with Markov chains. Estimation of the transition matrix [Bar51, AG57, Bil61, Sin64] and testing the order of Markov chains [CS00] have been well studied in the large-sample regime. More recently attention has been shifted towards large state space and

nonasymptotics. For example, [WK19] studied the estimation of transition matrix in $\ell_\infty \to \ell_\infty$ induced norm for Markov chains with prescribed pseudo spectral gap and minimum probability mass of the stationary distribution, and determined sample complexity bounds up to logarithmic factors. Similar results have been obtained for estimating properties of Markov chains, including mixing time and spectral gap [HKL+19], entropy rate [KV16, HJL+18, OS20], graph statistics based on random walk [BHOP18], as well as identity testing [DDG18, CB19, WK20, FW21]. Most of these results rely on assumptions on the Markov chains such as lower bounds on the spectral gap and the stationary distribution, which afford concentration for sample statistics of Markov chains. In contrast, one of the main contributions in this paper, in particular Theorem 1, is that optimal prediction can be achieved without these assumptions, thereby providing a novel way of tackling these seemingly unavoidable issues. This is ultimately accomplished by information-theoretic and combinatorial techniques from universal compression.

## 1.3 Notations and preliminaries

For $n \in \mathbb{N}$, let $[n] \triangleq \{1, \ldots, n\}$. Denote $x^n = (x_1, \ldots, x_n)$ and $x_t^n = (x_t, \ldots, x_n)$. The distribution of a random variable $X$ is denoted by $P_X$. In a Bayesian setting, the distribution of a parameter $\theta$ is referred to as a prior, denoted by $P_\theta$. We recall the following definitions from information theory [CK82, CT06]. The conditional KL divergence is defined as as an average of KL divergence between conditional distributions:

$$D(P_{A|B}\|Q_{A|B}|P_B) \triangleq \mathbb{E}_{B \sim P_B}[D(P_{A|B}\|Q_{A|B})] = \int P_B(db)D(P_{A|B=b}\|Q_{A|B=b}). \qquad (13)$$

The mutual information between random variables $A$ and $B$ with joint distribution $P_{AB}$ is $I(A;B) \triangleq D(P_{B|A}\|P_B|P_A)$; similarly, the conditional mutual information is defined as

$$I(A;B|C) \triangleq D(P_{B|A,C}\|P_{B|C}|P_{A,C}).$$

The following variational representation of (conditional) mutual information is well-known

$$I(A;B) = \min_{Q_B} D(P_{B|A}\|Q_B|P_A), \quad I(A;B|C) = \min_{Q_{B|C}} D(P_{B|A,C}\|Q_{B|C}|P_{AC}). \qquad (14)$$

The entropy of a discrete random variables $X$ is $H(X) \triangleq \sum_x P_X(x) \log \frac{1}{P_X(x)}$.

The rest of the paper is organized as follows. In Section 2 we describe the general paradigms of minimax redundancy and prediction risk and their dual representation in terms of mutual information. We give a general redundancy-based bound on the prediction risk, which leads to the upper bound in Theorem 1. Section 3 presents the main idea of the lower bound construction. In the supplement, Section 4 discusses the assumptions and implications of our results and several open problems. Sections 5 and 6 contain the full proof for the upper and lower bound in Theorem 1. Due to space limitations, proofs for spectral-gap dependent risk bounds in Theorems 2 and 3 are given in Section 7.

## 2 Two general paradigms

### 2.1 Redundancy, prediction risk, and mutual information representation

For $n \in \mathbb{N}$, let $\mathcal{P} = \{P_{X^{n+1}|\theta} : \theta \in \Theta\}$ be a collection of joint distributions parameterized by $\theta$.

**"Compression".** Consider a sample $X^n \triangleq (X_1, \ldots, X_n)$ of size $n$ drawn from $P_{X^n|\theta}$ for some unknown $\theta \in \Theta$. The *redundancy* of a probability assignment (joint distribution) $Q_{X^n}$ is defined as the worst-case KL risk of fitting the joint distribution of $X^n$, namely

$$\mathsf{Red}(Q_{X^n}) \triangleq \sup_{\theta \in \Theta} D(P_{X^n|\theta}\|Q_{X^n}). \qquad (15)$$

Optimizing over $Q_{X^n}$, the minimax redundancy is defined as

$$\mathsf{Red}_n \triangleq \inf_{Q_{X^n}} \mathsf{Red}_n(Q_{X^n}), \qquad (16)$$

where the infimum is over all joint distribution $Q_{X^n}$. This quantity can be operationalized as the redundancy (i.e. regret) in the setting of universal data compression, that is, the excess number of bits compared to the optimal compressor of $X^n$ that knows $\theta$ [CT06, Chapter 13].

The capacity-redundancy theorem (see [Kem74] for a very general result) provides the following mutual information characterization of (16):

$$\mathsf{Red}_n = \sup_{P_\theta} I(\theta; X^n), \tag{17}$$

where the supremum is over all distributions (priors) $P_\theta$ on $\Theta$. In view of the variational representation (14), this result can be interpreted as a minimax theorem:

$$\mathsf{Red}_n = \inf_{Q_{X^n}} \sup_{P_\theta} D(P_{X^n|\theta} \| Q_{X^n} | P_\theta) = \sup_{P_\theta} \inf_{Q_{X^n}} D(P_{X^n|\theta} \| Q_{X^n} | P_\theta).$$

Typically, for fixed model size and $n \to \infty$, one expects that $\mathsf{Red}_n = \frac{d}{2} \log n(1 + o(1))$, where $d$ is the number of parameters; see [Ris84] for a general theory of this type. Indeed, on a fixed alphabet of size $k$, we have $\mathsf{Red}_n = \frac{k-1}{2} \log n(1 + o(1))$ for iid model [Dav73] and $\mathsf{Red}_n = \frac{k^m(k-1)}{2} \log n(1 + o(1))$ for $m$-order Markov models [Tro74], with more refined asymptotics shown in [XB97, SW12]. For large alphabets, nonasymptotic results have also been obtained. For example, for first-order Markov model, $\mathsf{Red}_n \asymp k^2 \log \frac{n}{k^2}$ provided that $n \gtrsim k^2$ [Dav83, TJW18].

**"Prediction".** Consider the problem of predicting the next unseen data point $X_{n+1}$ based on the observations $X_1, \ldots, X_n$, where $(X_1, \ldots, X_{n+1})$ are jointly distributed as $P_{X^{n+1}|\theta}$ for some unknown $\theta \in \Theta$. Here, an estimator is a distribution (for $X_{n+1}$) as a function of $X^n$, which, in turn, can be written as a conditional distribution $Q_{X_{n+1}|X^n}$. As such, its worst-case average risk is

$$\mathsf{Risk}(Q_{X_{n+1}|X^n}) \triangleq \sup_{\theta \in \Theta} D(P_{X_{n+1}|X^n,\theta} \| Q_{X_{n+1}|X^n} | P_{X^n|\theta}), \tag{18}$$

where the conditional KL divergence is defined in (13). The minimax prediction risk is then defined as

$$\mathsf{Risk}_n \triangleq \inf_{Q_{X_{n+1}|X^n}} \mathsf{Risk}(Q_{X_{n+1}|X^n}), \tag{19}$$

While (16) does not directly correspond to a statistical estimation problem, (19) is exactly the familiar setting of "density estimation", where $Q_{X_{n+1}|X^n}$ is understood as an estimator for the distribution of the unseen $X_{n+1}$ based on the available data $X_1, \ldots, X_n$.

In the Bayesian setting where $\theta$ is drawn from a prior $P_\theta$, the Bayes prediction risk coincides with the conditional mutual information as a consequence of the variational representation (14):

$$\inf_{Q_{X_{n+1}|X^n}} \mathbb{E}_\theta[D(P_{X_{n+1}|X^n,\theta} \| Q_{X_{n+1}|X^n} | P_{X^n|\theta})] = I(\theta; X_{n+1}|X^n). \tag{20}$$

Furthermore, the Bayes estimator that achieves this infimum takes the following form:

$$Q_{X_{n+1}|X^n}^{\mathsf{Bayes}} = P_{X^{n+1}|X^n} = \frac{\int_\Theta P_{X^{n+1}|\theta} \, dP_\theta}{\int_\Theta P_{X^n|\theta} \, dP_\theta}, \tag{21}$$

known as the Bayes predictive density [Dav73, LB04]. These representations play a crucial role in the lower bound proof of Theorem 1. Under appropriate conditions which hold for Markov models (see Lemma 19 in Appendix A), the minimax prediction risk (19) also admits a dual representation analogous to (17):

$$\mathsf{Risk}_n = \sup_{\theta \sim \pi} I(\theta; X_{n+1}|X^n), \tag{22}$$

which, in view of (20), show that the principle of "minimax=worst-case Bayes" continues to hold for prediction problem in Markov models.

The following result (proved in Section 5) relates the redundancy and the prediction risk.

**Lemma 5.** *For any model* $\mathcal{P}$,

$$\mathsf{Red}_n \leq \sum_{t=0}^{n-1} \mathsf{Risk}_t. \tag{23}$$

*In addition, suppose that each* $P_{X^n|\theta} \in \mathcal{P}$ *is stationary and* $m^{\mathrm{th}}$*-order Markov. Then for all* $n \geq m+1$,

$$\mathsf{Risk}_n \leq \mathsf{Risk}_{n-1} \leq \frac{\mathsf{Red}_n}{n-m}. \tag{24}$$

*Furthermore, for any joint distribution $Q_{X^n}$ factorizing as $Q_{X^n} = \prod_{t=1}^{n} Q_{X_t|X^{t-1}}$, the prediction risk of the estimator*

$$\widetilde{Q}_{X_n|X^{n-1}}(x_n|x^{n-1}) \triangleq \frac{1}{n-m} \sum_{t=m+1}^{n} Q_{X_t|X^{t-1}}(x_n|x_{n-t+1}^{n-1}) \tag{25}$$

*is bounded by the redundancy of $Q_{X^n}$ as*

$$\mathsf{Risk}(\widetilde{Q}_{X_n|X^{n-1}}) \leq \frac{1}{n-m} \mathsf{Red}(Q_{X^n}). \tag{26}$$

**Remark 2.** Note that the upper bound (23) on redundancy, known as the "estimation-compression inequality" [KOPS15, FOPS16], holds without conditions, while the lower bound (24) relies on stationarity and Markovity. For iid data, the estimation-compression inequality is almost an equality; however, this is not the case for Markov chains, as both sides of (23) differ by an unbounded factor of $\Theta(\log \log n)$ for $k = 2$ and $\Theta(\log n)$ for fixed $k \geq 3$ – see (2) and Theorem 1. On the other hand, Markov chains with at least three states offers a rare instance where (24) is tight, namely, $\mathsf{Risk}_n \asymp \frac{\mathsf{Red}_n}{n}$ (cf. Lemma 6).

## 2.2 Proof of the upper bound part of Theorem 1

Specializing to first-order stationary Markov chains with $k$ states, we denote the redundancy and prediction risk in (16) and (19) by $\mathsf{Red}_{k,n}$ and $\mathsf{Risk}_{k,n}$, the latter of which is precisely the quantity previously defined in (1). Applying Lemma 5 yields $\mathsf{Risk}_{k,n} \leq \frac{1}{n-1} \mathsf{Red}_{k,n}$. To upper bound $\mathsf{Red}_{k,n}$, consider the following probability assignment:

$$Q(x_1, \cdots, x_n) = \frac{1}{k} \prod_{t=1}^{n-1} \widehat{M}_{x^t}^{+1}(x_{t+1}|x_t) \tag{27}$$

where $\widehat{M}^{+1}$ is the add-one estimator defined in (5). This $Q$ factorizes as $Q(x_1) = \frac{1}{k}$ and $Q(x_{t+1}|x^t) = \widehat{M}_{x^t}^{+1}(x_{t+1}|x_t)$. The following lemma (proved in Section 5.2) bounds the redundancy of $Q$:

**Lemma 6.** $\mathsf{Red}(Q) \leq k(k-1)\left[\log\left(1 + \frac{n-1}{k(k-1)}\right) + 1\right] + \log k.$

Combined with Lemma 5, Lemma 6 shows that $\mathsf{Risk}_{k,n} \leq C\frac{k^2}{n} \log \frac{n}{k^2}$ for all $k \leq \sqrt{n/C}$ and some universal constant $C$, achieved by the estimator (6), which is obtained by applying the rule (25) to (27).

# 3 Optimal rates without spectral gap

In this section, we sketch the proof of the lower bound part of Theorem 1, which shows the optimality of the average version of the add-one estimator (25). We first describe the lower bound construction for three-state chains, which is subsequently extended to $k$ states.

## 3.1 Warmup: an $\Omega(\frac{\log n}{n})$ lower bound for three-state chains

**Theorem 7.** $\mathsf{Risk}_{3,n} = \Omega\left(\frac{\log n}{n}\right).$

To show Theorem 7, consider the following one-parameter family of transition matrices:

$$\mathcal{M} = \left\{ M_p = \begin{bmatrix} 1 - \frac{2}{n} & \frac{1}{n} & \frac{1}{n} \\ \frac{1}{n} & 1 - \frac{1}{n} - p & p \\ \frac{1}{n} & p & 1 - \frac{1}{n} - p \end{bmatrix} : 0 \leq p \leq 1 - \frac{1}{n} \right\}. \tag{28}$$

Note that each transition matrix in $\mathcal{M}$ is symmetric (hence doubly stochastic), whose corresponding chain is reversible with a uniform stationary distribution and spectral gap $\Theta(\frac{1}{n})$; see Fig. 1.

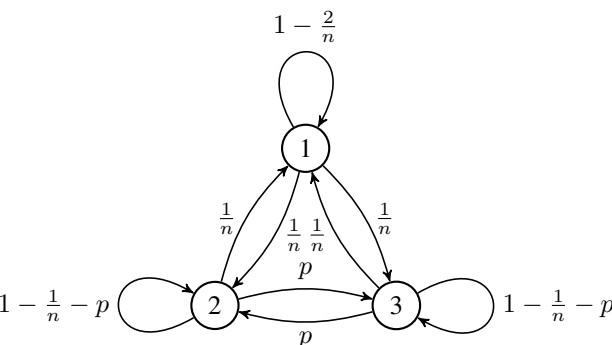

Figure 1: Lower bound construction for three-state chains.

The proof of Theorem 7 is given in Section 6.1 using a Bayesian argument. The main idea is as follows. Notice that by design, with constant probability, the trajectory is of the following form: The chain starts and stays at state 1 for $t$ steps, and then transitions into state 2 or 3 and never returns to state 1, where $t = 1, \ldots, n-1$. Since $p$ is the single unknown parameter, the only useful observations are visits to state 2 and 3 and each visit entails one observation about $p$ by flipping a coin with bias roughly $p$. Thus the effective sample size for estimating $p$ is $n - t - 1$ and we expect the best estimation error is of the order of $\frac{1}{n-t}$. However, $t$ is not fixed. In fact, conditioned on the trajectory is of this form, $t$ is roughly uniformly distributed between 1 and $n-1$. As such, we anticipate the estimation error of $p$ is approximately

$$\frac{1}{n-1} \sum_{i=1}^{n-1} \frac{1}{n-t} = \Theta\left(\frac{\log n}{n}\right).$$

Intuitively speaking, the construction in Fig. 1 "embeds" a symmetric two-state chain (with states 2 and 3) with unknown parameter $p$ into a space of three states, by adding a "nuisance" state 1, which effectively slows down the exploration of the useful part of the state space, so that in a trajectory of length $n$, the effective number of observations we get to make about $p$ is roughly uniformly distributed between 1 and $n$. This explains the extra log factor in Theorem 7, which actually stems from the harmonic sum in $\mathbb{E}[\frac{1}{\text{Uniform}([n])}]$. We will fully explore this embedding idea in Section 3.2 to deal with larger state space.

## 3.2  $k$-state chains

The above lower bound construction for 3-state chains can be generalized to $k$-state chains with $k \geq 3$. The high-level argument is again to augment a $(k-1)$-state chain into a $k$-state chain. Specifically, we partition the state space $[k]$ into two sets $\mathcal{S}_1 = \{1\}$ and $\mathcal{S}_2 = \{2, 3, \cdots, k\}$. Consider a $k$-state Markov chain such that the transition probabilities from $\mathcal{S}_1$ to $\mathcal{S}_2$, and from $\mathcal{S}_2$ to $\mathcal{S}_1$, are both very small (on the order of $\Theta(1/n)$). At state 1, the chain either stays at 1 with probability $1 - 1/n$ or moves to one of the states in $\mathcal{S}_2$ with equal probability $\frac{1}{n(k-1)}$; at each state in $\mathcal{S}_2$, the chain moves to 1 with probability $\frac{1}{n}$; otherwise, within the state subspace $\mathcal{S}_2$, the chain evolves according to some symmetric transition matrix $T$. See Fig. 2 in Section 6.2 for the precise transition diagram.

The key feature of such a chain is as follows. Let $\mathcal{X}_t$ be the event that $X_1, X_2, \cdots, X_t \in \mathcal{S}_1$ and $X_{t+1}, \cdots, X_n \in \mathcal{S}_2$. For each $t \in [n-1]$, one can show that $\mathbb{P}(\mathcal{X}_t) \geq c/n$ for some absolute constant $c > 0$. Moreover, conditioned on the event $\mathcal{X}_t$, $(X_{t+1}, \ldots, X_n)$ is equal in law to a stationary Markov chain $(Y_1, \cdots, Y_{n-t})$ on state space $\mathcal{S}_2$ with symmetric transition matrix $T$. It is not hard to show that estimating $M$ and $T$ are nearly equivalent. Consider the Bayesian setting where $T$ is drawn from some prior. We have

$$\inf_{\widehat{M}} \mathbb{E}_T\left[\mathbb{E}[D(M(\cdot|X_n)\|\widehat{M}(\cdot|X_n))|\mathcal{X}_t]\right] \approx \inf_{\widehat{T}} \mathbb{E}_T\left[\mathbb{E}[D(T(\cdot|Y_{n-t})\|\widehat{T}(\cdot|Y_{n-t}))]\right] = I(T; Y_{n-t+1}|Y^{n-t}),$$

where the last equality follows from the representation (20) of Bayes prediction risk as conditional mutual information. Lower bounding the minimax risk by the Bayes risk, we have

$$\mathsf{Risk}_{k,n} \geq \inf_{\widehat{M}} \mathbb{E}_T \left[ \mathbb{E}[D(M(\cdot|X_n)\|\widehat{M}(\cdot|X_n))] \right]$$

$$\geq \inf_{\widehat{M}} \sum_{t=1}^{n-1} \mathbb{E}_M \left[ \mathbb{E}[D(M(\cdot|X_n)\|\widehat{M}(\cdot|X_n))|\mathcal{X}_t] \cdot \mathbb{P}(\mathcal{X}_t) \right]$$

$$\geq \frac{c}{n} \cdot \sum_{t=1}^{n-1} \inf_{\widehat{M}} \mathbb{E}_M \left[ \mathbb{E}[D(M(\cdot|X_n)\|\widehat{M}(\cdot|X_n))|\mathcal{X}_t] \right]$$

$$\approx \frac{c}{n} \cdot \sum_{t=1}^{n-1} I(T; Y_{n-t+1}|Y^{n-t}) = \frac{c}{n} \cdot (I(T; Y^n) - I(T; Y_1)). \tag{29}$$

Note that $I(T; Y_1) \leq H(Y_1) \leq \log(k-1)$ since $Y_1$ takes values in $\mathcal{S}_2$. Maximizing the right hand side over the prior $P_T$ and recalling the dual representation for redundancy in (17), the above inequality (29) leads to a risk lower bound of $\mathsf{Risk}_{k,n} \gtrsim \frac{1}{n}(\mathsf{Red}_{k-1,n}^{\mathsf{sym}} - \log k)$, where $\mathsf{Red}_{k-1,n}^{\mathsf{sym}} = \sup I(T; Y_1)$ is the redundancy for *symmetric* Markov chains with $k-1$ states and sample size $n$. Since symmetric transition matrices still have $\Theta(k^2)$ degrees of freedom, it is expected that $\mathsf{Red}_{k,n}^{\mathsf{sym}} \asymp k^2 \log \frac{n}{k^2}$ for $n \gtrsim k^2$, so that (29) yields the desired lower bound $\mathsf{Risk}_{k,n} = \Omega(\frac{k^2}{n} \log \frac{n}{k^2})$ in Theorem 1. The rigorous proof is carried out in Section 6.2.

## Acknowledgment

Y. Wu is supported in part by NSF Grant CCF-1900507, an NSF CAREER award CCF-1651588, and an Alfred Sloan fellowship.

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
