# 4  Discussions and open problems

We discuss the assumptions and implications of our results as well as related open problems.

**Very large state space.**   Theorem 1 determines the optimal prediction risk under the assumption of $k \lesssim \sqrt{n}$. When $k \gtrsim \sqrt{n}$, Theorem 1 shows that the KL risk is bounded away from zero. However, as the KL risk can be as large as $\log k$, it is a meaningful question to determine the optimal rate in this case, which, thanks to the general reduction in (11), reduces to determining the redundancy for symmetric and general Markov chains. For iid data, the minimax *pointwise* redundancy is known to be $n \log \frac{k}{n} + O(\frac{n^2}{k})$ [SW12, Theorem 1] when $k \gg n$. Since the average and pointwise redundancy usually behave similarly, for Markov chains it is reasonable to conjecture that the redundancy is $\Theta(n \log \frac{k^2}{n})$ in the large alphabet regime of $k \gtrsim \sqrt{n}$, which, in view of (11), would imply the optimal prediction risk is $\Theta(\log \frac{k^2}{n})$ for $k \gg \sqrt{n}$. In comparison, we note that the prediction risk is at most $\log k$, achieved by the uniform distribution.

**Other loss functions**   As mentioned in Section 1.1, standard arguments based on concentration inequalities inevitably rely on mixing conditions such as the spectral gap. In contrast, the risk bound in Theorem 1, which is free of any mixing condition, is enabled by powerful techniques from universal compression which bound the redundancy by the pointwise maximum over all trajectories combined with information-theoretic or combinatorial argument. This program only relies on the Markovity of the process rather than stationarity or spectral gap assumptions. The limitation of this approach, however, is that the reduction between prediction and redundancy crucially depends on the form of the KL loss function[3] in (1), which allows one to use the mutual information representation and the chain rule to relate individual risks to the cumulative risk. More general loss in terms of $f$-divergence have been considered in [HOP18]. Obtaining spectral gap-independent risk bound for these loss functions, this time without the aid of universal compression, is an open question.

**Stationarity**   As mentioned above, the redundancy result in Lemma 6 (see also [Dav83, TJW18]) holds for nonstationary Markov chains as well. However, our redundancy-based risk upper bound in Lemma 5 crucially relies on stationarity. It is unclear whether the result of Theorem 1 carries over to nonstationary chains.

# 5  Proofs in Section 2

## 5.1  Proof of Lemma 5

*Proof.* The upper bound on the redundancy follows from the chain rule of KL divergence:

$$D(P_{X^n|\theta}\|Q_{X^n}) = \sum_{t=1}^{n} D(P_{X_t|X^{t-1},\theta}\|Q_{X_t|X^{t-1}}|P_{X^{t-1}}). \tag{30}$$

Thus

$$\sup_{\theta \in \Theta} D(P_{X^n|\theta}\|Q_{X^n}) \leq \sum_{t=1}^{n} \sup_{\theta \in \Theta} D(P_{X_t|X^{t-1},\theta}\|Q_{X_t|X^{t-1}}|P_{X^{t-1}}).$$

Minimizing both sides over $Q_{X^n}$ (or equivalently, $Q_{X_t|X^{t-1}}$ for $t = 1, \ldots, n$) yields (23).

To upper bound the prediction risk using redundancy, fix any $Q_{X^n}$, which gives rise to $Q_{X_t|X^{t-1}}$ for $t = 1, \ldots, n$. For clarity, let use denote the $t^{\text{th}}$ estimator as $\widehat{P}_t(\cdot|x^{t-1}) = Q_{X_t|X^{t-1}=x^{t-1}}$. Consider the estimator $\widetilde{Q}_{X_n|X^{n-1}}$ defined in (25), namely,

$$\widetilde{Q}_{X_n|X^{n-1}=x^{n-1}} \triangleq \frac{1}{n-m} \sum_{t=m+1}^{n} \widehat{P}_t(\cdot|x_{n-t+1}, \ldots, x_{n-1}). \tag{31}$$

That is, we apply $\widehat{P}_t$ to the most recent $t-1$ symbols prior to $X_n$ for predicting its distribution, then average over $t$. We may bound the prediction risk of this estimator by redundancy as follows: Fix

---

[3] In fact, this connection breaks down if one swap $M$ and $\widehat{M}$ in the KL divergence in (1).

$\theta \in \Theta$. To simplify notation, we suppress the dependency of $\theta$ and write $P_{X^n|\theta} \equiv P_{X^n}$. Then

$$D(P_{X_n|X^{n-1}}\|\widetilde{Q}_{X_n|X^{n-1}}|P_{X^{n-1}}) \overset{(a)}{=} \mathbb{E}\left[D\left(P_{X_n|X_{n-m}^{n-1}}\Big\|\frac{1}{n}\sum_{t=1}^n \widehat{P}_t(\cdot|X_{n-t+1}^{n-1})\right)\right]$$

$$\overset{(b)}{\leq} \frac{1}{n-m}\sum_{t=m+1}^n \mathbb{E}\left[D(P_{X_n|X_{n-m}^{n-1}}\|\widehat{P}_t(\cdot|X_{n-t+1}^{n-1}))\right]$$

$$\overset{(c)}{=} \frac{1}{n-m}\sum_{t=m+1}^n \mathbb{E}\left[D(P_{X_t|X_{t-m}^{t-1}}\|\widehat{P}_t(\cdot|X^{t-1}))\right]$$

$$\overset{(d)}{=} \frac{1}{n-m}\sum_{t=m+1}^n D(P_{X_t|X^{t-1}}\|Q_{X^t|X^{t-1}}|X^{t-1})$$

$$\leq \frac{1}{n-m}\sum_{t=1}^n D(P_{X_t|X^{t-1}}\|Q_{X^t|X^{t-1}}|X^{t-1})$$

$$\overset{(e)}{=} \frac{1}{n-m}D(P_{X^n}\|Q_{X^n}),$$

where (a) uses the $m^{\text{th}}$-order Markovian assumption; (b) is due to the convexity of the KL divergence; (c) uses the crucial fact that for all $t = 1, \ldots, n-1$, $(X_{n-t}, \ldots, X_{n-1}) \overset{\text{law}}{=} (X_1, \ldots, X_t)$, thanks to stationarity; (d) follows from substituting $\widehat{P}_t(\cdot|x^{t-1}) = Q_{X_t|X^{t-1}=x^{t-1}}$, the Markovian assumption $P_{X_t|X_{t-m}^{t-1}} = P_{X_t|X^{t-1}}$, and rewriting the expectation as conditional KL divergence; (e) is by the chain rule (30) of KL divergence. Since the above holds for any $\theta \in \Theta$, the desired (26) follows which implies that $\mathsf{Risk}_{n-1} \leq \frac{\mathsf{Red}_n}{n-m}$. Finally, $\mathsf{Risk}_{n-1} \leq \mathsf{Risk}_n$ follows from $\mathbb{E}[D(P_{X_{n+1}|X_n}\|\widehat{P}_n(X_2^n))] = \mathbb{E}[D(P_{X_n|X_{n-1}}\|\widehat{P}_n(X_1^{n-1}))]$, since $(X_2, \ldots, X_n)$ and $(X_1, \ldots, X_{n-1})$ are equal in law. $\square$

**Remark 3.** Alternatively, Lemma 5 also follows from the mutual information representation (17) and (22). Indeed, by the chain rule for mutual information,

$$I(\theta; X^n) = \sum_{t=1}^n I(\theta; X_t|X^{t-1}), \tag{32}$$

taking the supremum over $\pi$ (the distribution of $\theta$) on both sides yields (17). For (22), it suffices to show that $I(\theta; X_t|X^{t-1})$ is decreasing in $t$: for any $\theta \sim \pi$,

$$I(\theta; X_{n+1}|X^n) = \mathbb{E}\log\frac{P_{X_{n+1}|X^n,\theta}}{P_{X_{n+1}|X^n}} = \mathbb{E}\log\frac{P_{X_{n+1}|X^n,\theta}}{P_{X_{n+1}|X_2^n}} + \underbrace{\mathbb{E}\log\frac{P_{X_{n+1}|X_2^n}}{P_{X_{n+1}|X^n}}}_{-I(X_1;X_{n+1}|X_2^n)},$$

and the first term is

$$\mathbb{E}\log\frac{P_{X_{n+1}|X^n,\theta}}{P_{X_{n+1}|X_2^n}} = \mathbb{E}\log\frac{P_{X_{n+1}|X_{n-m+1}^n,\theta}}{P_{X_{n+1}|X_2^n}} = \mathbb{E}\log\frac{P_{X_n|X_{n-m}^{n-1},\theta}}{P_{X_n|X^{n-1}}} = I(\theta; X_n|X^{n-1})$$

where the first and second equalities follow from the $m^{\text{th}}$-order Markovity and stationarity, respectively. Taking supremum over $\pi$ yields $\mathsf{Risk}_n \leq \mathsf{Risk}_{n-1}$. Finally, by the chain rule (32), we have $I(\theta; X^n) \geq (n-m)I(\theta; X_n|X^{n-1})$, yielding $\mathsf{Risk}_{n-1} \leq \frac{\mathsf{Red}_n}{n-m}$.

### 5.2 Proof of Lemma 6

In this section we bound the pointwise redundancy of the add-one probability assignment (27) over all (not necessarily stationary) Markov chains on $k$ states. The proof is similar to those of [CS04, Theorems 6.3 and 6.5], which, in turn, follow the arguments of [DMPW81, Sec. III-B]. Specifically, we show that for every Markov chain with transition matrix $M$ and initial distribution $\pi$, and every trajectory $(x_1, \cdots, x_n)$, it holds that

$$\log\frac{\pi(x_1)\prod_{t=1}^{n-1}M(x_{t+1}|x_t)}{Q(x_1, \cdots, x_n)} \leq k(k-1)\left[\log\left(1 + \frac{n}{k(k-1)}\right) + 1\right] + \log k, \tag{33}$$

where we abbreviate the add-one estimator $M_{x^t}(x_{t+1}|x_t)$ defined in (5) as $M(x_{t+1}|x_t)$.

To establish (33), note that $Q(x_1, \cdots, x_n)$ could be equivalently expressed using the empirical counts $N_i$ and $N_{ij}$ in (4) as

$$Q(x_1, \cdots, x_n) = \frac{1}{k} \prod_{i=1}^{k} \frac{\prod_{j=1}^{k} N_{ij}!}{k \cdot (k+1) \cdot \cdots \cdot (N_i + k - 1)}.$$

Note that

$$\prod_{t=1}^{n-1} M(x_{t+1}|x_t) = \prod_{i=1}^{k} \prod_{j=1}^{k} M(j|i)^{N_{ij}} \leq \prod_{i=1}^{k} \prod_{j=1}^{k} (N_{ij}/N_i)^{N_{ij}},$$

where the inequality follows from $\sum_j \frac{N_{ij}}{N_i} \log \frac{N_{ij}/N_i}{M(j|i)} \geq 0$ for each $i$, by the nonnegativity of the KL divergence. Therefore, we have

$$\frac{\pi(x_1) \prod_{t=1}^{n-1} M(x_{t+1}|x_t)}{Q(x_1, \cdots, x_n)} \leq k \cdot \prod_{i=1}^{k} \frac{k \cdot (k+1) \cdot \cdots \cdot (N_i + k - 1)}{N_i^{N_i}} \prod_{j=1}^{k} \frac{N_{ij}^{N_{ij}}}{N_{ij}!}. \tag{34}$$

We claim that: for $n_1, \cdots, n_k \in \mathbb{Z}_+$ and $n = \sum_{i=1}^{k} n_i \in \mathbb{N}$, it holds that

$$\prod_{i=1}^{k} \left(\frac{n_i}{n}\right)^{n_i} \leq \frac{\prod_{i=1}^{k} n_i!}{n!}, \tag{35}$$

with the understanding that $(\frac{0}{n})^0 = 0! = 1$. Applying this claim to (34) gives

$$\log \frac{\pi(x_1) \prod_{t=1}^{n-1} M(x_{t+1}|x_t)}{Q(x_1, \cdots, x_n)} \leq \log k + \sum_{i=1}^{k} \log \frac{k \cdot (k+1) \cdot \cdots \cdot (N_i + k - 1)}{N_i!}$$

$$= \log k + \sum_{i=1}^{k} \sum_{\ell=1}^{N_i} \log\left(1 + \frac{k-1}{\ell}\right)$$

$$\leq \log k + \sum_{i=1}^{k} \int_0^{N_i} \log\left(1 + \frac{k-1}{x}\right) dx$$

$$= \log k + \sum_{i=1}^{k} \left((k-1)\log\left(1 + \frac{N_i}{k-1}\right) + N_i \log\left(1 + \frac{k-1}{N_i}\right)\right)$$

$$\overset{(a)}{\leq} k(k-1)\log\left(1 + \frac{n-1}{k(k-1)}\right) + k(k-1) + \log k,$$

where (a) follows from the concavity of $x \mapsto \log x$, $\sum_{i=1}^{k} N_i = n-1$, and $\log(1+x) \leq x$.

It remains to justify (35), which has a simple information-theoretic proof: Let $T$ denote the collection of sequences $x^n$ in $[k]^n$ whose *type* is given by $(n_1, \ldots, n_k)$. Namely, for each $x^n \in T$, $i$ appears exactly $n_i$ times for each $i \in [k]$. Let $(X_1, \ldots, X_n)$ be drawn uniformly at random from the set $T$. Then

$$\log \frac{n!}{\prod_{i=1}^{k} n_i!} = H(X_1, \ldots, X_n) \overset{(a)}{\leq} \sum_{j=1}^{n} H(X_j) \overset{(b)}{=} n \sum_{i=1}^{k} \frac{n_i}{n} \log \frac{n}{n_i},$$

where (a) follows from the fact that the joint entropy is at most the sum of marginal entropies; (b) is because each $X_j$ is distributed as $(\frac{n_1}{n}, \ldots, \frac{n_k}{n})$.

# 6 Proofs in Section 3

## 6.1 Proof of lower bound for three states

In this section we prove the optimal lower bound in Theorem 7 for three states. Let us start by recalling the following well-known lemma.

**Lemma 8.** *Let $q \sim \text{Uniform}(0,1)$. Conditioned on $q$, let $N \sim Binom(m,q)$. Then the Bayes estimator of $q$ given $N$ is the "add-one" estimator:*

$$\mathbb{E}[q|N] = \frac{N+1}{m+2}$$

*and the Bayes risk is given by*

$$\mathbb{E}[(q - \mathbb{E}[q|N])^2] = \frac{1}{6(m+2)}.$$

*Proof of Theorem 7.* Consider the following Bayesian setting: First, we draw $p$ uniformly at random from $[0, 1 - \frac{1}{n}]$. Then, we generate the sample path $X^n = (X_1, \ldots, X_n)$ of a stationary (uniform) Markov chain with transition matrix $M_p$ as defined in (28). For each $t = 1, \ldots, n-1$, define

$$\mathcal{X}_t = \{x^n : x_1 = \ldots = x_t = 1, x_i \neq 1, i = t+1, \ldots, n\}$$

and let $\mathcal{X} = \cup_{t=1}^n \mathcal{X}_t$. Let $\mu(x^n|p) = \mathbb{P}[X = x^n]$. Then

$$\mu(x^n|p) = \frac{1}{3}\left(1 - \frac{2}{n}\right)^{t-1} \frac{2}{n} p^{N(x^n)} \left(1 - \frac{1}{n} - p\right)^{n-t-1-N(x^n)}, \quad x^n \in \mathcal{X}_t, \qquad (36)$$

where $N(x^n)$ denotes the number of transitions from state 2 to 3 or from 3 to 2. Then

$$\mathbb{P}[X^n \in \mathcal{X}_t] = \frac{1}{3}\left(1 - \frac{2}{n}\right)^{t-1} \frac{2}{n} \sum_{k=0}^{n-t-1} \binom{n-t-1}{k} p^k \left(1 - \frac{1}{n} - p\right)^{n-t-1-k}$$

$$= \frac{1}{3}\left(1 - \frac{2}{n}\right)^{t-1} \frac{2}{n}\left(1 - \frac{1}{n}\right)^{n-t-1} = \frac{2}{3n}\left(1 - \frac{1}{n}\right)^{n-2}\left(1 - \frac{1}{n-1}\right)^{t-1} \qquad (37)$$

and hence

$$\mathbb{P}[X^n \in \mathcal{X}] = \sum_{t=1}^{n-1} \mathbb{P}[X^n \in \mathcal{X}_t] = \frac{2(n-1)}{3n}\left(1 - \frac{1}{n}\right)^{n-2}\left(1 - \left(1 - \frac{1}{n-1}\right)^{n-1}\right) \qquad (38)$$

$$= \frac{2(1 - 1/e)}{3e} + o_n(1).$$

Consider the Bayes estimator (for estimating $p$ under the mean-squared error)

$$\widehat{p}(x^n) = \mathbb{E}[p|x^n] = \frac{\mathbb{E}[p \cdot \mu(x^n|p)]}{\mathbb{E}[\mu(x^n|p)]}.$$

For $x^n \in \mathcal{X}_t$, using (36) we have

$$\widehat{p}(x^n) = \frac{\mathbb{E}\left[p^{N(x^n)+1}\left(1 - \frac{1}{n} - p\right)^{n-t-1-N(x^n)}\right]}{\mathbb{E}\left[p^{N(x^n)}\left(1 - \frac{1}{n} - p\right)^{n-t-1-N(x^n)}\right]}, \quad p \sim \text{Uniform}\left(0, \frac{n-1}{n}\right)$$

$$= \frac{n-1}{n}\frac{\mathbb{E}\left[U^{N(x^n)+1}(1-U)^{n-t-1-N(x^n)}\right]}{\mathbb{E}\left[U^{N(x^n)}(1-U)^{n-t-1-N(x^n)}\right]}, \quad U \sim \text{Uniform}(0,1)$$

$$= \frac{n-1}{n}\frac{N(x^n)+1}{n-t+1},$$

where the last step follows from Lemma 8. From (36), we conclude that conditioned on $X^n \in \mathcal{X}_t$ and on $p$, $N(X^n) \sim \text{Binom}(n-t-1, q)$, where $q = \frac{p}{1-\frac{1}{n}} \sim \text{Uniform}(0,1)$. Applying Lemma 8 (with $m = n-t-1$ and $N = N(X^n)$), we get

$$\mathbb{E}[(p - \widehat{p}(X^n))^2|X^n \in \mathcal{X}_t] = \left(\frac{n-1}{n}\right)^2 \mathbb{E}\left[\left(q - \frac{N(x^n)+1}{n-t+1}\right)^2\right]$$

$$= \left(\frac{n-1}{n}\right)^2 \frac{1}{6(n-t+1)}.$$

Finally, note that conditioned on $X^n \in \mathcal{X}$, the probability of $X^n \in \mathcal{X}_t$ is close to uniform. Indeed, from (37) and (38) we get

$$\mathbb{P}\left[X^n \in \mathcal{X}_t | \mathcal{X}\right] = \frac{1}{n-1} \frac{\left(1 - \frac{1}{n-1}\right)^{t-1}}{1 - \left(1 - \frac{1}{n-1}\right)^{n-1}} \geq \frac{1}{n-1} \left(\frac{1}{e-1} + o_n(1)\right), \quad t = 1, \ldots, n-1.$$

Thus

$$\mathbb{E}[(p - \widehat{p}(X^n))^2 \mathbf{1}_{\{X^n \in \mathcal{X}\}}] = \mathbb{P}\left[X^n \in \mathcal{X}\right] \sum_{t=1}^{n-1} \mathbb{E}[(p - \widehat{p}(X^n))^2 | X^n \in \mathcal{X}_t] \mathbb{P}\left[X^n \in \mathcal{X}_t | \mathcal{X}\right]$$

$$\gtrsim \frac{1}{n-1} \sum_{t=1}^{n-1} \frac{1}{n-t+1} = \Theta\left(\frac{\log n}{n}\right). \tag{39}$$

Finally, we relate (39) formally to the minimax prediction risk under the KL divergence. Consider any predictor $\widehat{M}(\cdot|i)$ (as a function of the sample path $X$) for the $i$th row of $M$, $i = 1, 2, 3$. By Pinsker inequality, we conclude that

$$D(M(\cdot|2)\|\widehat{M}(\cdot|2)) \geq \frac{1}{2}\|M(\cdot|2) - \widehat{M}(\cdot|2)\|_{\ell_1}^2 \geq \frac{1}{2}(p - \widehat{M}(3|2))^2 \tag{40}$$

and similarly, $D(M(\cdot|3)\|\widehat{M}(\cdot|3)) \geq \frac{1}{2}(p - \widehat{M}(2|3))^2$. Abbreviate $\widehat{M}(3|2) \equiv \widehat{p}_2$ and $\widehat{M}(2|3) \equiv \widehat{p}_3$, both functions of $X$. Taking expectations over both $p$ and $X$, the Bayes prediction risk can be bounded as follows

$$\sum_{i=1}^{3} \mathbb{E}[D(M(\cdot|i)\|\widehat{M}(\cdot|i))\mathbf{1}_{\{X_n=i\}}]$$

$$\geq \frac{1}{2}\mathbb{E}[(p - \widehat{p}_2)^2 \mathbf{1}_{\{X_n=2\}} + (p - \widehat{p}_3)^2 \mathbf{1}_{\{X_n=3\}}]$$

$$\geq \frac{1}{2} \sum_{x \in \mathcal{X}} \mu(x^n) \left(\mathbb{E}[(p - \widehat{p}_2)^2 | X = x^n]\mathbf{1}_{\{x_n=2\}} + \mathbb{E}[(p - \widehat{p}_3)^2 | X = x^n]\mathbf{1}_{\{x_n=3\}}\right)$$

$$\geq \frac{1}{2} \sum_{x^n \in \mathcal{X}} \mu(x^n)\mathbb{E}[(p - \widehat{p}(x^n))^2 | X = x^n](\mathbf{1}_{\{x_n=2\}} + \mathbf{1}_{\{x_n=3\}})$$

$$= \frac{1}{2} \sum_{x^n \in \mathcal{X}} \mu(x^n)\mathbb{E}[(p - \widehat{p}(x^n))^2 | X = x^n]$$

$$= \frac{1}{2}\mathbb{E}[(p - \widehat{p}(X))^2 \mathbf{1}_{\{X \in \mathcal{X}\}}] \overset{(39)}{=} \Theta\left(\frac{\log n}{n}\right).$$

$\square$

## 6.2 Proof of lower bound for $k$ states

In this section, we rigorously carry out the lower bound proof sketched in Section 3.2: In Section 6.2.1, we explicitly construct the $k$-state chain which satisfies the desired properties in Section 3.2. In Section 6.2.2, we make the steps in (29) precise and bound the Bayes risk from below by an appropriate mutual information. In Section 6.2.3, we choose a prior distribution on the transition probabilities and prove a lower bound on the resulting mutual information, thereby completing the proof of Theorem 1, with the added bonus that the construction is restricted to irreducible and reversible chains.

### 6.2.1 Construction of the $k$-state chain

We construct a $k$-state chain with the following transition probability matrix:

$$
M = \begin{bmatrix} 1 - \frac{1}{n} & \frac{1}{n(k-1)} & \frac{1}{n(k-1)} & \cdots & \frac{1}{n(k-1)} \\ 1/n & & & & \\ 1/n & & \left(1 - \frac{1}{n}\right) T & & \\ \vdots & & & & \\ 1/n & & & & \end{bmatrix},
\tag{41}
$$

where $T \in \mathbb{R}^{\mathcal{S}_2 \times \mathcal{S}_2}$ is a symmetric stochastic matrix to be chosen later. The transition diagram of $M$ is shown in Figure 2. One can also verify that the spectral gap of $M$ is $\Theta(\frac{1}{n})$.

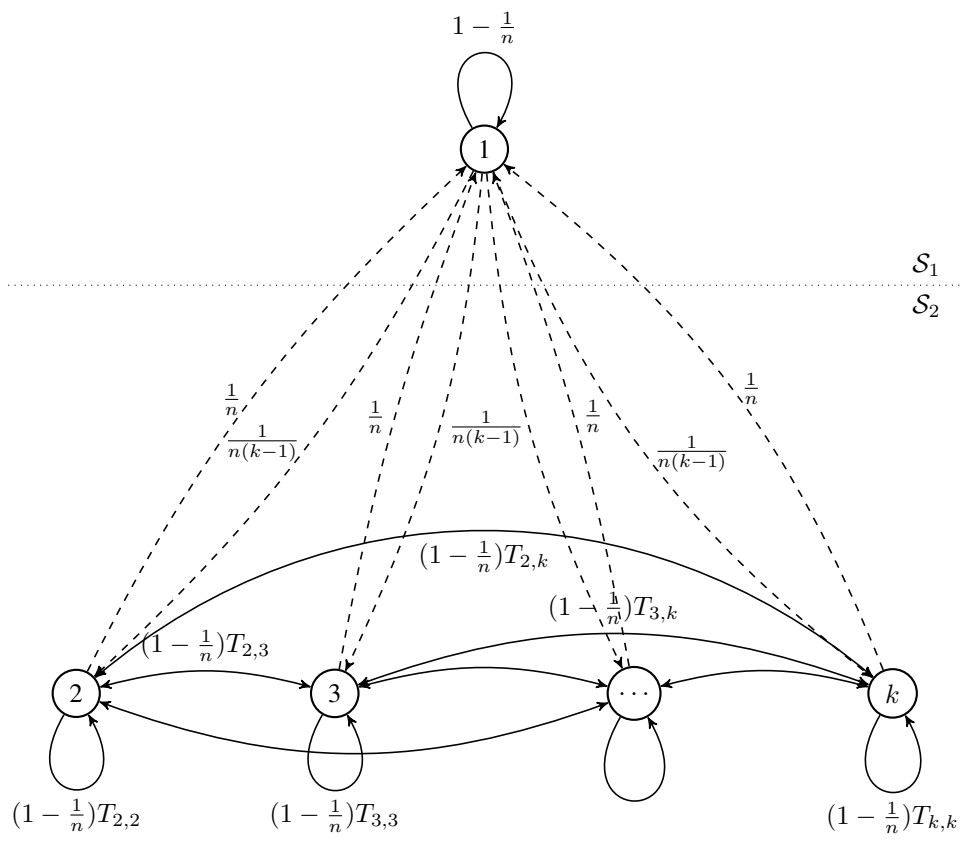

Figure 2: Lower bound construction for $k$-state chains. Solid arrows represent transitions within $\mathcal{S}_1$ and $\mathcal{S}_2$, and dashed arrows represent transitions between $\mathcal{S}_1$ and $\mathcal{S}_2$. The double-headed arrows denote transitions in both directions with equal probabilities.

Let $(X_1, \ldots, X_n)$ be the trajectory of a stationary Markov chain with transition matrix $M$. We observe the following properties:

(P1) This Markov chain is irreducible and reversible, with stationary distribution $(\frac{1}{2}, \frac{1}{2(k-1)}, \cdots, \frac{1}{2(k-1)})$;

(P2) For $t \in [n-1]$, let $\mathcal{X}_t$ denote the collections of trajectories $x^n$ such that $x_1, x_2, \cdots, x_t \in \mathcal{S}_1$ and $x_{t+1}, \cdots, x_n \in \mathcal{S}_2$. Then

$$
\mathbb{P}(X^n \in \mathcal{X}_t) = \mathbb{P}(X_1 = \cdots = X_t = 1) \cdot \mathbb{P}(X_{t+1} \neq 1 | X_t = 1) \cdot \prod_{s=t+1}^{n-1} \mathbb{P}(X_{s+1} \neq 1 | X_s \neq 1)
$$

$$
= \frac{1}{2} \cdot \left(1 - \frac{1}{n}\right)^{t-1} \cdot \frac{1}{n} \cdot \left(1 - \frac{1}{n}\right)^{n-1-t} \geq \frac{1}{2en}.
\tag{42}
$$

Moreover, this probability does not depend of the choice of $T$;

(P3) Conditioned on the event that $X^n \in \mathcal{X}_t$, the trajectory $(X_{t+1}, \cdots, X_n)$ has the same distribution as a length-$(n-t)$ trajectory of a stationary Markov chain with state space $\mathcal{S}_2 = \{2, 3, \cdots, k\}$ and transition probability $T$, and the uniform initial distribution. Indeed,

$$\mathbb{P}\left[X_{t+1} = x_{t+1}, \ldots, X_n = x_n | X^n \in \mathcal{X}_t\right] = \frac{\frac{1}{2} \cdot \left(1 - \frac{1}{n}\right)^{t-1} \cdot \frac{1}{n(k-1)} \prod_{s=t+1}^{n-1} M(x_{s+1}|x_s)}{\frac{1}{2} \cdot \left(1 - \frac{1}{n}\right)^{t-1} \cdot \frac{1}{n} \cdot \left(1 - \frac{1}{n}\right)^{n-1-t}}$$

$$= \frac{1}{k-1} \prod_{s=t+1}^{n-1} T(x_{s+1}|x_s).$$

### 6.2.2 Reducing the Bayes prediction risk to redundancy

Let $\mathcal{M}_{k-1}^{\mathsf{sym}}$ be the collection of all symmetric transition matrices on state space $\mathcal{S}_2 = \{2, \ldots, k\}$. Consider a Bayesian setting where the transition matrix $M$ is constructed in (41) and the submatrix $T$ is drawn from an arbitrary prior on $\mathcal{M}_{k-1}^{\mathsf{sym}}$. The following lemma lower bounds the Bayes prediction risk.

**Lemma 9.** *Conditioned on T, let $Y^n = (Y_1, \ldots, Y_n)$ denote a stationary Markov chain on state space $\mathcal{S}_2$ with transition matrix $T$ and uniform initial distribution. Then*

$$\inf_{\widehat{M}} \mathbb{E}_T \left[\mathbb{E}[D(M(\cdot|X_n)\|\widehat{M}(\cdot|X_n))]\right] \geq \frac{n-1}{2en^2}\left(I(T; Y^n) - \log(k-1)\right).$$

Lemma 9 is the formal statement of the inequality (29) presented in the proof sketch. Maximizing the lower bound over the prior on $T$ and in view of the mutual information representation (17), we obtain the following corollary.

**Corollary 10.** *Let $\mathsf{Risk}_{k,n}^{\mathsf{sym}}$ denote the minimax prediction risk for stationary irreducible and reversible Markov chains on $k$ states and $\mathsf{Red}_{k,n}^{\mathsf{sym}}$ the redundancy for stationary symmetric Markov chains on $k$ states. Then*

$$\mathsf{Risk}_{k,n}^{\mathsf{rev}} \geq \frac{n-1}{2en^2}(\mathsf{Red}_{k-1,n}^{\mathsf{sym}} - \log(k-1)).$$

We make use of the properties (P1)–(P3) in Section 6.2.1 to prove Lemma 9.

*Proof of Lemma 9.* Recall that in the Bayesian setting, we first draw $T$ from some prior on $\mathcal{M}_{k-1}^{\mathsf{sym}}$, then generate the stationary Markov chain $X^n = (X_1, \ldots, X_n)$ with state space $[k]$ and transition matrix $M$ in (41), and $(Y_1, \ldots, Y_n)$ with state space $\mathcal{S}_2 = \{2, \ldots, k\}$ and transition matrix $T$.

We first relate the Bayes estimator of $M$ and $T$ (given the $X$ and $Y$ chain respectively). For clarity, we spell out the explicit dependence of the estimators on the input trajectory. For each $t \in [n]$, denote by $\widehat{M}_t = \widehat{M}_t(\cdot|x^t)$ the Bayes estimator of $M(\cdot|x_t)$ give $X^t = x^t$, and $\widehat{T}_t(\cdot|y^t)$ the Bayes estimator of $T(\cdot|y_t)$ give $Y^t = y^t$. For each $t = 1, \ldots, n-1$ and for each trajectory $x^n = (1, \ldots, 1, x_{t+1}, \ldots, x_n) \in \mathcal{X}_t$, recalling the form (21) of the Bayes estimator, we have, for each $j \in \mathcal{S}_2$,

$$\widehat{M}_n(j|x^n) = \frac{\mathbb{P}\left[X^{n+1} = (x^n, j)\right]}{\mathbb{P}\left[X^n = x^n\right]}$$

$$= \frac{\mathbb{E}[\frac{1}{2}M(1|1)^{t-1}M(x_{t+1}|1)M(x_{t+2}|x_{t+1})\ldots M(x_n|x_{n-1})M(j|x_n)]}{\mathbb{E}[\frac{1}{2}M(1|1)^{t-1}M(x_{t+1}|1)M(x_{t+2}|x_{t+1})\ldots M(x_n|x_{n-1})]}$$

$$= \left(1 - \frac{1}{n}\right)\frac{\mathbb{E}[T(x_{t+2}|x_{t+1})\ldots T(x_n|x_{n-1})T(j|x_n)]}{\mathbb{E}[T(x_{t+2}|x_{t+1})\ldots T(x_n|x_{n-1})]}$$

$$= \left(1 - \frac{1}{n}\right)\widehat{T}_{n-t}(j|x_{t+1}^n),$$

where we used the stationary distribution of $X$ in (P1) and the uniformity of the stationary distribution of $Y$, neither of which depends on $T$. Furthermore, by construction in (41), $\widehat{M}_n(1|x^n) = \frac{1}{n}$ is

deterministic. In all, we have

$$\widehat{M}_n(\cdot|x^n) = \frac{1}{n}\delta_1 + \left(1 - \frac{1}{n}\right)\widehat{T}_{n-t}(\cdot|x_{t+1}^n), \quad x^n \in \mathcal{X}_t. \tag{43}$$

with $\delta_1$ denoting the point mass at state 1, which parallels the fact that

$$M(\cdot|x) = \frac{1}{n}\delta_1 + \left(1 - \frac{1}{n}\right)T(\cdot|x), \quad x \in \mathcal{S}_2. \tag{44}$$

By (P2), each event $\{X^n \in \mathcal{X}_t\}$ occurs with probability at least $1/(2en)$, and is independent of $T$. Therefore,

$$\mathbb{E}_T\left[\mathbb{E}[D(M(\cdot|X_n)\|\widehat{M}(\cdot|X^n))]\right] \geq \frac{1}{2en}\sum_{t=1}^{n-1}\mathbb{E}_T\left[\mathbb{E}[D(M(\cdot|X_n)\|\widehat{M}(\cdot|X^n))|X^n \in \mathcal{X}_t]\right]. \tag{45}$$

By (P3), the conditional joint law of $(T, X_{t+1}, \ldots, X_n)$ on the event $\{X^n \in \mathcal{X}_t\}$ is the same as the joint law of $(T, Y_1, \ldots, Y_{n-t})$. Thus, we may express the Bayes prediction risk in the $X$ chain as

$$\mathbb{E}_T\left[\mathbb{E}[D(M(\cdot|X_n)\|\widehat{M}(\cdot|X^n))|X^n \in \mathcal{X}_t]\right] \overset{\text{(a)}}{=} \left(1 - \frac{1}{n}\right) \cdot \mathbb{E}_T\left[\mathbb{E}[D(T(\cdot|Y_{n-t})\|\widehat{T}(\cdot|Y^{n-t}))]\right]$$

$$\overset{\text{(b)}}{=} \left(1 - \frac{1}{n}\right) \cdot I(T; Y_{n-t+1}|Y^{n-t}), \tag{46}$$

where (a) follows from (43), (44), and the fact that for distributions $P, Q$ supported on $\mathcal{S}_2$, $D(\epsilon\delta_1 + (1-\epsilon)P\|\epsilon\delta_1 + (1-\epsilon)Q) = (1-\epsilon)D(P\|Q)$; (b) is the mutual information representation (20) of the Bayes prediction risk. Finally, the lemma follows from (45), (46), and the chain rule

$$\sum_{t=1}^{n-1} I(T; Y_{n-t+1}|Y^{n-t}) = I(T; Y^n) - I(T; Y_1) \geq I(T; Y^n) - \log(k-1),$$

as $I(T; Y_1) \leq H(Y_1) \leq \log(k-1)$. $\qquad\square$

### 6.2.3 Prior construction and lower bounding the mutual information

In view of Lemma 9, it remains to find a prior on $\mathcal{M}_{k-1}^{\mathsf{sym}}$ for $T$, such that the mutual information $I(T; Y^n)$ is large. We make use of the connection identified in [DMPW81, Dav83, Ris84] between estimation error and mutual information (see also [CS04, Theorem 7.1] for a self-contained exposition). To lower the mutual information, a key step is to find a good estimator $\widehat{T}(Y^n)$ of $T$. This is carried out in the following lemma.

**Lemma 11.** *In the setting of Lemma 9, suppose that $T \in \mathcal{M}_k^{\mathsf{sym}}$ with $T_{ij} \in [\frac{1}{2k}, \frac{3}{2k}]$ for all $i, j \in [k]$. Then there is an estimator $\widehat{T}$ based on $Y^n$ such that*

$$\mathbb{E}[\|\widehat{T} - T\|_{\mathsf{F}}^2] \leq \frac{16k^2}{n-1}.$$

We show how Lemma 11 leads to the desired lower bound on the mutual information $I(T; Y^n)$. Since $k \geq 3$, we may assume that $k - 1 = 2k_0$ is an even integer. Consider the following prior distribution $\pi$ on $T$: let $u = (u_{i,j})_{i,j\in[k_0],i\leq j}$ be iid and uniformly distributed in $[1/(4k_0), 3/(4k_0)]$, and $u_{i,j} = u_{j,i}$ for $i > j$. Let the transition matrix $T$ be given by

$$T_{2i-1,2j-1} = T_{2i,2j} = u_{i,j}, \quad T_{2i-1,2j} = T_{2i,2j-1} = \frac{1}{k_0} - u_{i,j}, \quad \forall i, j \in [k]. \tag{47}$$

It is easy to verify that $T$ is symmetric and a stochastic matrix, and each entry of $T$ is supported in the interval $[1/(4k_0), 3/(4k_0)]$. Since $2k_0 = k - 1$, the condition of Lemma 11 is fulfilled, so there exist estimators $\widehat{T}(Y^n)$ and $\widehat{u}(Y^n)$ such that

$$\mathbb{E}[\|\widehat{u}(Y^n) - u\|_2^2] \leq \mathbb{E}[\|\widehat{T}(Y^n) - T\|_{\mathsf{F}}^2] \leq \frac{64k_0^2}{n-1}. \tag{48}$$

Here and below, we identify $u$ and $\widehat{u}$ as $\frac{k_0(k_0+1)}{2}$-dimensional vectors.

Let $h(X) = \int -f_X(x) \log f_X(x) dx$ denote the differential entropy of a continuous random vector $X$ with density $f_X$ w.r.t the Lebesgue measure and $h(X|Y) = \int -f_{XY}(xy) \log f_{X|Y}(x|y) dx dy$ the conditional differential entropy (cf. e.g. [CT06]). Then

$$h(u) = \sum_{i,j\in[k_0], i\leq j} h(u_{i,j}) = -\frac{k_0(k_0+1)}{2} \log(2k_0). \tag{49}$$

Then

$$
\begin{aligned}
I(T;Y^n) &\overset{(a)}{=} I(u;Y^n) \\
&\overset{(b)}{\geq} I(u;\widehat{u}(Y^n)) = h(u) - h(u|\widehat{u}(Y^n)) \\
&\overset{(c)}{\geq} h(u) - h(u - \widehat{u}(Y^n)) \\
&\overset{(d)}{\geq} \frac{k_0(k_0+1)}{4} \log\left(\frac{n-1}{1024\pi e k_0^2}\right) \geq \frac{k^2}{16} \log\left(\frac{n-1}{256\pi e k^2}\right).
\end{aligned}
$$

where (a) is because $u$ and $T$ are in one-to-one correspondence by (47); (b) follows from the data processing inequality; (c) is because $h(\cdot)$ is translation invariant and concave; (d) follows from the maximum entropy principle [CT06]: $h(u - \widehat{u}(Y^n)) \leq \frac{k_0(k_0+1)}{4} \log\left(\frac{2\pi e}{k_0(k_0+1)/2} \cdot \mathbb{E}[\|\widehat{u}(Y^n) - u\|_2^2]\right)$, which in turn is bounded by (48). Plugging this lower bound into Lemma 9 completes the lower bound proof of Theorem 1.

*Proof of Lemma 11.* Since $T$ is symmetric, the stationary distribution is uniform, and there is a one-to-one correspondence between the joint distribution of $(Y_1, Y_2)$ and the transition probabilities. Motivated by this observation, consider the following estimator $\widehat{T}$: for $i, j \in [k]$, let

$$\widehat{T}_{ij} = k \cdot \frac{\sum_{t=1}^n \mathbf{1}_{\{Y_t=i, Y_{t+1}=j\}}}{n-1}.$$

Clearly $\mathbb{E}[\widehat{T}_{ij}] = k \cdot \mathbb{P}(Y_1 = i, Y_2 = j) = T_{ij}$. The following variance bound is shown in [TJW18, Lemma 7, Lemma 8] using the concentration inequality of [Pau15]:

$$\mathrm{Var}(\widehat{T}_{ij}) \leq k^2 \cdot \frac{8 T_{ij} k^{-1}}{\gamma_*(T)(n-1)},$$

where $\gamma_*(T)$ is the absolute spectral gap of $T$ defined in (8). Note that $T = k^{-1}\mathbf{J} + \Delta$, where $\mathbf{J}$ is the all-one matrix and each entry of $\Delta$ lying in $[-1/(2k), 1/(2k)]$. Thus the spectral radius of $\Delta$ is at most $1/2$ and thus $\gamma_*(T) \geq 1/2$. Consequently, we have

$$\mathbb{E}[\|\widehat{T} - T\|_{\mathsf{F}}^2] = \sum_{i,j\in[k]} \mathrm{Var}(\widehat{T}_{ij}) \leq \sum_{i,j\in[k]} \frac{16k T_{ij}}{n-1} = \frac{16k^2}{n-1},$$

completing the proof. □

## 7 Proofs of spectral gap-dependent risk bounds

### 7.1 Two states

To show Theorem 2, let us prove a refined version. In addition to the absolute spectral gap defined in (8), define the spectral gap

$$\gamma \triangleq 1 - \lambda_2 \tag{50}$$

and $\mathcal{M}'_k(\gamma_0)$ the collection of transition matrices whose spectral gap exceeds $\gamma_0$. Paralleling $\mathsf{Risk}_{k,n}(\gamma_0)$ defined in (9), define $\mathsf{Risk}'_{k,n}(\gamma_0)$ as the minimax prediction risk restricted to $M \in \mathcal{M}'_k(\gamma_0)$ Since $\gamma \geq \gamma^*$, we have $\mathcal{M}_k(\gamma_0) \subseteq \mathcal{M}'_k(\gamma_0)$ and hence $\mathsf{Risk}'_{k,n}(\gamma_0) \geq \mathsf{Risk}_{k,n}(\gamma_0)$. Nevertheless, the next result shows that for $k = 2$ they have the same rate:

**Theorem 12** (Spectral gap dependent rates for binary chain). *For any $\gamma_0 \in (0, 1)$*

$$\mathsf{Risk}_{2,n}(\gamma_0) \asymp \mathsf{Risk}'_{2,n}(\gamma_0) \asymp \frac{1}{n} \max \left\{ 1, \log \log \left( \min \left\{ n, \frac{1}{\gamma_0} \right\} \right) \right\}.$$

We first prove the upper bound on $\mathsf{Risk}'_{2,n}$. Note that it is enough to show

$$\mathsf{Risk}'_{2,n}(\gamma_0) \lesssim \frac{\log \log (1/\gamma_0)}{n}, \quad \text{if } n^{-0.9} \leq \gamma_0 \leq e^{-e^5}. \tag{51}$$

Indeed, for any $\gamma_0 \leq n^{-0.9}$, the upper bound $O\left(\log \log n / n\right)$ proven in [FOPS16], which does not depend on the spectral gap, suffices; for any $\gamma_0 > e^{-e^5}$, by monotonicity we can use the upper bound $\mathsf{Risk}'_{2,n}(e^{-e^5})$.

We now define an estimator that achieves (51). Following [FOPS16], consider trajectories with a single transition, namely, $\left\{ 2^{n-\ell}1^\ell, 1^{n-\ell}2^\ell : 1 \leq \ell \leq n - 1 \right\}$, where $2^{n-\ell}1^\ell$ denotes the trajectory $(x_1, \cdots, x_n)$ with $x_1 = \cdots = x_{n-\ell} = 2$ and $x_{n-\ell+1} = \cdots = x_n = 1$. We refer to this type of $x^n$ as *step sequences*. For all non-step sequences $x^n$, we apply the add-$\frac{1}{2}$ estimator similar to (5), namely

$$\widehat{M}_{x^n}(j|i) = \frac{N_{ij} + \frac{1}{2}}{N_i + 1}, \qquad i, j \in \{1, 2\},$$

where the empirical counts $N_i$ and $N_{ij}$ are defined in (4); for step sequences of the form $2^{n-\ell}1^\ell$, we estimate by

$$\widehat{M}_\ell(2|1) = 1/(\ell \log(1/\gamma_0)), \quad \widehat{M}_\ell(1|1) = 1 - \widehat{M}_\ell(2|1). \tag{52}$$

The other type of step sequences $1^{n-\ell}2^\ell$ are dealt with by symmetry.

Due to symmetry it suffices to analyze the risk for sequences ending in 1. The risk of add-$\frac{1}{2}$ estimator for the non-step sequence $1^n$ is bounded as

$$\mathbb{E}\left[ \mathbf{1}_{\{X^n = 1^n\}} D(M(\cdot|1) \| \widehat{M}_{1^n}(\cdot|1)) \right] = P_{X^n}(1^n) \left\{ M(2|1) \log \left( \frac{M(2|1)}{1/(2n)} \right) + M(1|1) \log \left( \frac{M(1|1)}{(n - \frac{1}{2})/n} \right) \right\}$$

$$\leq (1 - M(2|1))^{n-1} \left\{ 2M(2|1)^2 n + \log \left( \frac{n}{n - \frac{1}{2}} \right) \right\} \lesssim \frac{1}{n}.$$

where the last step followed by using $(1 - x)^{n-1} x^2 \leq n^{-2}$ with $x = M(2|1)$ and $\log x \leq x - 1$. From [FOPS16, Lemma 7,8] we have that the total risk of other non-step sequences is bounded from above by $O\left(\frac{1}{n}\right)$ and hence it is enough to analyze the risk for step sequences, and further by symmetry, those in $\left\{ 2^{n-\ell}1^\ell : 1 \leq \ell \leq n - 1 \right\}$. The desired upper bound (51) then follows from Lemma 13 next.

**Lemma 13.** *For any $n^{-0.9} \leq \gamma_0 \leq e^{-e^5}$, $\widehat{M}_\ell(\cdot|1)$ in (52) satisfies*

$$\sup_{M \in \mathcal{M}'_2(\gamma_0)} \sum_{\ell=1}^{n-1} \mathbb{E}\left[ \mathbf{1}_{\{X^n = 2^{n-\ell}1^\ell\}} D(M(\cdot|1) \| \widehat{M}_\ell(\cdot|1)) \right] \lesssim \frac{\log \log(1/\gamma_0)}{n}.$$

*Proof.* For each $\ell$ using $\log \left( \frac{1}{1-x} \right) \leq 2x, x \leq \frac{1}{2}$ with $x = \frac{1}{\ell \log(1/\gamma_0)}$,

$$D(M(\cdot|1) \| \widehat{M}_\ell(\cdot|1)) = M(1|1) \log \left( \frac{M(1|1)}{1 - \frac{1}{\ell \log(1/\gamma_0)}} \right) + M(2|1) \log (M(2|1) \ell \log(1/\gamma_0))$$

$$\lesssim \frac{1}{\ell \log(1/\gamma_0)} + M(2|1) \log(M(2|1)\ell) + M(2|1) \log \log(1/\gamma_0)$$

$$\leq \frac{1}{\ell \log(1/\gamma_0)} + M(2|1) \log_+(M(2|1)\ell) + M(2|1)\log \log(1/\gamma_0), \tag{53}$$

where we define $\log_+(x) = \max\{1, \log x\}$. Recall the following Chebyshev's sum inequality: for $a_1 \leq a_2 \leq \cdots \leq a_n$ and $b_1 \geq b_2 \geq \cdots \geq b_n$, it holds that

$$\sum_{i=1}^{n} a_i b_i \leq \frac{1}{n} \left( \sum_{i=1}^{n} a_i \right) \left( \sum_{i=1}^{n} b_i \right).$$

The following inequalities are thus direct corollaries: for $x, y \in [0, 1]$,

$$\sum_{\ell=1}^{n-1} x(1-x)^{n-\ell-1} y(1-y)^{\ell-1} \leq \frac{1}{n-1} \left( \sum_{\ell=1}^{n-1} x(1-x)^{n-\ell-1} \right) \left( \sum_{\ell=1}^{n-1} y(1-y)^{\ell-1} \right)$$

$$\leq \frac{1}{n-1}, \tag{54}$$

$$\sum_{\ell=1}^{n-1} x(1-x)^{n-\ell-1} y(1-y)^{\ell-1} \log_+(\ell y) \leq \frac{1}{n-1} \left( \sum_{\ell=1}^{n-1} x(1-x)^{n-\ell-1} \right) \left( \sum_{\ell=1}^{n-1} y(1-y)^{\ell-1} \log_+(\ell y) \right)$$

$$\leq \frac{1}{n-1} \sum_{\ell=1}^{n-1} y(1-y)^{\ell-1}(1+\ell y) \leq \frac{2}{n-1}, \tag{55}$$

where in (55) we need to verify that $\ell \mapsto y(1-y)^{\ell-1} \log_+(\ell y)$ is non-increasing. To verify it, w.l.o.g. we may assume that $(\ell+1)y \geq e$, and therefore

$$\frac{y(1-y)^{\ell} \log_+((\ell+1)y)}{y(1-y)^{\ell-1} \log_+(\ell y)} = \frac{(1-y) \log((\ell+1)y)}{\log_+(\ell y)} \leq \left( 1 - \frac{e}{\ell+1} \right) \left( 1 + \frac{\log(1+1/\ell)}{\log_+(\ell y)} \right)$$

$$\leq \left( 1 - \frac{e}{\ell+1} \right) \left( 1 + \frac{1}{\ell} \right) < 1 + \frac{1}{\ell} - \frac{e}{\ell+1} < 1.$$

Therefore,

$$\sum_{\ell=1}^{n-1} \mathbb{E} \left[ \mathbf{1}_{\{X^n = 2^{n-\ell} 1^{\ell}\}} D(M(\cdot|1) \| \widehat{M}_{\ell}(\cdot|1)) \right]$$

$$\leq \sum_{\ell=1}^{n-1} M(2|2)^{n-\ell-1} M(1|2) M(1|1)^{\ell-1} D(M(\cdot|1) \| \widehat{M}_{\ell}(\cdot|1))$$

$$\overset{(a)}{\lesssim} \sum_{\ell=1}^{n-1} M(2|2)^{n-\ell-1} M(1|2) M(1|1)^{\ell-1} \left( \frac{1}{\ell \log(1/\gamma_0)} + M(2|1) \log_+(M(2|1)\ell) + M(2|1) \log\log(1/\gamma_0) \right)$$

$$\overset{(b)}{\leq} \sum_{\ell=1}^{n-1} \frac{M(2|2)^{n-\ell-1} M(1|2) M(1|1)^{\ell-1}}{\ell \log(1/\gamma_0)} + \frac{2 + \log\log(1/\gamma_0)}{n-1}, \tag{56}$$

where (a) is due to (53), (b) follows from (54) and (55) applied to $x = M(1|2), y = M(2|1)$. To deal with the remaining sum, we distinguish into two cases. Sticking to the above definitions of $x$ and $y$, if $y > \gamma_0/2$, then

$$\sum_{\ell=1}^{n-1} \frac{x(1-x)^{n-\ell-1}(1-y)^{\ell-1}}{\ell} \leq \frac{1}{n-1} \left( \sum_{\ell=1}^{n-1} x(1-x)^{n-\ell-1} \right) \left( \sum_{\ell=1}^{n-1} \frac{(1-y)^{\ell-1}}{\ell} \right) \leq \frac{\log(2/\gamma_0)}{n-1},$$

where the last step has used that $\sum_{\ell=1}^{\infty} t^{\ell-1}/\ell = \log(1/(1-t))$ for $|t| < 1$. If $y \leq \gamma_0/2$, notice that for two-state chain the spectral gap is given explicitly by $\gamma = M(1|2) + M(2|1) = x + y$, so that the assumption $\gamma \geq \gamma_0$ implies that $x \geq \gamma_0/2$. In this case,

$$\sum_{\ell=1}^{n-1} \frac{x(1-x)^{n-\ell-1}(1-y)^{\ell-1}}{\ell} \leq \sum_{\ell < n/2} (1-x)^{n/2-1} + \sum_{\ell \geq n/2} \frac{x(1-x)^{n-\ell-1}}{n/2}$$

$$\leq \frac{n}{2} e^{-(n/2-1)\gamma_0} + \frac{2}{n} \lesssim \frac{1}{n},$$

thanks to the assumption $\gamma_0 \geq n^{-0.9}$. Therefore, in both cases, the first term in (56) is $O(1/n)$, as desired. $\qquad \square$

Next we prove the lower bound on $\mathsf{Risk}_{2,n}$. It is enough to show that $\mathsf{Risk}_{2,n}(\gamma_0) \gtrsim \frac{1}{n} \log\log(1/\gamma_0)$ for $n^{-1} \leq \gamma_0 \leq e^{-e^5}$. Indeed, for $\gamma_0 \geq e^{-e^5}$, we can apply the result in the i.i.d. setting (see, e.g.,

[BFSS02]), in which the absolute spectral gap is 1, to obtain the usual parametric-rate lower bound $\Omega\left(\frac{1}{n}\right)$; for $\gamma_0 < n^{-1}$, we simply bound $\mathsf{Risk}_{2,n}(\gamma_0)$ from below by $\mathsf{Risk}_{2,n}(n^{-1})$. Define

$$\alpha = \log(1/\gamma_0), \quad \beta = \left\lceil \frac{\alpha}{5 \log \alpha} \right\rceil, \tag{57}$$

and consider the prior distribution

$$\mathscr{M} = \mathrm{Uniform}(\mathcal{M}), \quad \mathcal{M} = \left\{ M : M(1|2) = \frac{1}{n}, M(2|1) = \frac{1}{\alpha^m} : m \in \mathbb{N} \cap (\beta, 5\beta) \right\}. \tag{58}$$

Then the lower bound part of Theorem 2 follows from the next lemma.

**Lemma 14.** *Assume that $n^{-0.9} \leq \gamma_0 \leq e^{-e^5}$. Then*

(i) $\gamma_* > \gamma_0$ *for each $M \in \mathcal{M}$;*

(ii) *the Bayes risk with respect to the prior $\mathscr{M}$ is at least $\Omega\left(\frac{\log\log(1/\gamma_0)}{n}\right)$.*

*Proof.* Part (i) follows by noting that absolute spectral gap for any two states matrix $M$ is $1 - |1 - M(2|1) - M(1|2)|$ and for any $M \in \mathcal{M}$, $M(2|1) \in \left(\alpha^{-5\beta}, \alpha^{-\beta}\right) \subseteq \left(\gamma_0, \gamma_0^{1/5}\right) \subseteq (\gamma_0, 1/2)$ which guarantees $\gamma_* = M(1|2) + M(2|1) > \gamma_0$.

To show part (ii) we lower bound the Bayes risk when the observed trajectory $X^n$ is a step sequence in $\left\{2^{n-\ell}1^\ell : 1 \leq \ell \leq n-1\right\}$. Our argument closely follows that of [HOP18, Theorem 1]. Since $\gamma_0 \geq n^{-1}$, for each $M \in \mathcal{M}$, the corresponding stationary distribution $\pi$ satisfies

$$\pi_2 = \frac{M(2|1)}{M(2|1) + M(1|2)} \geq \frac{1}{2}.$$

Denote by $\mathsf{Risk}(\mathscr{M})$ the Bayes risk with respect to the prior $\mathscr{M}$ and by $\widehat{M}_\ell^{\mathsf{B}}(\cdot|1)$ the Bayes estimator for prior $\mathscr{M}$ given $X^n = 2^{n-\ell}1^\ell$. Note that

$$\mathbb{P}\left[X^n = 2^{n-\ell}1^\ell\right] = \pi_2 \left(1 - \frac{1}{n}\right)^{n-\ell-1} \frac{1}{n} M(1|1)^{\ell-1} \geq \frac{1}{2en} M(1|1)^{\ell-1}. \tag{59}$$

Then

$$\mathsf{Risk}(\mathscr{M}) \geq \mathbb{E}_{M \sim \mathscr{M}} \left[ \sum_{\ell=1}^{n-1} \mathbb{E}\left[ \mathbf{1}_{\{X^n = 2^{n-\ell}1^\ell\}} D(M(\cdot|1) \| \widehat{M}_\ell^{\mathsf{B}}(\cdot|1)) \right] \right]$$

$$\geq \mathbb{E}_{M \sim \mathscr{M}} \left[ \sum_{\ell=1}^{n-1} \frac{M(1|1)^{\ell-1}}{2en} D(M(\cdot|1) \| \widehat{M}_\ell^{\mathsf{B}}(\cdot|1)) \right]$$

$$= \frac{1}{2en} \sum_{\ell=1}^{n-1} \mathbb{E}_{M \sim \mathscr{M}} \left[ M(1|1)^{\ell-1} D(M(\cdot|1) \| \widehat{M}_\ell^{\mathsf{B}}(\cdot|1)) \right]. \tag{60}$$

Recalling the general form of the Bayes estimator in (21) and in view of (59), we get

$$\widehat{M}_\ell^{\mathsf{B}}(2|1) = \frac{\mathbb{E}_{M \sim \mathscr{M}}[M(1|1)^{\ell-1} M(2|1)]}{\mathbb{E}_{M \sim \mathscr{M}}[M(1|1)^{\ell-1}]}, \quad \widehat{M}_\ell^{\mathsf{B}}(1|1) = 1 - \widehat{M}_\ell^{\mathsf{B}}(2|1). \tag{61}$$

Plugging (61) into (60), and using

$$D((x, 1-x) \| (y, 1-y)) = x \log \frac{x}{y} + (1-x) \log \frac{1-x}{1-y} \geq x \max\left\{0, \log \frac{x}{y} - 1\right\},$$

we arrive at the following lower bound for the Bayes risk:

$\mathsf{Risk}(\mathscr{M})$

$$\geq \frac{1}{2en} \sum_{\ell=1}^{n-1} \mathbb{E}_{M \sim \mathscr{M}} \left[ M(1|1)^{\ell-1} M(2|1) \max\left\{0, \log\left(\frac{M(2|1) \cdot \mathbb{E}_{M \sim \mathscr{M}}[M(1|1)^{\ell-1}]}{\mathbb{E}_{M \sim \mathscr{M}}[M(1|1)^{\ell-1} M(2|1)]}\right) - 1\right\} \right]. \tag{62}$$

Under the prior $\mathscr{M}$, $M(2|1) = 1 - M(1|1) = \alpha^{-m}$ with $\beta \leq m \leq 5\beta$.

We further lower bound (62) by summing over an appropriate range of $\ell$. For any $m \in [\beta, 3\beta]$, define

$$\ell_1(m) = \left\lceil \frac{\alpha^m}{\log \alpha} \right\rceil, \qquad \ell_2(m) = \lfloor \alpha^m \log \alpha \rfloor.$$

Since $\gamma_0 \leq e^{-e^5}$, our choice of $\alpha$ ensures that the intervals $\{[\ell_1(m), \ell_2(m)]\}_{\beta \leq m \leq 3\beta}$ are disjoint. We will establish the following claim: for all $m \in [\beta, 3\beta]$ and $\ell \in [\ell_1(m), \ell_2(m)]$, it holds that

$$\frac{\alpha^{-m} \cdot \mathbb{E}_{M \sim \mathscr{M}}[M(1|1)^{\ell-1}]}{\mathbb{E}_{M \sim \mathscr{M}}[M(1|1)^{\ell-1} M(2|1)]} \gtrsim \frac{\log(1/\gamma_0)}{\log \log(1/\gamma_0)}. \tag{63}$$

We first complete the proof of the Bayes risk bound assuming (63). Using (62) and (63), we have

$$\mathsf{Risk}(\mathscr{M}) \gtrsim \frac{1}{n} \cdot \frac{1}{4\beta} \sum_{m=\beta}^{3\beta} \sum_{\ell=\ell_1(m)}^{\ell_2(m)} \alpha^{-m}(1 - \alpha^{-m})^{\ell-1} \cdot \log \log(1/\gamma_0)$$

$$= \frac{\log \log(1/\gamma_0)}{4n\beta} \sum_{m=\beta}^{3\beta} \left\{ (1 - \alpha^{-m})^{\ell_1(m)-1} - (1 - \alpha^{-m})^{\ell_2(m)} \right\}$$

$$\overset{(a)}{\geq} \frac{\log \log(1/\gamma_0)}{4n\beta} \sum_{m=\beta}^{3\beta} \left( \left(\frac{1}{4}\right)^{\frac{1}{\log \alpha}} - \left(\frac{1}{e}\right)^{-1+\log \alpha} \right) \gtrsim \frac{\log \log(1/\gamma_0)}{n},$$

with (a) following from $\frac{1}{4} \leq (1-x)^{\frac{1}{x}} \leq \frac{1}{e}$ if $x \leq \frac{1}{2}$, and $\alpha^{-m} \leq \alpha^{-\beta} \leq \gamma_0^{1/5} \leq \frac{1}{2}$.

Next we prove the claim (63). Expanding the expectation in (58), we write the LHS of (63) as

$$\frac{\alpha^{-m} \cdot \mathbb{E}_{M \sim \mathscr{M}}[M(1|1)^{\ell-1}]}{\mathbb{E}_{M \sim \mathscr{M}}[M(1|1)^{\ell-1} M(2|1)]} = \frac{X_\ell + A_\ell + B_\ell}{X_\ell + C_\ell + D_\ell},$$

where

$$X_\ell = \left(1 - \alpha^{-m}\right)^\ell, \quad A_\ell = \sum_{j=\beta}^{m-1} \left(1 - \alpha^{-j}\right)^\ell, \quad B_\ell = \sum_{j=m+1}^{5\beta} \left(1 - \alpha^{-j}\right)^\ell,$$

$$C_\ell = \sum_{j=\beta}^{m-1} \left(1 - \alpha^{-j}\right)^\ell \alpha^{m-j}, \quad D_\ell = \sum_{j=m+1}^{5\beta} \left(1 - \alpha^{-j}\right)^\ell \alpha^{m-j}.$$

We bound each of the terms individually. Clearly, $X_\ell \in (0,1)$ and $A_\ell \geq 0$. Thus it suffices to show that $B_\ell \gtrsim \beta$ and $C_\ell, D_\ell \lesssim 1$, for $m \in [\beta, 3\beta]$ and $\ell_1(m) \leq \ell \leq \ell_2(m)$. Indeed,

- For $j \geq m+1$, we have

$$\left(1 - \alpha^{-j}\right)^\ell \geq \left(1 - \alpha^{-j}\right)^{\ell_2(m)} \overset{(a)}{\geq} (1/4)^{\frac{\ell_2(m)}{\alpha^j}} \geq (1/4)^{\frac{\log \alpha}{\alpha}} \geq 1/4,$$

  where in (a) we use the inequality $(1-x)^{1/x} \geq 1/4$ for $x \leq 1/2$. Consequently, $B_\ell \geq \beta/2$;

- For $j \leq m-1$, we have

$$\left(1 - \alpha^{-j}\right)^\ell \leq \left(1 - \alpha^{-j}\right)^{\ell_1(m)} \overset{(b)}{\leq} e^{-\frac{\alpha^{m-j}}{\log \alpha}} = \gamma_0^{\frac{\alpha^{m-j-1}}{\log \alpha}},$$

  where (b) follows from $(1-x)^{1/x} \leq 1/e$ and the definition of $\ell_1(m)$. Consequently,

$$C_\ell \leq \gamma_0^{\frac{\alpha}{\log \alpha}} \sum_{j=\beta}^{m-2} \alpha^{m-j} + \alpha \gamma_0^{\frac{1}{\log \alpha}} \leq e^{-\frac{\alpha^2}{\log \alpha} + (2\beta+1)\log \alpha} + e^{\log \alpha - \frac{\alpha}{\log \alpha}} \leq 2,$$

  where the last step uses the definition of $\beta$ in (57);

- $D_\ell \leq \sum_{j=m+1}^{5\beta} \alpha^{m-j} \leq 1$, since $\alpha = \log \frac{1}{\gamma_0} \geq e^5$.

Combining the above bounds completes the proof of (63). $\qquad\square$

## 7.2 $k$ states

### 7.2.1 Proof of Theorem 3 (i)

Notice that the prediction problem consists of $k$ sub-problems of estimating the individual rows of $M$, so it suffices show the contribution from each of them is $O\left(\frac{k}{n}\right)$. In particular, assuming the chain terminates in state 1 we bound the risk of estimating the first row by the add-one estimator $\widehat{M}^{+1}(j|1) = \frac{N_{1j}+1}{N_1+k}$. Under the absolute spectral gap condition of $\gamma_* \geq \gamma_0$, we show

$$\mathbb{E}\left[\mathbf{1}_{\{X_n=1\}}D\left(M(\cdot|1)\|\widehat{M}^{+1}(\cdot|1)\right)\right] \lesssim \frac{k}{n}\left(1 + \sqrt{\frac{\log k}{k\gamma_0^4}}\right). \tag{64}$$

By symmetry, we get the desired $\mathrm{Risk}_{k,n}(\gamma_0) \lesssim \frac{k^2}{n}\left(1 + \sqrt{\frac{\log k}{k\gamma_0^4}}\right)$. The basic steps of our analysis are as follows:

- When $N_1$ is substantially smaller than its mean, we can bound the risk using the worst-case risk bound for add-one estimators and the probability of this rare event.

- Otherwise, we decompose the prediction risk as

$$D(M(\cdot|1)\|\widehat{M}^{+1}(\cdot|1)) = \sum_{j=1}^{k}\left[M(j|1)\log\left(\frac{M(j|1)(N_1+k)}{N_{1j}+1}\right) - M(j|1) + \frac{N_{1j}+1}{N_1+k}\right].$$

We then analyze each term depending on whether $N_{1j}$ is typical or not. Unless $N_{1j}$ is atypically small, the add-one estimator works well whose risk can be bounded quadratically.

To analyze the concentration of the empirical counts we use the following moment bounds. The proofs are deferred to Appendix B.

**Lemma 15.** *Finite reversible and irreducible chains observe the following moment bounds:*

*(i)* $\mathbb{E}\left[(N_{ij} - N_i M(j|i))^2 \,|\, X_n = i\right] \lesssim n\pi_i M(j|i)(1 - M(j|i)) + \frac{\sqrt{M(j|i)}}{\gamma_*} + \frac{M(j|i)}{\gamma_*^2}$

*(ii)* $\mathbb{E}\left[(N_{ij} - N_i M(j|i))^4 \,|\, X_n = i\right] \lesssim (n\pi_i M(j|i)(1 - M(j|i)))^2 + \frac{\sqrt{M(j|i)}}{\gamma_*} + \frac{M(j|i)^2}{\gamma_*^4}$

*(iii)* $\mathbb{E}\left[(N_i - (n-1)\pi_i)^4 \,|\, X_n = i\right] \lesssim \frac{n^2\pi_i^2}{\gamma_*^2} + \frac{1}{\gamma_*^4}$.

When $\gamma_*$ is high this shows that the moments behave as if for each $i \in [k]$, $N_1$ is approximately Binomial$(n-1, \pi_i)$ and $N_{ij}$ is approximately Binomial$(N_i, M(j|i))$, which happens in case of i.i.d. sampling. For i.i.d. models [KOPS15] showed that the add-one estimator achieves $O\left(\frac{k}{n}\right)$ risk bound which we aim here too. In addition, dependency of the above moments on $\gamma_*$ gives rise to sufficient conditions that guarantees parametric rate. The technical details are given below.

We decompose the left hand side in (64) based on $N_1$ as

$$\mathbb{E}\left[\mathbf{1}_{\{X_n=1\}}D\left(M(\cdot|1)\|\widehat{M}^{+1}(\cdot|1)\right)\right] = \mathbb{E}\left[\mathbf{1}_{\{A^\leq\}}D\left(M(\cdot|1)\|\widehat{M}^{+1}(\cdot|1)\right)\right] + \mathbb{E}\left[\mathbf{1}_{\{A^>\}}D\left(M(\cdot|1)\|\widehat{M}^{+1}(\cdot|1)\right)\right]$$

where the typical set $A^>$ and atypical set $A^\leq$ are defined as

$$A^\leq \triangleq \{X_n = 1, N_1 \leq (n-1)\pi_1/2\}, \quad A^> \triangleq \{X_n = 1, N_1 > (n-1)\pi_1/2\}.$$

For the atypical case, note the following deterministic property of the add-one estimator. Let $\widehat{Q}$ be an add-one estimator with sample size $n$ and alphabet size $k$ of the form $\widehat{Q}_i = \frac{n_i+1}{n+k}$, where $\sum n_i = n$. Since $\widehat{Q}$ is bounded below by $\frac{1}{n+k}$ everywhere, for any distribution $P$, we have

$$D(P\|\widehat{Q}) \leq \log(n+k). \tag{65}$$

Applying this bound on the event $A^{\leq}$, we have

$$\mathbb{E}\left[\mathbf{1}_{\{A^{\leq}\}}D\left(M(\cdot|1)\|\widehat{M}^{+1}(\cdot|1)\right)\right]$$

$$\leq \log\left(n\pi_1+k\right)\mathbb{P}\left[X_n=1, N_1\leq(n-1)\pi_1/2\right]$$

$$\overset{(a)}{\lesssim} \mathbf{1}_{\{n\pi_1\gamma_*\leq10\}}\pi_1\log\left(n\pi_1+k\right)+\mathbf{1}_{\{n\pi_1\gamma_*>10\}}\pi_1\log\left(n\pi_1+k\right)\frac{\mathbb{E}\left[\left(N_1-(n-1)\pi_1\right)^4\big|X_n=1\right]}{n^4\pi_1^4}$$

$$\tag{66}$$

$$\overset{(b)}{\leq} \mathbf{1}_{\{n\pi_1\gamma_*\leq10\}}\frac{10}{n\gamma_*}\log\left(\frac{10}{\gamma_*}+k\right)+\mathbf{1}_{\{n\pi_1\gamma_*>10\}}\log\left(n\pi_1+k\right)\left(\frac{1}{n^2\pi_1\gamma_*^2}+\frac{1}{n^4\pi_1^3\gamma_*^4}\right)$$

$$\overset{(c)}{\lesssim} \frac{1}{n}\left\{\mathbf{1}_{\{n\pi_1\gamma_*\leq10\}}\frac{\log(1/\gamma_*)+\log k}{\gamma_*}+\mathbf{1}_{\{n\pi_1\gamma_*>10\}}\left(n\pi_1+\log k\right)\left(\frac{1}{n\pi_1\gamma_*^2}+\frac{1}{n^3\pi_1^3\gamma_*^4}\right)\right\}$$

$$\lesssim\frac{1}{n}\left\{\mathbf{1}_{\{n\pi_1\gamma_*\leq10\}}\left(\frac{1}{\gamma_*^2}+\frac{\log k}{\gamma_*}\right)+\mathbf{1}_{\{n\pi_1\gamma_*>10\}}\left(\frac{1}{\gamma_*^2}+\frac{\log k}{\gamma_*}\right)\right\}\lesssim\frac{1}{n\gamma_0^2}+\frac{\log k}{n\gamma_0}.\tag{67}$$

where we got (a) from Markov inequality, (b) from Lemma 15(iii) and (c) using $x+y\leq xy, x, y\geq 2$.

Next we bound $\mathbb{E}\left[\mathbf{1}_{\{A^{>}\}}D\left(M(\cdot|1)\|\widehat{M}^{+1}(\cdot|1)\right)\right]$. Define

$$\Delta_i = M(i|1)\log\left(\frac{M(i|1)}{\widehat{M}^{+1}(i|1)}\right)-M(i|1)+\widehat{M}^{+1}(i|1).$$

As $D(M(\cdot|1)\|\widehat{M}^{+1}(\cdot|1))=\sum_{i=1}^{k}\Delta_i$ it suffices to bound $\mathbb{E}\left[\mathbf{1}_{\{A^{>}\}}\Delta_i\right]$ for each $i$. For some $r\geq 1$ to be optimized later consider the following cases separately

**Case (a)** $n\pi_1\leq r$ **or** $n\pi_1M(i|1)\leq 10$: Using the fact $y\log(y)-y+1\leq(y-1)^2$ with $y=\frac{M(i|1)}{\widehat{M}^{+1}(i|1)}=\frac{M(i|1)(N_1+k)}{N_{1i}+1}$ we get

$$\Delta_i\leq\frac{\left(M(i|1)N_1-N_{1i}+M(i|1)k-1\right)^2}{\left(N_1+k\right)\left(N_{1i}+1\right)}.\tag{68}$$

This implies

$$\mathbb{E}\left[\mathbf{1}_{\{A^{>}\}}\Delta_i\right]\leq\mathbb{E}\left[\frac{\mathbf{1}_{\{A^{>}\}}\left(M(i|1)N_1-N_{1i}+M(i|1)k-1\right)^2}{\left(N_1+k\right)\left(N_{1i}+1\right)}\right]$$

$$\overset{(a)}{\lesssim}\frac{\mathbb{E}\left[\mathbf{1}_{\{A^{>}\}}\left(M(i|1)N_1-N_{1i}\right)^2\right]+k^2\pi_1M(i|1)^2+\pi_1}{n\pi_1+k}$$

$$\overset{(b)}{\lesssim}\frac{\pi_1\mathbb{E}\left[\left(M(i|1)N_1-N_{1i}\right)^2\big|X_n=1\right]}{n\pi_1+k}+\frac{1+rkM(i|1)}{n}\tag{69}$$

where (a) follows from $N_1>\frac{(n-1)\pi_1}{2}$ in $A^{>}$ and the fact that $(x+y+z)^2\leq 3(x^2+y^2+z^2)$; (b) uses the assumption that either $n\pi_1\leq r$ or $n\pi_1M(i|1)\leq 10$. Applying Lemma 15(i) and the fact that $x+x^2\leq 2(1+x^2)$, continuing the last display we get

$$\mathbb{E}\left[\mathbf{1}_{\{A^{>}\}}\Delta_i\right]\lesssim\frac{n\pi_1M(i|1)+\left(1+\frac{M(i|1)}{\gamma_*^2}\right)}{n}+\frac{1+rkM(i|1)}{n}\lesssim\frac{1+rkM(i|1)}{n}+\frac{M(i|1)}{n\gamma_0^2}.$$

Hence

$$\mathbb{E}\left[\mathbf{1}_{\{A^{>}\}}D(M(\cdot|1)\|\widehat{M}^{+1}(\cdot|1))\right]=\sum_{i=1}^{k}\mathbb{E}\left[\mathbf{1}_{\{A^{>}\}}\Delta_i\right]\lesssim\frac{rk}{n}+\frac{1}{\gamma_0^2}.\tag{70}$$

**Case(b)** $n\pi_1 > r$ **and** $n\pi_1 M(i|1) > 10$: We decompose $A^>$ based on count of $N_{1i}$ into atypical part $B^{\leq}$ and typical part $B^>$

$$B^{\leq} \triangleq \{X_n = 1, N_1 > (n-1)\pi_1/2, N_{1i} \leq (n-1)\pi_1 M(i|1)/4\}$$
$$B^> \triangleq \{X_n = 1, N_1 > (n-1)\pi_1/2, N_{1i} > (n-1)\pi_1 M(i|1)/4\}$$

and bound each of $\mathbb{E}\left[\mathbf{1}_{\{B^{\leq}\}}\Delta_i\right]$ and $\mathbb{E}\left[\mathbf{1}_{\{B^>\}}\Delta_i\right]$ separately.

**Bound on** $\mathbb{E}\left[\mathbf{1}_{\{B^{\leq}\}}\Delta_i\right]$    Using $\widehat{M}^{+1}(i|1) \geq \frac{1}{N_1+k}$ and $N_{1i} < N_1 M(i|1)/2$ in $B^{\leq}$ we get

$$
\begin{aligned}
\mathbb{E}\left[\mathbf{1}_{\{B^{\leq}\}}\Delta_i\right] &= \mathbb{E}\left[\mathbf{1}_{\{B^{\leq}\}}M(i|1)\log\left(\frac{M(i|1)(N_1+k)}{N_{1i}+1}\right)\right] + \mathbb{E}\left[\mathbf{1}_{\{B^{\leq}\}}\left(\frac{N_{1i}+1}{N_1+k} - M(i|1)\right)\right] \\
&\leq \mathbb{E}\left[\mathbf{1}_{\{B^{\leq}\}}M(i|1)\log\left(M(i|1)(N_1+k)\right)\right] + \mathbb{E}\left[\mathbf{1}_{\{B^{\leq}\}}\left(\frac{N_{1i}}{N_1} - M(i|1)\right)\right] + \mathbb{E}\left[\frac{\mathbf{1}_{\{B^{\leq}\}}}{N_1}\right] \\
&\lesssim \mathbb{E}\left[\mathbf{1}_{\{B^{\leq}\}}M(i|1)\log\left(M(i|1)(N_1+k)\right)\right] + \frac{1}{n} \qquad (71)
\end{aligned}
$$

where the last inequality followed as $\mathbb{E}\left[\mathbf{1}_{\{B^{\leq}\}}/N_1\right] \lesssim \mathbb{P}[X_n = 1]/n\pi_1 = \frac{1}{n}$. Note that for any event $B$ and any function $g$,

$$\mathbb{E}\left[g(N_1)\mathbf{1}_{\{N_1 \geq t_0, B\}}\right] = g(t_0)\mathbb{P}[N_1 \geq t_0, B] + \sum_{t=t_0+1}^{n}(g(t) - g(t-1))\,\mathbb{P}[N_1 \geq t, B].$$

Applying this identity with $t_0 = \lceil (n-1)\pi_1/2 \rceil$, we can bound the expectation term in (71) as

$$
\begin{aligned}
&\mathbb{E}\left[\mathbf{1}_{\{B^{\leq}\}}M(i|1)\log\left(M(i|1)(N_1+k)\right)\right] \\
&= M(i|1)\log\left(M(i|1)(t_0+k)\right)\mathbb{P}\left[N_1 \geq t_0, N_{1i} \leq \frac{n\pi_1 M(i|1)}{4}, X_n = 1\right] \\
&\quad + M(i|1)\sum_{t=t_0+1}^{n-1}\log\left(1 + \frac{1}{t-1+k}\right)\mathbb{P}\left[N_1 \geq t+1, N_{1i} \leq \frac{n\pi_1 M(i|1)}{4}, X_n = 1\right] \\
&\leq \pi_1 M(i|1)\log\left(M(i|1)(t_0+k)\right)\mathbb{P}\left[M(i|1)N_1 - N_{1i} \geq \frac{M(i|1)t_0}{4}\,\middle|\, X_n = 1\right] \\
&\quad + \frac{M(i|1)}{n}\sum_{t=t_0+1}^{n-1}\mathbb{P}\left[M(i|1)N_1 - N_{1i} \geq \frac{M(i|1)t}{4}\,\middle|\, X_n = 1\right] \qquad (72)
\end{aligned}
$$

where last inequality uses $\log\left(1 + \frac{1}{t-1+k}\right) \leq \frac{1}{t} \lesssim \frac{1}{n\pi_1}$ for all $t \geq t_0$. Using Markov inequality $\mathbb{P}[Z > c] \leq c^{-4}\mathbb{E}\left[Z^4\right]$ for $c > 0$, Lemma 15(ii) and $x + x^4 \leq 2(1 + x^4)$ with $x = \sqrt{M(i|1)}/\gamma_*$

$$\mathbb{P}\left[M(i|1)N_1 - N_{1i} \geq \frac{M(i|1)t}{4}\,\middle|\, X_n = 1\right] \lesssim \frac{(n\pi_1 M(i|1))^2 + \frac{M(i|1)^2}{\gamma_*^4}}{(tM(i|1))^4}.$$

In view of above continuing (72) we get

$$\mathbb{E}\left[\mathbf{1}_{\{B^{\le}\}}M(i|1)\log\left(M(i|1)(N_1+k)\right)\right]$$

$$\lesssim \left((n\pi_1 M(i|1))^2 + \frac{M(i|1)^2}{\gamma_*^4}\right)\left(\frac{\pi_1 M(i|1)\log(M(i|1)(n\pi_1+k))}{(n\pi_1 M(i|1))^4} + \frac{1}{n(M(i|1))^3}\sum_{t=t_0+1}^{n}\frac{1}{t^4}\right)$$

$$\lesssim \left(\frac{(n\pi_1 M(i|1))^2 + \frac{M(i|1)^2}{\gamma_*^4}}{n}\right)\left(\frac{\log(n\pi_1 M(i|1)+kM(i|1))}{(n\pi_1 M(i|1))^3} + \frac{1}{(n\pi_1 M(i|1))^3}\right)$$

$$\lesssim \frac{1}{n}\left((n\pi_1 M(i|1))^2 + \frac{M(i|1)^2}{\gamma_*^4}\right)\frac{\log(n\pi_1 M(i|1)+kM(i|1))}{(n\pi_1 M(i|1))^3}$$

$$\lesssim \frac{1}{n}\left(\frac{\log(n\pi_1 M(i|1)+kM(i|1))}{n\pi_1 M(i|1)} + \frac{M(i|1)\log(n\pi_1 M(i|1)+k)}{n\pi_1\gamma_*^4(n\pi_1 M(i|1))^2}\right)$$

$$\overset{(a)}{\lesssim} \frac{1}{n}\left(\frac{n\pi_1 M(i|1)+kM(i|1)}{n\pi_1 M(i|1)} + \frac{M(i|1)\log(n\pi_1 M(i|1))}{n\pi_1\gamma_*^4(n\pi_1 M(i|1))^2} + \frac{M(i|1)\log k}{n\pi_1\gamma_*^4(n\pi_1 M(i|1))^2}\right)$$

$$\overset{(b)}{\lesssim} \frac{1}{n}\left(1 + kM(i|1) + \frac{M(i|1)\log k}{r\gamma_0^4}\right)$$

where (a) followed using $x+y \le xy$ for $x,y \ge 2$ and (b) followed as $n\pi_1 \ge r, n\pi_1 M(i|1) \ge 10$ and $\log(n\pi_1 M(i|1)) \le n\pi_1 M(i|1)$. In view of (71) this implies

$$\sum_{i=1}^{k}\mathbb{E}\left[\mathbf{1}_{\{B^{\le}\}}\Delta_i\right] \lesssim \sum_{i=1}^{k}\frac{1}{n}\left(1 + kM(i|1)\left(1 + \frac{\log k}{rk\gamma_0^4}\right)\right) \lesssim \frac{k}{n}\left(1 + \frac{\log k}{rk\gamma_0^4}\right). \qquad (73)$$

**Bound on $\mathbb{E}\left[\mathbf{1}_{\{B^{>}\}}\Delta_i\right]$**

Using the inequality (68)

$$\mathbb{E}\left[\mathbf{1}_{\{B^{>}\}}\Delta_i\right] \le \mathbb{E}\left[\frac{\mathbf{1}_{\{B^{>}\}}\left(M(i|1)N_1 - N_{1i} + M(i|1)k - 1\right)^2}{(N_1+k)(N_{1i}+1)}\right]$$

$$\lesssim \frac{\mathbb{E}\left[\mathbf{1}_{\{B^{>}\}}\left\{(M(i|1)N_1 - N_{1i})^2\right\}\right] + k^2\pi_1 M(i|1)^2 + \pi_1}{(n\pi_1+k)(n\pi_1 M(i|1)+1)}$$

$$\lesssim \frac{\pi_1\mathbb{E}\left[(M(i|1)N_1 - N_{1i})^2\,\middle|\,X_n=1\right]}{(n\pi_1+k)(n\pi_1 M(i|1)+1)} + \frac{kM(i|1)}{n}$$

where (a) follows using properties of the set $B^{>}$ along with $(x+y+z)^2 \le 3(x^2+y^2+z^2)$. Using Lemma 15(i) we get

$$\mathbb{E}\left[\mathbf{1}_{\{B^{>}\}}\Delta_i\right] \lesssim \frac{n\pi_1 M(i|1) + \left(1 + \frac{M(i|1)}{\gamma_*^2}\right)}{n(n\pi_1 M(i|1)+1)} + \frac{kM(i|1)}{n} \lesssim \frac{1+kM(i|1)}{n} + \frac{M(i|1)}{n\gamma_0^2}.$$

Summing up the last bound over $i \in [k]$ and using we get for $n\pi_1 > r, n\pi_1 M(i|1) > 10$

$$\mathbb{E}\left[\mathbf{1}_{\{A^{>}\}}D(M(\cdot|1)\|\widehat{M}^{+1}(\cdot|1))\right] = \sum_{i=1}^{k}\left[\mathbb{E}\left[\mathbf{1}_{\{B^{\le}\}}\Delta_i\right] + \mathbb{E}\left[\mathbf{1}_{\{B^{>}\}}\Delta_i\right]\right] \lesssim \frac{k}{n}\left(1 + \frac{1}{k\gamma_0^2} + \frac{\log k}{rk\gamma_0^4}\right).$$

Combining this with (70) we obtain

$$\mathbb{E}\left[\mathbf{1}_{\{A^{>}\}}D(M(\cdot|1)\|\widehat{M}^{+1}(\cdot|1))\right] \lesssim \frac{k}{n}\left(\frac{1}{k\gamma_0^2} + r + \frac{\log k}{rk\gamma_0^4}\right) \lesssim \frac{k}{n}\left(1 + \sqrt{\frac{\log k}{k\gamma_0^4}}\right)$$

where we chose $r = 10 + \sqrt{\frac{\log k}{k\gamma_0^4}}$ for the last inequality. In view of (67) this implies the required bound.

**Remark 4.** We explain the subtlety of the concentration bound in Lemma 15 based on fourth moment and why existing Chernoff bound or Chebyshev inequality falls short. For example, the risk bound in (67) relies on bounding the probability that $N_1$ is atypically small. To this end, one may use the classical Chernoff-type inequality for reversible chains (see [Lez98, Theorem 1.1] or [Pau15, Proposition 3.10 and Theorem 3.3])

$$\mathbb{P}\left[N_1 \leq (n-1)\pi_1/2 | X_1 = 1\right] \lesssim \frac{1}{\sqrt{\pi_1}} e^{-\Theta(n\pi_1\gamma_*)}; \tag{74}$$

in contrast, the fourth moment bound in (66) yields $\mathbb{P}\left[N_1 \leq (n-1)\pi_1/2 | X_1 = 1\right] = O(\frac{1}{(n\pi_1\gamma_*)^2})$. Although the exponential tail in (74) is much better, the pre-factor $\frac{1}{\sqrt{\pi_1}}$, due to conditioning on the initial state, can lead to a suboptimal result when $\pi_1$ is small. (As a concrete example, consider two states with $M(2|1) = \Theta(\frac{1}{n})$ and $M(1|2) = \Theta(1)$. Then $\pi_1 = \Theta(\frac{1}{n}), \gamma = \gamma_* \approx \Theta(1)$, and (74) leads to $\mathbb{P}\left[N_1 \leq (n-1)\pi_1/2, X_n = 1\right] = O(\frac{1}{\sqrt{n}})$ as opposed to the desired $O(\frac{1}{n})$.)

In the same context it is also insufficient to use 2nd moment based bound (Chebyshev), which leads to $\mathbb{P}\left[N_1 \leq (n-1)\pi_1/2 | X_1 = 1\right] = O(\frac{1}{n\pi_1\gamma_*})$. This bound is too loose, which, upon substitution into (66), results in an extra $\log n$ factor in the final risk bound when $\pi_1$ and $\gamma_*$ are large.

### 7.2.2 Proof of Theorem 3 (ii)

Let $k \geq (\log n)^6$ and $\gamma_0 \geq \frac{(\log(n+k))^2}{k}$. We prove a stronger result using spectral gap as opposed to the absolute spectral gap. Fix $M$ such that $\gamma \geq \gamma_0$. Denote its stationary distribution by $\pi$. For absolute constants $\tau > 0$ to be chosen later and $c_0$ as in Lemma 16 below, define

$$\epsilon(m) = \frac{2k}{m} + \frac{c_0(\log n)^3\sqrt{k}}{m}, \quad c_n = 100\tau^2 \frac{\log n}{n\gamma},$$

$$n_i^{\pm} = n\pi_i \pm \tau \max\left\{\frac{\log n}{n\gamma}, \sqrt{\frac{\pi_i \log n}{n\gamma}}\right\}, \quad i = 1, \ldots, k. \tag{75}$$

Let $N_i$ be the number of visits to state $i$ as in (4). We bound the risk by accounting for the contributions from different ranges of $N_i$ and $\pi_i$ separately:

$$\mathbb{E}\left[\sum_{i=1}^{k} \mathbf{1}_{\{X_n=i\}} D\left(M(\cdot|i)\|\widehat{M}^{+1}(\cdot|i)\right)\right]$$

$$= \sum_{i:\pi_i \geq c_n} \mathbb{E}\left[\mathbf{1}_{\{X_n=i, n_i^- \leq N_i \leq n_i^+\}} D\left(M(\cdot|i)\|\widehat{M}^{+1}(\cdot|i)\right)\right]$$

$$+ \sum_{i:\pi_i \geq c_n} \mathbb{E}\left[\mathbf{1}_{\{X_n=i, N_i > n_i^+ \text{ or } N_i < n_i^-\}} D\left(M(\cdot|i)\|\widehat{M}^{+1}(\cdot|i)\right)\right] + \sum_{i:\pi_i < c_n} \mathbb{E}\left[\mathbf{1}_{\{X_n=i\}} D\left(M(\cdot|i)\|\widehat{M}^{+1}(\cdot|i)\right)\right]$$

$$\leq \log(n+k) \sum_{i:\pi_i \geq c_n} \mathbb{P}\left[D(M(\cdot|i)\|\widehat{M}^{+1}(\cdot|i)) > \epsilon(N_i), n_i^- \leq N_i \leq n_i^+\right] + \sum_{i:\pi_i \geq c_n} \mathbb{E}\left[\mathbf{1}_{\{X_n=i, n_i^- \leq N_i \leq n_i^+\}} \epsilon(N_i)\right]$$

$$+ \log(n+k) \sum_{i:\pi_i \geq c_n} \left[\mathbb{P}\left[N_i \geq n_i^+\right] + \mathbb{P}\left[N_i \leq n_i^-\right]\right] + \sum_{i:\pi_i \leq c_n} \pi_i \log(n+k)$$

$$\lesssim \log(n+k) \sum_{i:\pi_i \geq c_n} \mathbb{P}\left[D(M(\cdot|i)\|\widehat{M}^{+1}(\cdot|i)) > \epsilon(N_i), n_i^- \leq N_i \leq n_i^+\right] + \sum_{i:\pi_i \geq c_n} \pi_i \max_{n_i^- \leq m \leq n_i^+} \epsilon(m)$$

$$+ \log(n+k) \sum_{i:\pi_i \geq c_n} \left(\mathbb{P}\left[N_i > n_i^+\right] + \mathbb{P}\left[N_i < n_i^-\right]\right) + \frac{k(\log(n+k))^2}{n\gamma}. \tag{76}$$

where the first inequality uses the worst-case bound (65) for add-one estimator. We analyze the terms separately as follows.

For the second term, given any $i$ such that $\pi_i \geq c_n$, we have, by definition in (75), $n_i^- \geq 9n\pi_i/10$ and $n_i^+ - n_i^- \leq n\pi_i/5$, which implies

$$\sum_{i:\pi_i \geq c_n} \pi_i \max_{n_i^- \leq m \leq n_i^+} \epsilon(m) \leq \sum_{i:\pi_i \geq c_n} \pi_i \left( \frac{2k}{0.9n\pi_i} + \frac{10}{9} \frac{c_0(\log n)^3\sqrt{k}}{n\pi_i} \right) \lesssim \frac{k^2}{n} + \frac{(\log n)^3 k^{3/2}}{n}. \tag{77}$$

For the third term, applying [HJL$^+$18, Lemma 16] (which, in turn, is based on ther Bernstein inequality in [Pau15]), we get $\mathbb{P}\left[N_i > n_i^+\right] + \mathbb{P}\left[N_i < n_i^-\right] \leq 2n^{\frac{-\tau^2}{4+10\tau}}$.

To bound the first term in (76), we follow the method in [Bil61, HJL$^+$18] of representing the sample path of the Markov chain using independent samples generated from $M(\cdot|i)$ which we describe below. Consider a random variable $X_1 \sim \pi$ and an array $W = \{W_{i\ell} : i = 1, \ldots, k \text{ and } \ell = 1, 2, \ldots\}$ of independent random variables, such that $X$ and $W$ are independent and $W_{i\ell} \overset{\text{i.i.d.}}{\sim} M(\cdot|i)$ for each $i$. Starting with generating $X_1$ from $\pi$, at every step $i \geq 2$ we set $X_i$ as the first element in the $X_{i-1}$-th row of $W$ that has not been sampled yet. Then one can verify that $\{X_1, \ldots, X_n\}$ is a Markov chain with initial distribution $\pi$ and transition matrix $M$. Furthermore, the transition counts satisfy $N_{ij} = \sum_{\ell=1}^{N_i} \mathbf{1}_{\{W_{i\ell}=j\}}$, where $N_i$ be the number of elements sampled from the $i$th row of $W$. Note the conditioned on $N_i = m$, the random variables $\{W_{i1}, \ldots, W_{im}\}$ are no longer iid. Instead, we apply a union bound. Note that for each fixed $m$, the estimator

$$\widehat{M}^{+1}(j|i) = \frac{\sum_{\ell=1}^m \mathbf{1}_{\{W_{i\ell}=j\}} + 1}{m+k} \triangleq \widehat{M}_m^{+1}(j|i), \quad j \in [k]$$

is an add-one estimator for $M(j|i)$ based on an i.i.d. sample of size $m$. Lemma 16 below provides a high-probability bound for the add-one estimator in this iid setting. Using this result and the union bound, we have

$$\sum_{i:\pi_i \geq c_n} \mathbb{P}\left[ D(M(\cdot|i)\|\widehat{M}^{+1}(\cdot|i)) > \epsilon(N_i), n_i^- \leq N_i \leq n_i^+ \right]$$

$$\leq \sum_{i:\pi_i \geq c_n} \left( n_i^+ - n_i^- \right) \max_{n_i^- \leq m \leq n_i^+} \mathbb{P}\left[ D(M(\cdot|i)\|\widehat{M}_m^{+1}(\cdot|i)) > \epsilon(m) \right] \leq \sum_{i:\pi_i \geq c_n} \frac{1}{n^2} \leq \frac{k}{n^2}$$

where the second inequality applies Lemma 16 with $t = n \geq n_i^+ \geq m$ and uses $n_i^+ - n_i^- \leq n\pi_i/5$ for $\pi_i \geq c_n$.

Combining the above with (77), we continue (76) with $\tau = 25$ to get

$$\mathbb{E}\left[ \sum_{i=1}^k \mathbf{1}_{\{X_n=i\}} D\left( M(\cdot|i)\|\widehat{M}^{+1}(\cdot|i) \right) \right] \lesssim \frac{k^2}{n} + \frac{(\log n)^3 k^{3/2}}{n} + \frac{k(\log(n+k))^2}{n\gamma}$$

which is $O\left(\frac{k^2}{n}\right)$ whenever $k \geq (\log n)^6$ and $\gamma \geq \frac{(\log(n+k))^2}{k}$.

**Lemma 16** (KL risk bound for add-one estimator). *Let* $V_1, \ldots, V_m \overset{iid}{\sim} Q$ *for some distribution* $Q = \{Q_i\}_{i=1}^k$ *on* $[k]$. *Consider the add-one estimator* $\widehat{Q}^{+1}$ *with* $\widehat{Q}_i^{+1} = \frac{1}{m+k}(\sum_{j=1}^m \mathbf{1}_{\{V_j=i\}} + 1)$. *There exists an absolute constant* $c_0$ *such that for any* $t \geq m$,

$$\mathbb{P}\left[ D(Q\|\widehat{Q}^{+1}) \geq \frac{2k}{m} + \frac{c_0(\log t)^3\sqrt{k}}{m} \right] \leq \frac{1}{t^3}.$$

*Proof.* Let $\widehat{Q}$ be the empirical estimator $\widehat{Q}_i = \frac{1}{m} \sum_{j=1}^{m} \mathbf{1}_{\{V_j=i\}}$. Then $\widehat{Q}_i^{+1} = \frac{m\widehat{Q}_i+1}{m+k}$ and hence

$$
\begin{aligned}
D(Q\|\widehat{Q}^{+1}) &= \sum_{i=1}^{k} \left( Q_i \log \frac{Q_i}{\widehat{Q}_i^{+1}} - Q_i + \widehat{Q}_i^{+1} \right) \\
&= \sum_{i=1}^{k} \left( Q_i \log \frac{Q_i(m+k)}{m\widehat{Q}_i+1} - Q_i + \frac{m\widehat{Q}_i+1}{m+k} \right) \\
&= \sum_{i=1}^{k} \left( Q_i \log \frac{Q_i}{\widehat{Q}_i + \frac{1}{m}} - Q_i + \widehat{Q}_i + \frac{1}{m} \right) + \sum_{i=1}^{k} \left( Q_i \log \frac{m+k}{m} - \frac{k\widehat{Q}_i}{m+k} - \frac{k}{m(m+k)} \right) \\
&\leq \sum_{i=1}^{k} \left( Q_i \log \frac{Q_i}{\widehat{Q}_i + \frac{1}{m}} - Q_i + \widehat{Q}_i + \frac{1}{m} \right) + \frac{k}{m} \qquad (78)
\end{aligned}
$$

with last equality following by $0 \leq \log\left(\frac{m+k}{m}\right) \leq k/m$.

To control the sum in the above display it suffices to consider its Poissonized version. Specifically, we aim to show

$$
\mathbb{P}\left[ \sum_{i=1}^{k} \left( Q_i \log \frac{Q_i}{\widehat{Q}_i^{\mathsf{poi}} + \frac{1}{m}} - Q_i + \widehat{Q}_i^{\mathsf{poi}} + \frac{1}{m} \right) > \frac{k}{m} + \frac{c_0(\log t)^3\sqrt{k}}{m} \right] \leq \frac{1}{t^4} \qquad (79)
$$

where $m\widehat{Q}_i^{\mathsf{poi}}$, $i = 1,\ldots,k$ are distributed independently as $\mathrm{Poi}(mQ_i)$. (Here and below $\mathrm{Poi}(\lambda)$ denotes the Poisson distribution with mean $\lambda$.) To see why (79) implies the desired result, letting $w = \frac{k}{m} + \frac{c_0(\log t)^3\sqrt{k}}{m}$ and $Y = \sum_{i=1}^{k} m\widehat{Q}_i^{\mathsf{poi}} \sim \mathrm{Poi}(m)$, we have

$$
\begin{aligned}
&\mathbb{P}\left[ \sum_{i=1}^{k} \left( Q_i \log \frac{Q_i}{\widehat{Q}_i + \frac{1}{m}} - Q_i + \widehat{Q}_i + \frac{1}{m} \right) > w \right] \\
&\overset{(a)}{=} \mathbb{P}\left[ \sum_{i=1}^{k} \left( Q_i \log \frac{Q_i}{\widehat{Q}_i^{\mathsf{poi}} + \frac{1}{m}} - Q_i + \widehat{Q}_i^{\mathsf{poi}} + \frac{1}{m} \right) > w \,\bigg|\, \sum_{i=1}^{k} Q_i^{\mathsf{poi}} = 1 \right] \\
&\overset{(b)}{\leq} \frac{1}{t^4 \mathbb{P}[Y=m]} = \frac{m!}{t^4 e^{-m} m^m} \overset{(c)}{\lesssim} \frac{\sqrt{m}}{t^4} \leq \frac{1}{t^3}. \qquad (80)
\end{aligned}
$$

where (a) followed from the fact that conditioned on their sum independent Poisson random variables follow a multinomial distribution; (b) applies (79); (c) follows from Stirling's approximation.

To prove (79) we rely on concentration inequalities for sub-exponential distributions. A random variable $X$ is called sub-exponential with parameters $\sigma^2, b > 0$, denoted as $\mathsf{SE}(\sigma^2, b)$ if

$$
\mathbb{E}\left[ e^{\lambda(X - \mathbb{E}[X])} \right] \leq e^{\frac{\lambda^2 \sigma^2}{2}}, \quad \forall |\lambda| < \frac{1}{b}. \qquad (81)
$$

Sub-exponential random variables satisfy the following properties [Wai19, Sec. 2.1.3]:

- If $X$ is $\mathsf{SE}(\sigma^2, b)$ for any $t > 0$

$$
\mathbb{P}\left[ |X - \mathbb{E}[X]| \geq v \right] \leq \begin{cases} 2e^{-v^2/(2\sigma^2)}, & 0 < v \leq \frac{\sigma^2}{b} \\ 2e^{-v/(2b)}, & v > \frac{\sigma^2}{b}. \end{cases} \qquad (82)
$$

- Bernstein condition: A random variable $X$ is $\mathsf{SE}(\sigma^2, b)$ if it satisfies

$$
\mathbb{E}\left[ |X - \mathbb{E}[X]|^\ell \right] \leq \frac{1}{2} \ell! \sigma^2 b^{\ell-2}, \quad \ell = 2, 3, \ldots. \qquad (83)
$$

- If $X_1, \ldots, X_k$ are independent $\mathsf{SE}(\sigma^2, b)$, then $\sum_{i=1}^{k} X_i$ is $\mathsf{SE}(k\sigma^2, b)$.

Define $X_i = Q_i \log \frac{Q_i}{\widehat{Q}_i^{\text{poi}} + \frac{1}{m}} - Q_i + \widehat{Q}_i^{\text{poi}} + \frac{1}{m}, i \in [k]$. Then Lemma 17 below shows that $X_i$'s are independent $\mathsf{SE}(\sigma^2, b)$ with $\sigma^2 = \frac{c_1 (\log m)^4}{m^2}, b = \frac{c_2 (\log m)^2}{n}$ for absolute constants $c_1, c_2$, and hence $\sum_{i=1}^{k} (X_i - \mathbb{E}[X_i])$ is $\mathsf{SE}(k\sigma^2, b)$. In view of (82) for the choice $c_0 = 8(c_1 + c_2)$ this implies

$$\mathbb{P}\left[ \sum_{i=1}^{k} (X_i - \mathbb{E}[X_i]) \geq c_0 \frac{(\log t)^3 \sqrt{k}}{m} \right] \leq 2e^{-\frac{c_0^2 k (\log t)^6}{2m^2 \sigma^2}} + 2e^{-\frac{c_0 \sqrt{k} (\log t)^3}{2mb}} \leq \frac{1}{t^3}. \qquad (84)$$

Using $0 \leq y \log y - y + 1 \leq (y-1)^2, y > 0$ and $\mathbb{E}\left[ \frac{\lambda}{\text{Poi}(\lambda)+1} \right] = \sum_{v=0}^{\infty} \frac{e^{-\lambda} \lambda^{v+1}}{(v+1)!} = 1 - e^{-\lambda}$

$$\mathbb{E}\left[ \sum_{i=1}^{k} X_i \right] \leq \mathbb{E}\left[ \sum_{i=1}^{k} \frac{\left( Q_i - \left( \widehat{Q}_i^{\text{poi}} + \frac{1}{m} \right) \right)^2}{\widehat{Q}_i^{\text{poi}} + \frac{1}{m}} \right]$$

$$= \sum_{i=1}^{k} m Q_i^2 \mathbb{E}\left[ \frac{1}{m\widehat{Q}_i^{\text{poi}} + 1} \right] - 1 + \frac{k}{m} = \sum_{i=1}^{k} Q_i \left( 1 - e^{-mQ_i} \right) - 1 + \frac{k}{m} \leq \frac{k}{m}.$$

Combining the above with (84) we get (79) as required. $\qquad \square$

**Lemma 17.** *There exist absolute constants $c_1, c_2$ such that the following holds. For any $p \in (0,1)$ and $nY \sim \text{Poi}(np)$, $X = p \log \frac{p}{Y + \frac{1}{n}} - p + Y + \frac{1}{n}$ is $\mathsf{SE}\left( \frac{c_1 (\log n)^4}{n^2}, \frac{c_2 (\log n)^2}{n} \right)$.*

*Proof.* Note that $X$ is a non-negative random variable. Since $\mathbb{E}\left[ (X - \mathbb{E}[X])^{\ell} \right] \leq 2^{\ell} \mathbb{E}\left[ X^{\ell} \right]$, by the Bernstein condition (83), it suffices to show $\mathbb{E}[X^{\ell}] \leq \left( \frac{c_3 \ell (\log n)^2}{n} \right)^{\ell}, \ell = 2, 3, \dots$ for some absolute constant $c_3$. guarantees the desired sub-exponential behavior. The analysis is divided into following two cases for some absolute constant $c_4 \geq 24$.

**Case I** $p \geq \frac{c_4 \ell \log n}{n}$**:** Using Chernoff bound for Poisson [Jan02, Theorem 3]

$$\mathbb{P}\left[ |\text{Poi}(\lambda) - \lambda| > x \right] \leq 2e^{-\frac{x^2}{2(\lambda + x/3)}}, \quad \lambda, x > 0, \qquad (85)$$

we get

$$\mathbb{P}\left[ |Y - p| > \sqrt{\frac{c_4 \ell p \log n}{4n}} \right] \leq 2 \exp\left( -\frac{c_4 n \ell p \log n}{8np + 2\sqrt{c_4 n \ell p \log n}} \right)$$

$$\leq 2 \exp\left( -\frac{c_4 \ell \log n}{8 + 2\sqrt{c_4 \ell \log n / np}} \right) \leq \frac{1}{n^{2\ell}} \qquad (86)$$

which implies $p/2 \leq Y \leq 2p$ with probability at least $1 - n^{-2\ell}$. Since $0 \leq X \leq \frac{(Y - p - \frac{1}{n})^2}{Y + \frac{1}{n}}$, we get

$$\mathbb{E}[X^{\ell}] \lesssim \frac{\left( \sqrt{c_4 \ell p \log n / 4n} \right)^{2\ell}}{(p/2)^{\ell}} + \frac{n^{\ell}}{n^{2\ell}} \lesssim \left( \frac{c_4 \ell \log n}{n} \right)^{\ell}.$$

**Case II** $p < \frac{c_4 \ell \log n}{n}$**:**

- On the event $\{Y > p\}$, we have $X \leq Y + \frac{1}{n} \leq 2Y$, where the last inequality follows because $nY$ takes non-negative integer values. Since $X \geq 0$, we have $X^{\ell} \mathbf{1}_{\{Y > p\}} \leq (2Y)^{\ell} \mathbf{1}_{\{Y > p\}}$ for any $\ell \geq 2$. Using the Chernoff bound (85), we get $Y \leq \frac{2c_4 \ell \log n}{n}$ with probability at least $1 - n^{-2\ell}$, which implies

$$\mathbb{E}\left[ X^{\ell} \mathbf{1}_{\{Y \geq p\}} \right] \leq \mathbb{E}\left[ (2Y)^{\ell} \mathbf{1}_{\{Y > p, Y \leq \frac{2c_4 \ell \log n}{n}\}} \right] + \mathbb{E}\left[ (2Y)^{\ell} \mathbf{1}_{\{Y > p, Y > \frac{2c_4 \ell \log n}{n}\}} \right]$$

$$\leq \left( \frac{4c_4 \ell \log n}{n} \right)^{\ell} + 2^{\ell} \left( \mathbb{E}[Y^{2\ell}] \mathbb{P}\left[ Y > \frac{2c_4 \ell \log n}{n} \right] \right)^{\frac{1}{2}} \leq \left( \frac{c_5 \ell \log n}{n} \right)^{\ell}$$

for absolute constant $c_5$. Here, the last inequality follows from Cauchy-Schwarz and using the Poisson moment bound [Ahl21, Theorem 2.1]:[4] $\mathbb{E}[(nY)^{2\ell}] \leq \left( \frac{2\ell}{\log\left(1+\frac{2\ell}{np}\right)} \right)^{2\ell} \leq$ $(c_6 \ell \log n)^{2\ell}$ for some absolute constant $c_6$, with the second inequality applying the assumption $p < \frac{c_4 \ell \log n}{n}$.

- As $X\mathbf{1}_{\{Y \leq p\}} \leq p \log n + \frac{1}{n} \lesssim \frac{\ell(\log n)^2}{n}$, we get $\mathbb{E}\left[ X^\ell \mathbf{1}_{\{Y \leq p\}} \right] \leq \left( \frac{c_7 \ell (\log n)^2}{n} \right)^\ell$ for some absolute constant $c_7$.

$\square$

### 7.2.3 Proof of Corollary 4

We show the following monotonicity result of the prediction risk. In view of this result, Corollary 4 immediately follows from Theorem 2 and Theorem 3 (i).

**Lemma 18.** $\mathsf{Risk}_{k+1,n}(\gamma_0) \geq \mathsf{Risk}_{k,n}(\gamma_0)$ for all $\gamma_0 \in (0,1), k \geq 2$.

*Proof.* Fix an $M \in \mathcal{M}_k(\gamma_0)$ such that $\gamma_*(M) > \gamma_0$. Denote the stationary distribution $\pi$ such that $\pi M = \pi$. Fix $\delta \in (0,1)$ and define a transition matrix $\widetilde{M}$ with $k+1$ states as follows:

$$\widetilde{M} = \begin{pmatrix} (1-\delta)M & \delta\mathbf{1} \\ (1-\delta)\pi & \delta \end{pmatrix}$$

One can verify the following:

- $\widetilde{M}$ is irreducible and reversible;

- The stationary distribution for $\widetilde{M}$ is $\widetilde{\pi} = ((1-\delta)\pi, \delta)$

- The absolute spectral gap of $\widetilde{M}$ is $\gamma_*(\widetilde{M}) = (1-\delta)\gamma_*(M)$, so that $\widetilde{M} \in \mathcal{M}_{k+1}(\gamma_0)$ for all sufficiently small $\delta$.

- Let $(X_1, \ldots, X_n)$ and $(\widetilde{X}_1, \ldots, \widetilde{X}_n)$ be stationary Markov chains with transition matrices $M$ and $\widetilde{M}$, respectively. Then as $\delta \to 0$, $(X_1, \ldots, X_n)$ converges to $(\widetilde{X}_1, \ldots, \widetilde{X}_n)$ in law, i.e., the joint probability mass function converges pointwise.

Next fix any estimator $\widehat{M}$ for state space $[k+1]$. Note that without loss of generality we can assume $\widehat{M}(j|i) > 0$ for all $i, j \in [k+1]$ for otherwise the KL risk is infinite. Define $\widehat{M}^{\mathrm{trunc}}$ as $\widehat{M}$ without the $k+1$-th row and column, and denote by $\widehat{M}'$ its normalized version, namely, $\widehat{M}'(\cdot|i) = \frac{\widehat{M}^{\mathrm{trunc}}(\cdot|i)}{1-\widehat{M}^{\mathrm{trunc}}(k+1|i)}$ for $i = 1, \ldots, k$. Then

$$
\begin{aligned}
\mathbb{E}_{\widetilde{X}^n}\left[ D(\widetilde{M}(\cdot|\widetilde{X}_n)\|\widehat{M}(\cdot|\widetilde{X}_n)) \right] \xrightarrow{\delta \to 0} & \mathbb{E}_{X^n}\left[ D(M(\cdot|X_n)\|\widehat{M}(\cdot|X_n)) \right] \\
\geq & \mathbb{E}_{X^n}\left[ D(M(\cdot|X_n)\|\widehat{M}'(\cdot|X_n)) \right] \\
\geq & \inf_{\widehat{M}} \mathbb{E}_{X^n}\left[ D(M(\cdot|X_n)\|\widehat{M}(\cdot|X_n)) \right]
\end{aligned}
$$

where in the first step we applied the convergence in law of $\widetilde{X}^n$ to $X^n$ and the continuity of $P \mapsto D(P\|Q)$ for fixed componentwise positive $Q$; in the second step we used the fact that for any sub-probability measure $Q = (q_i)$ and its normalized version $\bar{Q} = Q/\alpha$ with $\alpha = \sum q_i \leq 1$, we have $D(P\|Q) = D(P\|\bar{Q}) + \log\frac{1}{\alpha} \geq D(P\|\bar{Q})$. Taking the supremum over $M \in \mathcal{M}_k(\gamma_0)$ on the LHS and the supremum over $\widetilde{M} \in \mathcal{M}_{k+1}(\gamma_0)$ on the RHS, and finally the infimum over $\widehat{M}$ on the LHS, we conclude $\mathsf{Risk}_{k+1,n}(\gamma_0) \geq \mathsf{Risk}_{k,n}(\gamma_0)$. $\square$

---

[4] For a result with less precise constants, see also [Ahl21, Eq. (1)] based on [Lat97, Corollary 1].

# A  Mutual information representation of prediction risk

The following lemma justifies the representation (22) for the prediction risk as maximal conditional mutual information. Unlike (17) for redundancy which holds essentially without any condition [Kem74], here we impose certain compactness assumptions which hold finite alphabets such as finite-state Markov chains studied in this paper.

**Lemma 19.** *Let $\mathcal{X}$ be finite and let $\Theta$ be a compact subset of $\mathbb{R}^d$. Given $\{P_{X^{n+1}|\theta} : \theta \in \Theta\}$, define the prediction risk*

$$\mathsf{Risk}_n \triangleq \inf_{Q_{X_{n+1}|X^n}} \sup_{\theta \in \Theta} D(P_{X_{n+1}|X^n,\theta}\|Q_{X_{n+1}|X^n}|P_{X^n|\theta}), \tag{87}$$

*Then*

$$\mathsf{Risk}_n = \sup_{P_\theta \in \mathcal{M}(\Theta)} I(\theta; X_{n+1}|X^n). \tag{88}$$

*where $\mathcal{M}(\Theta)$ denotes the collection of all (Borel) probability measures on $\Theta$.*

Note that for stationary Markov chains, (22) follows from Lemma 19 since one can take $\theta$ to be the joint distribution of $(X_1, \ldots, X_{n+1})$ itself which forms a compact subset of the probability simplex on $\mathcal{X}^{n+1}$.

*Proof.* It is clear that (87) is equivalent to

$$\mathsf{Risk}_n = \inf_{Q_{X_{n+1}|X^n}} \sup_{P_\theta \in \mathcal{M}(\Theta)} D(P_{X_{n+1}|X^n,\theta}\|Q_{X_{n+1}|X^n}|P_{X^n,\theta}).$$

By the variational representation (14) of conditional mutual information, we have

$$I(\theta; X_{n+1}|X^n) = \inf_{Q_{X_{n+1}|X^n}} D(P_{X_{n+1}|X^n,\theta}\|Q_{X_{n+1}|X^n}|P_{X^n,\theta}). \tag{89}$$

Thus (88) amounts to justifying the interchange of infimum and supremum in (87). It suffices to prove the upper bound.

Let $|\mathcal{X}| = K$. For $\epsilon \in (0, 1)$, define an auxiliary quantity:

$$\mathsf{Risk}_{n,\epsilon} \triangleq \inf_{Q_{X_{n+1}|X^n} \geq \frac{\epsilon}{K}} \sup_{P_\theta \in \mathcal{M}(\Theta)} D(P_{X_{n+1}|X^n,\theta}\|Q_{X_{n+1}|X^n}|P_{X^n,\theta}), \tag{90}$$

where the constraint in the infimum is pointwise, namely, $Q_{X_{n+1}=x_{n+1}|X^n=x^n} \geq \frac{\epsilon}{K}$ for all $x_1, \ldots, x_{n+1} \in \mathcal{X}$. By definition, we have $\mathsf{Risk}_n \leq \mathsf{Risk}_{n,\epsilon}$. Furthermore, $\mathsf{Risk}_{n,\epsilon}$ can be equivalently written as

$$\mathsf{Risk}_{n,\epsilon} = \inf_{Q_{X_{n+1}|X^n}} \sup_{P_\theta \in \mathcal{M}(\Theta)} D(P_{X_{n+1}|X^n,\theta}\|(1-\epsilon)Q_{X_{n+1}|X^n} + \epsilon U|P_{X^n,\theta}), \tag{91}$$

where $U$ denotes the uniform distribution on $\mathcal{X}$.

We first show that the infimum and supremum in (91) can be interchanged. This follows from the standard minimax theorem. Indeed, note that $D(P_{X_{n+1}|X^n,\theta}\|(1-\epsilon)Q_{X_{n+1}|X^n} + \epsilon U|P_{X^n,\theta})$ is convex in $Q_{X_{n+1}|X^n}$, affine in $P_\theta$, continuous in each argument, and takes values in $[0, \log \frac{K}{\epsilon}]$. Since $\mathcal{M}(\Theta)$ is convex and weakly compact (by Prokhorov's theorem) and the collection of conditional distributions $Q_{X_{n+1}|X^n}$ is convex, the minimax theorem (see, e.g., [Fan53, Theorem 2]) yields

$$\mathsf{Risk}_{n,\epsilon} = \sup_{\pi \in \mathcal{M}(\Theta)} \inf_{Q_{X_{n+1}|X^n}} D(P_{X_{n+1}|X^n,\theta}\|(1-\epsilon)Q_{X_{n+1}|X^n} + \epsilon U|P_{X^n,\theta}). \tag{92}$$

Finally, by the convexity of the KL divergence, for any $P$ on $\mathcal{X}$, we have

$$D(P\|(1-\epsilon)Q + \epsilon U) \leq (1-\epsilon)D(P\|Q) + \epsilon D(P\|U) \leq (1-\epsilon)D(P\|Q) + \epsilon \log K,$$

which, in view of (89) and (92), implies

$$\mathsf{Risk}_n \leq \mathsf{Risk}_{n,\epsilon} \leq \sup_{P_\theta \in \mathcal{M}(\Theta)} I(\theta; X_{n+1}|X^n) + \epsilon \log K.$$

By the arbitrariness of $\epsilon$, (88) follows.  $\square$

# B Proof of Lemma 15

Recall that for any irreducible and reversible finite states transition matrix $M$ with stationary distribution $\pi$ the followings are satisfied:

1. $\pi_i > 0$ for all $i$.
2. $M(j|i)\pi_i = M(i|j)\pi_j$ for all $i, j$.

The following is a direct consequence of the Markov property.

**Lemma 20.** *For any $1 \leq t_1 < \cdots < t_m < \cdots < t_k$ and any $Z_2 = f\left(X_{t_k}, \ldots, X_{t_m}\right), Z_1 = g\left(X_{t_{m-1}}, \ldots, X_{t_1}\right)$ we have*

$$\mathbb{E}\left[Z_2 \mathbf{1}_{\{X_{t_m}=j\}} Z_1 | X_1 = i\right] = \mathbb{E}\left[Z_2 | X_{t_m} = j\right] \mathbb{E}\left[\mathbf{1}_{\{X_{t_m}=j\}} Z_1 | X_1 = i\right] \tag{93}$$

For $t \geq 0$, denote the $t$-step transition probability by $\mathbb{P}\left[X_{t+1} = j | X_1 = i\right] = M^t(j|i)$, which is the $ij$th entry of $M^t$. The following result is standard (see, e.g., [LP17, Chap. 12]). We include the proof mainly for the purpose of introducing the spectral decomposition.

**Lemma 21.** *Define $\lambda_* \triangleq 1 - \gamma_* = \max\{|\lambda_i| : i \neq 1\}$. For any $t \geq 0$, $|M^t(j|i) - \pi_j| \leq \lambda_*^t \sqrt{\frac{\pi_j}{\pi_i}}$.*

*Proof.* Throughout the proof all vectors are column vectors except for $\pi$. Let $D_\pi$ denote the diagonal matrix with entries $D_\pi(i, i) = \pi_i$. By reversibility, $D_\pi^{\frac{1}{2}} M D_\pi^{-\frac{1}{2}}$, which shares the same spectrum with $M$, is a symmetric matrix and admits the spectral decomposition $D_\pi^{\frac{1}{2}} M D_\pi^{-\frac{1}{2}} = \sum_{a=1}^k \lambda_a u_a u_a^\top$ for some orthonormal basis $\{u_1, \ldots, u_k\}$; in particular, $\lambda_1 = 1$ and $u_{1i} = \sqrt{\pi_i}$. Then for each $t \geq 1$,

$$M^t = \sum_{a=1}^k \lambda_a^t D_\pi^{-\frac{1}{2}} u_a u_a^\top D_\pi^{\frac{1}{2}} = \mathbf{1}\pi + \sum_{a=2}^k \lambda_a^t D_\pi^{-\frac{1}{2}} u_a u_a^\top D_\pi^{\frac{1}{2}}. \tag{94}$$

where $\mathbf{1}$ is the all-ones vector. As $u_a$'s satisfy $\sum_{a=1}^k u_a u_a^\top = I$ we get $\sum_{a=2}^k u_{ab}^2 = 1 - u_{a1}^2 \leq 1$ for any $b = 1, \ldots, k$. Using this along with Cauchy-Schwarz inequality we get

$$\left|M^t(j|i) - \pi_j\right| \leq \sqrt{\frac{\pi_j}{\pi_i}} \sum_{a=2}^k |\lambda_a|^t |u_{ai} u_{aj}| \leq \lambda_*^t \sqrt{\frac{\pi_j}{\pi_i}} \left(\sum_{a=2}^k u_{ai}^2\right)^{\frac{1}{2}} \left(\sum_{a=2}^k u_{aj}^2\right)^{\frac{1}{2}} \leq \lambda_*^t \sqrt{\frac{\pi_j}{\pi_i}}$$

as required. $\square$

**Lemma 22.** *Fix states $i, j$. For any integers $a \geq b \geq 1$, define*

$$h_s(a, b) = \left|\mathbb{E}\left[\mathbf{1}_{\{X_{a+1}=i\}} \left(\mathbf{1}_{\{X_a=j\}} - M(j|i)\right)^s | X_b = i\right]\right|, \quad s = 1, 2, 3, 4.$$

*Then*

*(i)* $h_1(a, b) \leq 2\sqrt{M(j|i)}\lambda_*^{a-b}$

*(ii)* $|h_2(a, b) - \pi_i M(j|i)(1 - M(j|i))| \leq 4\sqrt{M(j|i)}\lambda_*^{a-b}$.

*(iii)* $h_3(a, b), h_4(a, b) \leq \pi_i M(j|i)(1 - M(j|i)) + 4\sqrt{M(j|i)}\lambda_*^{a-b}$.

*Proof.* We apply Lemma 21 and time reversibility:

(i)

$$\begin{aligned}
h_1(a, b) &= |\mathbb{P}\left[X_{a+1} = i, X_a = j | X_b = i\right] - M(j|i)\mathbb{P}\left[X_{a+1} = i | X_b = i\right]| \\
&= \left|M(i|j)M^{a-b}(j|i) - M(j|i)M^{a-b+1}(i|i)\right| \\
&\leq M(i|j)\left|M^{a-b}(j|i) - \pi_j\right| + M(j|i)\left|M^{a-b+1}(i|i) - \pi_i\right| \\
&\leq \lambda_*^{a-b} M(i|j)\sqrt{\frac{\pi_j}{\pi_i}} + M(j|i)\lambda_*^{a-b+1} \\
&= \lambda_*^{a-b}\sqrt{M(j|i)M(i|j)} + M(j|i)\lambda_*^{a-b+1} \leq 2\sqrt{M(j|i)}\lambda_*^{a-b}.
\end{aligned}$$

(ii)

$$|h_2(a,b) - \pi_i M(j|i)(1 - M(j|i))|$$

$$= \Big| \mathbb{E}\left[\mathbf{1}_{\{X_{a+1}=i, X_a=j\}}|X_b=i\right] - \pi_i M(j|i) + (M(j|i))^2 \left(\mathbb{E}\left[\mathbf{1}_{\{X_{a+1}=i\}}|X_b=i\right] - \pi_i\right)$$

$$\qquad - 2M(j|i)(\mathbb{E}\left[\mathbf{1}_{\{X_{a+1}=i, X_a=j\}}|X_b=i\right] - \pi_i M(j|i)) \Big|$$

$$\leq |\mathbb{P}\left[X_{a+1}=i, X_a=j|X_b=i\right] - \pi_j M(i|j)| + (M(j|i))^2 |\mathbb{P}\left[X_{a+1}=i|X_b=i\right] - \pi_i|$$

$$\qquad + 2M(j|i)|\mathbb{P}\left[X_{a+1}=i, X_a=j|X_b=i\right] - \pi_j M(i|j)|$$

$$= M(i|j)\left|M^{a-b}(j|i) - \pi_j\right| + (M(j|i))^2 \left|M^{a-b+1}(i|i) - \pi_i\right| + 2M(j|i)M(i|j)\left|M^{a-b}(j|i) - \pi_j\right|$$

$$\leq M(i|j)\sqrt{\frac{\pi_j}{\pi_i}}\lambda_*^{a-b} + (M(j|i))^2\lambda_*^{a-b+1} + 2M(j|i)M(i|j)\sqrt{\frac{\pi_j}{\pi_i}}\lambda_*^{a-b}$$

$$\leq \lambda_*^{a-b}\left(\sqrt{M(i|j)}\sqrt{\frac{M(i|j)\pi_j}{\pi_i}} + (M(j|i))^2 + 2M(j|i)\sqrt{M(i|j)}\sqrt{\frac{M(i|j)\pi_j}{\pi_i}}\right)$$

$$\leq 4\sqrt{M(j|i)}\lambda_*^{a-b}.$$

(iii) $h_3(a,b), h_4(a,b) \leq h_2(a,b)$. $\qquad\qquad\qquad\qquad\qquad\qquad\square$

*Proof of Lemma 15(i).* For ease of notation we use $c_0$ to denote an absolute constant whose value may vary at each occurrence. Fix $i, j \in [k]$. Note that the empirical count defined in (4) can be written as $N_i = \sum_{a=1}^{n-1} \mathbf{1}_{\{X_{n-a}=i\}}$ and $N_{ij} = \sum_{a=1}^{n-1} \mathbf{1}_{\{X_{n-a}=i, X_{n-a+1}=j\}}$. Then

$$\mathbb{E}\left[(M(j|i)N_i - N_{ij})^2 |X_n=i\right]$$

$$= \mathbb{E}\left[\left(\sum_{a=1}^{n-1} \mathbf{1}_{\{X_{n-a}=i\}}\left(\mathbf{1}_{\{X_{n-a+1}=j\}} - M(j|i)\right)\right)^2 \Bigg| X_n=i\right]$$

$$\overset{(a)}{=} \mathbb{E}\left[\left(\sum_{a=1}^{n-1} \mathbf{1}_{\{X_{a+1}=i\}}\left(\mathbf{1}_{\{X_a=j\}} - M(j|i)\right)\right)^2 \Bigg| X_1=i\right]$$

$$\overset{(b)}{=} \left|\sum_{a,b} \mathbb{E}\left[\eta_a\eta_b|X_1=i\right]\right| \leq 2\sum_{a\geq b} |\mathbb{E}\left[\eta_a\eta_b|X_1=i\right]|,$$

where (a) is due to time reversibility; in (b) we defined $\eta_a \triangleq \mathbf{1}_{\{X_{a+1}=i\}}\left(\mathbf{1}_{\{X_a=j\}} - M(j|i)\right)$. We divide the summands into different cases and apply Lemma 22.

**Case I: Two distinct indices.** For any $a > b$, using Lemma 20 we get

$$|\mathbb{E}\left[\eta_a\eta_b|X_1=i\right]| = |\mathbb{E}\left[\eta_a|X_{b+1}=i\right]| \, |\mathbb{E}\left[\eta_b|X_1=1\right]| = h_1(a, b+1)h_1(b, 1) \qquad (95)$$

which implies

$$\sum_{n-1\geq a>b\geq 1}\sum |\mathbb{E}\left[\eta_a\eta_b|X_1=i\right]| = \sum_{n-1\geq a>b\geq 1}\sum h_1(a, b+1)h_1(b, 1) \lesssim M(j|i) \sum_{n-1\geq a>b\geq 1}\sum \lambda_*^{a-2} \lesssim \frac{M(j|i)}{\gamma_*^2}.$$

Here the last inequality (and similar sums in later deductions) can be explained as follows. Note that for $\gamma_* \geq \frac{1}{2}$ (i.e. $\lambda_* \leq \frac{1}{2}$), the sum is clearly bounded by an absolute constant; for $\gamma_* < \frac{1}{2}$ (i.e. $\lambda_* > \frac{1}{2}$), we compare the sum with the mean (or higher moments in other calculations) of a geometric random variable.

**Case II: Single index.**

$$\sum_{a=1}^{n-1} \mathbb{E}\left[\eta_a^2|X_1=i\right] = \sum_{a=1}^{n-1} h_2(a, 1) \lesssim n\pi_i M(j|i)(1 - M(j|i)) + \frac{\sqrt{M(j|i)}}{\gamma_*}. \qquad (96)$$

Combining the above we get

$$\mathbb{E}\left[(N_{ij} - M(j|i)N_i)^2 | X_n = i\right] \lesssim n\pi_i M(j|i)(1 - M(j|i)) + \frac{\sqrt{M(j|i)}}{\gamma_*} + \frac{M(j|i)}{\gamma_*^2}$$

as required. □

*Proof of Lemma 15(ii).* We first note that due to reversibility we can write (similar as in proof of Lemma 15(i)) with $\eta_a = \mathbf{1}_{\{X_{a+1}=i\}}\left(\mathbf{1}_{\{X_a=j\}} - M(j|i)\right)$

$$\mathbb{E}\left[(M(j|i)N_i - N_{ij})^4 | X_n = i\right]$$

$$= \mathbb{E}\left[\left(\sum_{a=1}^{n-1} \mathbf{1}_{\{X_{a+1}=i\}}\left(\mathbf{1}_{\{X_a=j\}} - M(j|i)\right)\right)^4 \Bigg| X_1 = i\right]$$

$$= \left|\sum_{a,b,d,e} \mathbb{E}\left[\eta_a \eta_b \eta_d \eta_e | X_1 = i\right]\right| \le \sum_{a,b,d,e} |\mathbb{E}\left[\eta_a \eta_b \eta_d \eta_e | X_1 = i\right]| \lesssim \sum_{a \ge b \ge d \ge e} |\mathbb{E}\left[\eta_a \eta_b \eta_d \eta_e | X_1 = i\right]|.$$

(97)

We bound the sum over different combinations of $a \ge b \ge d \ge e$ to come up with a bound on the required fourth moment. We first divide the $\eta$'s into groups depending on how many distinct indices of $\eta$ there are. We use the following identities which follow from Lemma 20: for indices $a > b > d > e$

- $|\mathbb{E}\left[\eta_a \eta_b \eta_d \eta_e | X_1 = i\right]| = h_1(a, b+1)h_1(b, d+1)h_1(d, e+1)h_1(e, 1)$

- For $s_1, s_2, s_3 \in \{1, 2\}$, $|\mathbb{E}\left[\eta_a^{s_1} \eta_b^{s_2} \eta_d^{s_3} | X_1 = i\right]| = h_{s_1}(a, b+1)h_{s_2}(b, d+1)h_{s_3}(d, 1)$

- For $s_1, s_2 \in \{1, 2, 3\}$, $|\mathbb{E}\left[\eta_a^{s_1} \eta_b^{s_2} | X_1 = i\right]| = h_{s_1}(a, b+1)h_{s_2}(b, 1)$

- $\mathbb{E}\left[\eta_a^4 | X_1 = 1\right] = h_4(a, 1)$

and then use Lemma 22 to bound the $h$ functions.

**Case I: Four distinct indices.** Using Lemma 22 we have

$$\sum_{n-1 \ge a > b > d > e \ge 1} |\mathbb{E}\left[\eta_a \eta_b \eta_d \eta_e | X_1 = i\right]| = \sum_{n-1 \ge a > b > d > e \ge 1} h_1(a, b+1)h_1(b, d+1)h_1(d, e+1)h_1(e, 1)$$

$$\le M(j|i)^2 \sum_{n-1 \ge a > b > d > e \ge 1} \lambda_*^{a-4} \lesssim \frac{M(j|i)^2}{\gamma_*^4}.$$

**Case II: Three distinct indices.** There are three cases, namely $\eta_a^2 \eta_b \eta_d$, $\eta_a \eta_b^2 \eta_d$ and $\eta_a \eta_b \eta_d^2$.

1. Bounding $\sum\sum\sum_{n-1 \ge a > b > d \ge 1} |\mathbb{E}\left[\eta_a^2 \eta_b \eta_d | X_1 = i\right]|$:

$$\sum_{n-1 \ge a > b > d \ge 1} |\mathbb{E}\left[\eta_a^2 \eta_b \eta_d | X_1 = i\right]| = \sum_{n-1 \ge a > b > d \ge 1} h_2(a, b+1)h_1(b, d+1)h_1(d, 1)$$

$$\lesssim \sum_{n-1 \ge a > b > d \ge 1} \left(\pi_i M(j|i)(1 - M(j|i)) + \sqrt{M(j|i)}\lambda_*^{a-b-1}\right) M(j|i)\lambda_*^{b-2}$$

$$\lesssim \frac{M(j|i)}{\gamma_*^2} n\pi_i M(j|i)(1 - M(j|i)) + \frac{M(j|i)^{\frac{3}{2}}}{\gamma_*^3}$$

$$\lesssim (n\pi_i M(j|i)(1 - M(j|i)))^2 + \frac{M(j|i)^{\frac{3}{2}}}{\gamma_*^3} + \frac{M(j|i)^2}{\gamma_*^4}$$

where the last inequality followed by using $xy \le x^2 + y^2$.

2. Bounding $\sum\sum\sum_{n-2\geq a>b>d\geq 1}\left|\mathbb{E}\left[\eta_a\eta_b^2\eta_d|X_1=i\right]\right|$:

$$\sum\sum\sum_{n-2\geq a>b>d\geq 1}\left|\mathbb{E}\left[\eta_a\eta_b^2\eta_d|X_1=i\right]\right|$$

$$=\sum\sum\sum_{n-2\geq a>b>d\geq 1}h_1(a,b+1)h_2(b,d+1)h_1(d,1)$$

$$\lesssim\sum\sum\sum_{n-2\geq a>b>d\geq 1}\left(\pi_iM(j|i)(1-M(j|i))+\sqrt{M(j|i)}\lambda_*^{b-d-1}\right)M(j|i)\lambda_*^{a-b+d-2}$$

$$\lesssim\frac{M(j|i)}{\gamma_*^2}n\pi_iM(j|i)(1-M(j|i))+\frac{M(j|i)^{\frac{3}{2}}}{\gamma_*^3}$$

$$\lesssim n\pi_iM(j|i)(1-M(j|i))^2+\frac{M(j|i)^{\frac{3}{2}}}{\gamma_*^3}+\frac{M(j|i)^2}{\gamma_*^4}.$$

3. Bounding $\sum\sum\sum_{n-2\geq a>b>d\geq 1}\left|\mathbb{E}\left[\eta_a\eta_b\eta_d^2|X_1=i\right]\right|$:

$$\sum\sum\sum_{n-2\geq a>b>d\geq 1}\left|\mathbb{E}\left[\eta_a\eta_b\eta_d^2|X_1=i\right]\right|$$

$$=\sum\sum\sum_{n-2\geq a>b>d\geq 1}h_1(a,b+1)h_1(b,d+1)h_2(d,1)$$

$$\lesssim\sum\sum\sum_{n-2\geq a>b>d\geq 1}\left(\pi_iM(j|i)(1-M(j|i))+\sqrt{M(j|i)}\lambda_*^{d-1}\right)M(j|i)\lambda_*^{a-d-2}$$

$$\lesssim\frac{M(j|i)}{\gamma_*^2}n\pi_iM(j|i)(1-M(j|i))+\frac{M(j|i)^{\frac{3}{2}}}{\gamma_*^3}$$

$$\lesssim(n\pi_iM(j|i)(1-M(j|i)))^2+\frac{M(j|i)^{\frac{3}{2}}}{\gamma_*^3}+\frac{M(j|i)^2}{\gamma_*^4}.$$

**Case III: Two distinct indices.** There are three different cases, namely $\eta_a^2\eta_b^2$, $\eta_a^3\eta_b$ and $\eta_a\eta_b^3$.

1. Bounding $\sum\sum_{n-2\geq a>b\geq 1}\left|\mathbb{E}\left[\eta_a^2\eta_b^2|X_1=i\right]\right|$:

$$\sum\sum_{n-2\geq a>b\geq 1}\mathbb{E}\left[\eta_a^2\eta_b^2|X_1=i\right]$$

$$=\sum\sum_{n-2\geq a>b\geq 1}h_2(a,b+1)h_2(b,1)$$

$$\lesssim\sum\sum_{n-2\geq a>b\geq 1}\left(\pi_iM(j|i)(1-M(j|i))+\sqrt{M(j|i)}\lambda_*^{a-b-1}\right)\left(\pi_iM(j|i)(1-M(j|i))+\sqrt{M(j|i)}\lambda_*^{b-1}\right)$$

$$\lesssim\sum\sum_{n-2\geq a>b\geq 1}\left\{\pi_iM(j|i)(1-M(j|i))\sqrt{M(j|i)}(\lambda_*^{a-b-1}+\lambda_*^{b-1})\right.$$

$$\left.+(\pi_iM(j|i)(1-M(j|i)))^2+M(j|i)\lambda_*^{a-2}\right\}$$

$$\lesssim(n\pi_iM(j|i)(1-M(j|i)))^2+\frac{\sqrt{M(j|i)}}{\gamma_*}n\pi_iM(j|i)(1-M(j|i))+\frac{M(j|i)}{\gamma_*^2}$$

$$\lesssim(n\pi_iM(j|i)(1-M(j|i)))^2+\frac{M(j|i)}{\gamma_*^2}.$$

2. Bounding $\sum\sum_{n-2\geq a>b\geq 1}\left|\mathbb{E}\left[\eta_a^3\eta_b|X_1=i\right]\right|$:

$$\sum_{n-2\geq a>b\geq 1}\sum \left|\mathbb{E}\left[\eta_a^3\eta_b|X_1=i\right]\right|$$

$$=\sum_{n-2\geq a>b\geq 1}\sum h_3(a,b+1)h_1(b,1)$$

$$\lesssim \sum_{n-2\geq a>b\geq 1}\sum \left(\pi_i M(j|i)(1-M(j|i))+\sqrt{M(j|i)}\lambda_*^{a-b-1}\right)\sqrt{M(j|i)}\lambda_*^{b-1}$$

$$\lesssim \frac{\sqrt{M(j|i)}}{\gamma_*}n\pi_i M(j|i)(1-M(j|i))+\frac{M(j|i)}{\gamma_*^2}\lesssim (n\pi_i M(j|i)(1-M(j|i)))^2+\frac{M(j|i)}{\gamma_*^2}.$$

3. Bounding $\sum\sum_{n-2\geq a>b\geq 1}\left|\mathbb{E}\left[\eta_a\eta_b^3|X_1=i\right]\right|$:

$$\sum_{n-2\geq a>b\geq 1}\sum \left|\mathbb{E}\left[\eta_a\eta_b^3|X_1=i\right]\right|$$

$$=\sum_{n-2\geq a>b\geq 1}\sum h_1(a,b+1)h_3(b,1)$$

$$\lesssim \sum_{n-2\geq a>b\geq 1}\sum \left(\pi_i M(j|i)(1-M(j|i))+\sqrt{M(j|i)}\lambda_*^{b-1}\right)\sqrt{M(j|i)}\lambda_*^{a-b-1}$$

$$\lesssim \frac{\sqrt{M(j|i)}}{\gamma_*}n\pi_i M(j|i)(1-M(j|i))+\frac{M(j|i)}{\gamma_*^2}\lesssim (n\pi_i M(j|i)(1-M(j|i)))^2+\frac{M(j|i)}{\gamma_*^2}.$$

**Case IV: Single index.** Bound on $\sum_{a=1}^{n-1}\mathbb{E}\left[\eta_a^4|X_1=i\right]$:

$$\sum_{a=1}^{n-1}\mathbb{E}\left[\eta_a^4|X_1=i\right]=\sum_{a=1}^{n-1}h_4(a,1)\leq n\pi_i M(j|i)(1-M(j|i))+\frac{\sqrt{M(j|i)}}{\gamma_*}.$$

Combining all cases we get

$$\mathbb{E}\left[(M(j|i)N_i-N_{ij})^4|X_n=i\right]\lesssim (n\pi_i M(j|i)(1-M(j|i)))^2+\frac{\sqrt{M(j|i)}}{\gamma_*}+\frac{M(j|i)}{\gamma_*^2}+\frac{M(j|i)^{\frac{3}{2}}}{\gamma_*^3}+\frac{M(j|i)^2}{\gamma_*^4}$$

$$\lesssim (n\pi_i M(j|i)(1-M(j|i)))^2+\frac{\sqrt{M(j|i)}}{\gamma_*}+\frac{M(j|i)^2}{\gamma_*^4}$$

as required. □

*Proof of Lemma 15(iii).* Throughout our proof we repeatedly use the spectral decomposition (94) applied to the diagonal elements:

$$M^t(i|i)=\pi_i+\sum_{v\geq 2}\lambda_v^t u_{vi}^2,\quad \sum_{v\geq 2}u_{vi}^2\leq 1.$$

Write $N_i-(n-1)\pi_i=\sum_{a=1}^{n-1}\xi_a$ where $\xi_a=\mathbf{1}_{\{X_a=i\}}-\pi_i$. For $a\geq b\geq d\geq e$,

$$\mathbb{E}\left[\xi_a\xi_b\xi_d\xi_e|X_1=i\right]$$

$$=\mathbb{E}\left[\xi_a\xi_b\left(\mathbf{1}_{\{X_d=i,X_e=i\}}-\pi_i\mathbf{1}_{\{X_d=i\}}-\pi_i\mathbf{1}_{\{X_e=i\}}+\pi_i^2\right)|X_1=i\right]$$

$$=\mathbb{E}\left[\xi_a\xi_b\mathbf{1}_{\{X_d=i,X_e=i\}}|X_1=i\right]-\pi_i\mathbb{E}\left[\xi_a\xi_b\mathbf{1}_{\{X_d=i\}}|X_1=i\right]$$

$$\quad-\pi_i\mathbb{E}\left[\xi_a\xi_b\mathbf{1}_{\{X_e=i\}}|X_1=i\right]+\pi_i^2\mathbb{E}\left[\xi_a\xi_b|X_1=i\right]$$

$$=\mathbb{E}\left[\xi_a\xi_b|X_d=i\right]\mathbb{P}\left[X_d=i|X_e=i\right]\mathbb{P}\left[X_e=i|X_1=i\right]-\pi_i\mathbb{E}\left[\xi_a\xi_b|X_d=i\right]\mathbb{P}[X_d=i|X_1=i]$$

$$\quad-\pi_i\mathbb{E}\left[\xi_a\xi_b|X_e=i\right]\mathbb{P}[X_e=i|X_1=i]+\pi_i^2\mathbb{E}\left[\xi_a\xi_b|X_1=i\right]$$

$$=\mathbb{E}\left[\xi_a\xi_b|X_d=i\right]\left\{M^{d-e}(i|i)M^{e-1}(i|i)-\pi_i M^{d-1}(i|i)\right\}$$

$$\quad-\left\{\pi_i\mathbb{E}\left[\xi_a\xi_b|X_e=i\right]M^{e-1}(i|i)-\pi_i^2\mathbb{E}\left[\xi_a\xi_b|X_1=i\right]\right\} \tag{98}$$

Using the Markov property for any $d \leq b \leq a$, we get

$$\left| \mathbb{E}[\xi_a \xi_b | X_d = i] - \pi_i \sum_{v \geq 2} u_{vi}^2 \lambda_v^{a-b} \right|$$

$$= \left| \mathbb{E}\left[ \mathbf{1}_{\{X_a = i, X_b = i\}} - \pi_i \mathbf{1}_{\{X_a = i\}} - \pi_i \mathbf{1}_{\{X_b = i\}} + \pi_i^2 | X_d = i \right] - \pi_i \sum_{v \geq 2} u_{vi}^2 \lambda_v^{a-b} \right|$$

$$= \left| M^{a-b}(i|i) M^{b-d}(i|i) - \pi_i M^{a-d}(i|i) - \pi_i M^{b-d}(i|i) + \pi_i^2 - \pi_i \sum_{v \geq 2} u_{vi}^2 \lambda_v^{a-b} \right|$$

$$= \left| \left( \pi_i + \sum_{v \geq 2} u_{vi}^2 \lambda_v^{a-b} \right) \left( \pi_i + \sum_{v \geq 2} u_{vi}^2 \lambda_v^{b-d} \right) - \pi_i \left( \pi_i + \sum_{v \geq 2} u_{vi}^2 \lambda_v^{a-d} \right) \right.$$

$$\left. - \pi_i \left( \pi_i + \sum_{v \geq 2} u_{vi}^2 \lambda_v^{b-d} \right) + \pi_i^2 - \pi_i \sum_{v \geq 2} u_{vi}^2 \lambda_v^{a-b} \right|$$

$$= \left| \left( \sum_{v \geq 2} u_{vi}^2 \lambda_v^{a-b} \right) \left( \sum_{v \geq 2} u_{vi}^2 \lambda_v^{b-d} \right) - \pi_i \sum_{v \geq 2} u_{vi}^2 \lambda_v^{a-d} \right|$$

$$\leq \lambda_*^{a-d} \left( \sum_{v \geq 2} u_{vi}^2 \right) \left( \sum_{v \geq 2} u_{vi}^2 \right) + \lambda_*^{a-d} \pi_i \sum_{v \geq 2} u_{vi}^2 \leq 2 \lambda_*^{a-d}. \tag{99}$$

We also get for $d \geq e$

$$\left| M^{d-e}(i|i) M^{e-1}(i|i) - \pi_i M^{d-1}(i|i) \right|$$

$$= \left| \left( \pi_i + \sum_{v \geq 2} u_{vi}^2 \lambda_v^{d-e} \right) \left( \pi_i + \sum_{v \geq 2} u_{vi}^2 \lambda_v^{e-1} \right) - \pi_i \left( \pi_i + \sum_{v \geq 2} u_{vi}^2 \lambda_v^{d-1} \right) \right|$$

$$= \left| \pi_i \sum_{v \geq 2} u_{vi}^2 \lambda_v^{e-1} + \pi_i \sum_{v \geq 2} u_{vi}^2 \lambda_v^{d-e} + \left( \sum_{v \geq 2} u_{vi}^2 \lambda_v^{e-1} \right) \left( \sum_{v \geq 2} u_{vi}^2 \lambda_v^{d-e} \right) - \pi_i \sum_{v \geq 2} u_{vi}^2 \lambda_v^{d-1} \right|$$

$$\leq 2 \lambda_*^{d-1} + \pi_i \lambda_*^{e-1} + \pi_i \lambda_*^{d-e}. \tag{100}$$

This implies

$$\left| \mathbb{E}\left[ \xi_a \xi_b | X_d = i \right] \right| \left| M^{d-e}(i|i) M^{e-1}(i|i) - \pi_i M^{d-1}(i|i) \right|$$

$$\leq \left( \pi_i \sum_{v \geq 2} u_{vi}^2 \lambda_v^{a-b} + 2 \lambda_*^{a-d} \right) \left( 2 \lambda_*^{d-1} + \pi_i \lambda_*^{e-1} + \pi_i \lambda_*^{d-e} \right)$$

$$\leq \left( \pi_i \lambda_*^{a-b} + 2 \lambda_*^{a-d} \right) \left( 2 \lambda_*^{d-1} + \pi_i \lambda_*^{e-1} + \pi_i \lambda_*^{d-e} \right)$$

$$\leq 4 \left[ \pi_i^2 \lambda_*^{a-b+d-e} + \pi_i^2 \lambda_*^{a-b+e-1} + \pi_i \left( \lambda_*^{a-b+d-1} + \lambda_*^{a-d+e-1} + \lambda_*^{a-e} \right) + \lambda_*^{a-1} \right] \tag{101}$$

Using (99) along with Lemma 21 for any $e \le b \le a$ we get

$$\left| \pi_i \mathbb{E}\left[\xi_a \xi_b | X_e = i\right] M^{e-1}(i|i) - \pi_i^2 \mathbb{E}\left[\xi_a \xi_b | X_1 = i\right] \right|$$

$$\le \pi_i \left| \mathbb{E}\left[\xi_a \xi_b | X_e = i\right] \right| \left| M^{e-1}(i|i) - \pi_i \right| + \pi_i^2 \left| \mathbb{E}\left[\xi_a \xi_b | X_e = i\right] - \pi_i \sum_{v \ge 2} u_{vi}^2 \lambda_v^{a-b} \right| \tag{102}$$

$$+ \pi_i^2 \left| \mathbb{E}\left[\xi_a \xi_b | X_1 = i\right] - \pi_i \sum_{v \ge 2} u_{vi}^2 \lambda_v^{a-b} \right|$$

$$\le \pi_i \left[ \pi_i \sum_{v \ge 2} u_{vi}^2 \lambda_v^{a-b} + 2\lambda_*^{a-e} \right] 2\lambda_*^{e-1} + 2\pi_i^2 \lambda_*^{a-e} + 2\pi_i^2 \lambda_*^{a-1}$$

$$\le 2\pi_i^2 \lambda_*^{a-b+e-1} + 4\pi_i^2 \lambda_*^{a-e} + 4\pi_i^2 \lambda_*^{a-1}. \tag{103}$$

This together with (101) and (98) implies

$$\left| \mathbb{E}\left[\xi_a \xi_b \xi_d \xi_e | X_1 = i\right] \right| \lesssim \pi_i^2 \left(\lambda_*^{a-b+d-e} + \lambda_*^{a-b+e-1}\right) + \lambda_*^{a-1}$$
$$+ \pi_i \left(\lambda_*^{a-b+d-1} + \lambda_*^{a-d+e-1} + \lambda_*^{a-e}\right) \tag{104}$$

To bound the sum over $n - 1 \ge a \ge b \ge d \ge e \ge 1$, we divide the analysis according to the number of distinct ordered indices related variations in terms.

**Case I: four distinct indices.** We sum (104) over all possible $a > b > d > e$.

- For the first term,

$$\pi_i^2 \sum_{n-1 \ge a > b > d > e \ge 1} \lambda_*^{a-b+d-e} \lesssim \frac{n\pi_i^2}{\gamma_*} \sum_{n-1 \ge a > b \ge 3} \lambda_*^{a-b} \lesssim \frac{n^2 \pi_i^2}{\gamma_*^2}.$$

- For the second term,

$$\pi_i^2 \sum_{n-1 \ge a > b > d > e \ge 1} \lambda_*^{a-b+e-1} \lesssim \frac{n\pi_i^2}{\gamma_*} \sum_{n-1 \ge a > b \ge 3} \lambda_*^{a-b} \lesssim \frac{n^2 \pi_i^2}{\gamma_*^2}$$

- For the third term,

$$\sum_{n-1 \ge a > b > d > e \ge 1} \lambda_*^{a-1} \lesssim \sum_{n-1 \ge a \ge 4} a^3 \lambda_*^{a-1} \lesssim \frac{1}{\gamma_*^4}.$$

- For the fourth term,

$$\pi_i \sum_{n-1 \ge a > b > d > e \ge 1} \lambda_*^{a-b+d-1} \le \frac{\pi_i}{\gamma_*^2} \sum_{n-1 \ge a > b \ge 3} \lambda_*^{a-b} \lesssim \frac{n\pi_i}{\gamma_*^3}$$

- For the fifth term,

$$\pi_i \sum_{n-1 \ge a > b > d > e \ge 1} \lambda_*^{a-d+e-1} \lesssim \frac{\pi_i}{\gamma_*} \left( \sum_{n-1 \ge a > b \ge 3} \lambda_*^{a-b} \right) \left( \sum_{d \ge 2}^{b-1} \lambda_*^{b-d} \right) \lesssim \frac{n\pi_i}{\gamma_*^3}.$$

- For the sixth term,

$$\pi_i \sum_{n-1 \ge a > b > d > e \ge 1} \lambda_*^{a-e} \lesssim \pi_i \left( \sum_{n-1 \ge a > b \ge 3} \lambda_*^{a-b} \right) \left( \sum_{d \ge 2}^{b-1} \lambda_*^{b-d} \right) \left( \sum_{e \ge 1}^{d-1} \lambda_*^{d-e} \right) \lesssim \frac{n\pi_i}{\gamma_*^3}.$$

Combining the above bounds and using the fact that $ab \le a^2 + b^2$, we obtain

$$\sum_{n-1 \ge a > b > d > e \ge 1} \left| \mathbb{E}\left[\xi_a \xi_b \xi_d \xi_e | X_1 = i\right] \right| \lesssim \frac{n^2 \pi_i^2}{\gamma_*^2} + \frac{n\pi_i}{\gamma_*^3} + \frac{1}{\gamma_*^4} \lesssim \frac{n^2 \pi_i^2}{\gamma_*^2} + \frac{1}{\gamma_*^4}. \tag{105}$$

**Case II: three distinct indices.** There are three cases, namely, $\xi_a\xi_b^2\xi_e$, $\xi_a\xi_b\xi_e^2$, and $\xi_a^2\xi_b\xi_e$.

1. Bounding $\sum\sum\sum_{n-1\geq a>b>e\geq1}\left|\mathbb{E}\left[\xi_a\xi_b^2\xi_e|X_1=i\right]\right|$: We specialize (104) with $b=d$ to get

$$\left|\mathbb{E}\left[\xi_a\xi_b^2\xi_e|X_1=i\right]\right|\lesssim\pi_i\left(\lambda_*^{a-b+e-1}+\lambda_*^{a-e}\right)+\lambda_*^{a-1}.$$

Summing over $a,b,e$ we have

$$\sum_{n-1\geq a>b>e\geq1}\sum\sum\left|\mathbb{E}\left[\xi_a\xi_b^2\xi_e|X_1=i\right]\right|$$
$$\lesssim\sum_{n-1\geq a>b>e\geq1}\sum\sum\left\{\pi_i\left(\lambda_*^{a-b+e-1}+\lambda_*^{a-e}\right)+\lambda_*^{a-1}\right\}$$
$$\lesssim\frac{\pi_i}{\gamma_*}\sum_{n-1\geq a>b\geq2}\sum\lambda_*^{a-b}+\pi_i\left(\sum_{n-1\geq a>b\geq2}\sum\lambda_*^{a-b}\right)\left(\sum_{e\geq1}^{b-1}\lambda_*^{b-e}\right)+\sum_{n-1\geq a\geq3}a^3\lambda_*^{a-1}$$
$$\lesssim\frac{n\pi_i}{\gamma_*^2}+\frac{1}{\gamma_*^3}\lesssim\frac{n^2\pi_i^2}{\gamma_*^2}+\frac{1}{\gamma_*^3} \tag{106}$$

with last inequality following from $xy\leq x^2+y^2$.

2. Bounding $\sum\sum\sum_{n-1\geq a>b>e\geq1}\left|\mathbb{E}\left[\xi_a\xi_b\xi_e^2|X_1=i\right]\right|$: We specialize (104) with $e=d$ to get

$$\left|\mathbb{E}\left[\xi_a\xi_b\xi_e^2|X_1=i\right]\right|\lesssim\pi_i^2\lambda_*^{a-b}+\pi_i\left(\lambda_*^{a-b+e-1}+\lambda_*^{a-e}\right)+\lambda_*^{a-1}.$$

Summing over $a,b,e$ and applying (106), we get

$$\sum_{n-1\geq a>b>e\geq1}\sum\sum\left|\mathbb{E}\left[\xi_a\xi_b\xi_e^2|X_1=i\right]\right|$$
$$\lesssim\sum_{n-1\geq a>b>e\geq1}\sum\sum\left\{\pi_i^2\lambda_*^{a-b}+\pi_i\left(\lambda_*^{a-b+e-1}+\lambda_*^{a-e}\right)+\lambda_*^{a-1}\right\}$$
$$\lesssim n\pi_i^2\sum_{n-1\geq a>b\geq2}\sum\lambda_*^{a-b}+\frac{n\pi_i}{\gamma_*^2}+\frac{1}{\gamma_*^3}\lesssim\frac{n^2\pi_i^2}{\gamma_*}+\frac{n\pi_i}{\gamma_*^2}+\frac{1}{\gamma_*^3}\lesssim\frac{n^2\pi_i^2}{\gamma_*^2}+\frac{1}{\gamma_*^3}. \tag{107}$$

3. Bounding $\sum\sum\sum_{n-1\geq a>b>e\geq1}\left|\mathbb{E}\left[\xi_a^2\xi_b\xi_e|X_1=i\right]\right|$: Specializing (104) with $a=b$ we get

$$\left|\mathbb{E}\left[\xi_b^2\xi_d\xi_e|X_1=i\right]\right|\lesssim\pi_i^2\left(\lambda_*^{d-e}+\lambda_*^{e-1}\right)+\lambda_*^{b-1}+\pi_i\left(\lambda_*^{d-1}+\lambda_*^{b-d+e-1}+\lambda_*^{b-e}\right),$$

which is equivalent to

$$\left|\mathbb{E}\left[\xi_a^2\xi_b\xi_e|X_1=i\right]\right|\lesssim\pi_i^2\left(\lambda_*^{b-e}+\lambda_*^{e-1}\right)+\lambda_*^{a-1}+\pi_i\left(\lambda_*^{b-1}+\lambda_*^{a-b+e-1}+\lambda_*^{a-e}\right).$$

For the first, second and fourth terms

$$\sum_{n-1\geq a>b>e\geq1}\sum\sum\left\{\pi_i^2\left(\lambda_*^{b-e}+\lambda_*^{e-1}\right)+\pi_i\lambda_*^{b-1}\right\}\lesssim\frac{\pi_i^2}{\gamma_*}\sum_{n-1\geq a>b\geq2}\sum1+\frac{n\pi_i}{\gamma_*^2}\lesssim\frac{n^2\pi_i^2}{\gamma_*}+\frac{n\pi_i}{\gamma_*^2},$$

and for summing the remaining terms we use (106), which implies

$$\sum_{n-1\geq a>b>e\geq1}\sum\sum\left|\mathbb{E}\left[\xi_a^2\xi_b\xi_e|X_1=i\right]\right|\lesssim\frac{n^2\pi_i^2}{\gamma_*}+\frac{n\pi_i}{\gamma_*^2}+\frac{1}{\gamma_*^3}\lesssim\frac{n^2\pi_i^2}{\gamma_*^2}+\frac{1}{\gamma_*^3}. \tag{108}$$

**Case III: two distinct indices.** There are three cases, namely, $\eta_a^2\eta_e^2$, $\eta_a\eta_e^3$ and $\eta_a^3\eta_e$.

1. Bounding $\sum\sum_{n-1\geq a>e\geq1}\mathbb{E}\left[\xi_a^2\xi_e^2|X_1=i\right]$: Specializing (104) for $a=b$ and $e=d$ we get

$$\mathbb{E}\left[\xi_a^2\xi_e^2|X_1=i\right]\lesssim\pi_i^2+\pi_i\left(\lambda_*^{e-1}+\lambda_*^{a-e}\right)+\lambda_*^{a-1}.$$

Summing up over $a, e$ we have

$$\sum_{n-1\geq a>e\geq 1}\sum \mathbb{E}\left[\xi_a^2\xi_e^2|X_1=i\right] \lesssim \sum_{n-1\geq a>e\geq 1}\sum \left\{\pi_i^2+\pi_i\left(\lambda_*^{e-1}+\lambda_*^{a-e}\right)+\lambda_*^{a-1}\right\} \lesssim n^2\pi_i^2+\frac{n\pi_i}{\gamma_*}+\frac{1}{\gamma_*^2}.$$
(109)

2. Bounding $\sum\sum_{n-1\geq a>e\geq 1}\left|\mathbb{E}\left[\xi_a\xi_e^3|X_1=i\right]\right|$: Specializing (104) for $e=b=d$ we get

$$\left|\mathbb{E}\left[\xi_a\xi_e^3|X_1=i\right]\right| \lesssim \pi_i\lambda_*^{a-e}+\lambda_*^{a-1}$$

which sums up to

$$\sum_{n-1\geq a>e\geq 1}\sum \left|\mathbb{E}\left[\xi_a\xi_e^3|X_1=i\right]\right| \lesssim \pi_i\sum_{n-1\geq a>e\geq 1}\sum \lambda_*^{a-e}+\sum_{n-1\geq a>e\geq 1}\sum \lambda_*^{a-1} \lesssim \frac{n\pi_i}{\gamma_*}+\frac{1}{\gamma_*^2}.$$
(110)

3. Bounding $\sum\sum_{n-1\geq a>e\geq 1}\left|\mathbb{E}\left[\xi_a^3\xi_e|X_1=i\right]\right|$: Specializing (104) for $a=b=d$ we get

$$\left|\mathbb{E}\left[\xi_a^3\xi_e|X_1=i\right]\right| \lesssim \pi_i\left(\lambda_*^{a-e}+\lambda_*^{e-1}\right)+\lambda_*^{a-1}$$

which sums up to

$$\sum_{n-1\geq a>e\geq 1}\sum \left|\mathbb{E}\left[\xi_a^3\xi_e|X_1=i\right]\right| \lesssim \sum_{n-1\geq a>e\geq 1}\sum \left\{\pi_i\left(\lambda_*^{a-e}+\lambda_*^{e-1}\right)+\lambda_*^{a-1}\right\} \lesssim \frac{n\pi_i}{\gamma_*}+\frac{1}{\gamma_*^2}.$$
(111)

**Case IV: single distinct index.** We specialize (104) to $a=b=d=e$ to get

$$\mathbb{E}\left[\xi_a^4|X_1=i\right] \lesssim \pi_i+\lambda_*^{a-1}.$$

Summing the above over $a$

$$\sum_{a=1}^{n-1}\mathbb{E}\left[\xi_a^4|X_1=i\right] \lesssim n\pi_i+\frac{1}{\gamma_*}.$$
(112)

Combining (105)–(112) and using $\frac{n\pi_i}{\gamma_*} \lesssim \frac{n^2\pi_i^2}{\gamma_*^2}+\frac{1}{\gamma_*^4}$, we get

$$\mathbb{E}\left[(N_i-(n-1)\pi_i)^4|X_1=i\right] \lesssim \frac{n^2\pi_i^2}{\gamma_*^2}+\frac{1}{\gamma_*^4}.$$

$\square$