# OpenReview forum: "Optimal prediction of Markov chains with and without spectral gap"
_NeurIPS.cc/2021/Conference — NeurIPS 2021 Poster_

### Official Review · Reviewer_CoqU · 2021-07-09

**Rating:** 9
**Confidence:** 3

**Summary:**

Applying techniques from universal compression, the authors characterize the optimal prediction risk in KL-divergence of the following problem: Based on a trajectory of length $n$ drawn from an unknown Markov chain with a state space of size $3\leq k\leq O(\sqrt{n})$, predict the next state.

**Limitations And Societal Impact:**

I don't see any potential negative societal impact.
I suggest formulating the theorems in a more detailed manner that makes the assumptions made explicit.

**Main Review:**

The paper is of high quality and the results are significant and interesting. Nevertheless, while the authors do make efforts to explain their ideas and machinery, I find that the paper is hard to read and that several points are not clear or confusing. For example, even after several reads I am not sure what "with and without spectral gap" means. Does it mean with and without ASSUMPTIONS on the spectral gap? Section 3 is called Optimal rates without spectral gap but there is no corresponding section for "with". Furthermore, Section 3 is concerned merely with the lower bound. Additionally, in Section 2 there are results that make assumptions on the spectral gap and results that do not.

Adding to the confusion, the connection between subsequent paragraphs is sometimes not clear and sometimes explanations for results come AFTER they appear. This makes reading difficult and hard to understand what exactly wad done by the authors and what is already known.

I would also make the following specific suggestions. Whenever my remark is formulated as a question I mean that I find that an explanation IN THE TEXT is necessary.

1. [line 25] While it can be assumed that if $Y\sim P$ then $p_{i}=P(Y=i)$ and similarly for $Q$, this should be made clear.
2. [lines 36-38] I am not sure about the correctness of the claim that “if certain state has very small probability under the stationary distribution, consistent estimation of the transition matrix may not be possible”. Under irreducibility, every state is visited infinitely many times and consistent estimation is guaranteed. Under reducibility, some states may not be visited at all and the corresponding transition probabilities cannot be estimated.
3. [lines 50-51] Since highly related to this work, the authors should clarify what exactly “implicit assumptions on mixing time such as spectral gap conditions” that were made by HOP18 meant. (or at least point to where they are stated in HOP18, since I have not found them there).
4. [line 53] The authors should make the phrase “with and without spectral gap” more precise. Isn't “without assumptions on the spectral gap“ or “arbitrary spectral gap” clearer?
5. [line 87] It would be informative to explain the sudden restriction to reversible Markov chains. Is it an artifact of the analysis? Do reversibilization methods fail? What about the pseudo spectral gap?
6. [lines 92-93] From $\gamma_{0}>0$ it follows that the Markov chain is aperiodic. Together with irreducibility this give ergodicity. The authors might want to use this term explicitly.
7. [line 96] In what sense is Theorem 2 an extension of (2) of [FOPS16]?
6. [lines 124 and 131] “known as redundancy” and “is standard”. The authors might want to give references for the definition and for the result.
8. [line 142] To which reduction do the authors refer?
9. [line 213] Is there a reason for $X^{n+1}$ instead of $X^{n}$? What is $\Theta$?
10. [line 214] Shouldn't the definition $X^{n}\overset{\Delta}{=}(X_{1},\ldots,X_{n})$ come before line 213? Actually, $X^{n}$ was already defined in line 23.
11. [line 220] Here a more exact reference is given ([CT06, Chapter 13]). It would be nice if the authors could provide exact references for the well known definitions and results that are being used.

**Time Spent Reviewing:**

20

---

> ### Author Response · Authors · 2021-08-10
> **Responses to Reviewer 4**
>
> We thank the reviewer for the constructive and positive comments. We will try to implement the comments about wordings that have been pointed out. Specific responses to comments are given below.
>
> 1. “… For example, even after several reads I am not sure what "with and without spectral gap" means. Does it mean with and without ASSUMPTIONS on the spectral gap?”
>
> Response: We will change the title to “...with and without spectral gap conditions” for further clarification.
>
> 2. “Section 3 is called Optimal rates without spectral gap but there is no corresponding section for "with".”
>
> Response: Due to page restrictions the main text has been dedicated to explaining the spectral gap independent results. The results based on spectral gap assumptions are more technical in nature and have been provided in supplemental file, section 7. We have mentioned this at the end of section 1.
>
> 3. “[line 25] While it can be assumed that if $Y\sim P$ then $p_i=P(Y=i)$ and similarly for $Q$, this should be made clear.”
>
> Response: We will clarify this in the revision.
>
>  4. “[lines 36-38] I am not sure about the correctness of the claim that “if a certain state has very small probability under the stationary distribution, consistent estimation of the transition matrix may not be possible”. Under irreducibility, every state is visited infinitely many times and consistent estimation is guaranteed. Under reducibility, some states may not be visited at all and the corresponding transition probabilities cannot be estimated.”
>
> Response:  Thanks for the question. The reviewer's reasoning is correct in the usual large-sample asymptotics, namely when the transition matrix is fixed and the length n of the trajectory tends to infinity. If the transition matrix depends on n (which is allowed in the worst-case analysis), then consistent estimation may not be possible.
>
> 5. “[lines 50-51] Since highly related to this work, the authors should clarify what exactly “implicit assumptions on mixing time such as spectral gap conditions” that were made by HOP18 meant.”
>
> Response: In HOP18, the proof of the upper bound for prediction (Lemma 9 in the main paper and Lemma 7  in the supplement) and for estimation (Lemma 17 of the supplement) is based on Berstein-type concentration results of the empirical counts $N_i,N_{ij}$, which depends on the spectral gap. We will clarify this in the revision.
>
> 6. “[line 87] It would be informative to explain the sudden restriction to reversible Markov chains. Is it an artifact of the analysis? … What about the pseudo spectral gap?”
>
> Response: The definition of spectral gap and absolute spectral gap requires irreducibility and reversibility. Thus when presenting spectral gap dependent risk bounds, we restrict this class of Markov chains. We agree it is possible to use more general notions such as pseudo spectral gap to quantify the memory of the process, which is beyond the scope of the current paper.
>
> 7. “[line 96] In what sense is Theorem 2 an extension of (2) of [FOPS16]?”
>
> Response:  FOPS16 gives the minimax rates over all binary Markov chains, which corresponds to the case of $\gamma_0=0$ in Theorem 2. Theorem 2 determines the minimax rate for all $\gamma_0$.
>
> 8. “[line 142] To which reduction do the authors refer?”
>
> Response:  We referred to the reduction in equation (11). We will clarify this in the revision.

---

### Official Review · Reviewer_oC8r · 2021-07-14

**Rating:** 7
**Confidence:** 4

**Summary:**

From a single trajectory of $n$ observations drawn from a Markov chain over $k$ different states, we are interested in predicting the next one.
For some notion of risk they define, the authors prove that when $k \geq 3$,
the rate is of the order $\Theta((k^2/n) \log (n/k))$.
In particular, this shows the surprising fact that this rate has a different (worse) dependence in $n$ than chains over $k = 2$ states, known to be $\Theta((1/n)\log \log n)$.
Furthermore, they analyze the minimax problem over the set of chains whose spectral gap is controlled from below, and
show that under favorable (quantified) mixing conditions, one even recovers the "iid parametric" rate $\Theta(k^2/n)$.

**Ethical Concerns:**

None.

**Limitations And Societal Impact:**

Addressed adequately.

**Main Review:**

The paper is very-well written, and offers strong results. The problem is hard, and was until now only understood for the restricted 2-state case. The bounds are matching up to universal constants. The techniques are involved, enabling the authors to obtain bounds that are independent of the mixing properties of the chain (Th.1).
Although this reviewer did not verify the proofs carefully, they have found no reason to doubt them. This is a strong submission; this reviewer recommends acceptance.

* (L31-L38;L186-192)
In these two paragraphs, the authors seem to imply that there is something inherently simpler about the prediction problem than, say, estimation, illustrating this by saying that consistent estimation of the transition kernel is not possible if the stationary distribution comprises extremely small values. This last part, however, is a bit misleading and depends on the notion of distance one considers.
It is true that some notions of distance lead to a minimax rate that also captures a dependence in the stationary distribution.
But for example, by weighing the state-wise discrepancy in terms of the stationary distribution as in Hao et al. [2018, p.2], or considering a distance in terms of the edge-measures of the chains as  in Wolfer and  Kontorovich  [2021,  Theorem  3.3], it is possible to obtain rates that are independent on the stationary distribution for the estimation problem.

* (L79-81) Note that such interesting decoupling has also been observed in the Markov chain testing literature for some notion of distance (cited by the authors).

* Add a few words on algorithmic complexity.

* On page 1, when taking the infimum, indicate whether estimators must be (proper) Markov kernels.

* Since the results are expressed up to universal constants, one can remove squares from the logs in the bounds $\log (1 / k^2) = C \log 1/k$.

**REFERENCES**

* Y. Hao, A. Orlitsky, and V. Pichapati. On learning markov chains. In NeurIPS, 2018.

* Wolfer  and  A.  Kontorovich.    Statistical  estimation  of  ergodic  markov  chain  kernel  over  discrete  space. Bernoulli, 27(1):532–553, 02 2021.  doi:  10.3150/20-BEJ1248.

**Time Spent Reviewing:**

4

---

> ### Author Response · Authors · 2021-08-10
> **Responses to Reviewer 3**
>
> We thank the reviewer for the constructive and positive comments. Specific responses to comments are given below.
>
> 1. "(L31-L38;L186-192) In these two paragraphs, the authors seem to imply that there is something inherently simpler about the prediction problem than, say, estimation, illustrating this by saying that consistent estimation of the transition kernel is not possible if the stationary distribution comprises extremely small values. This last part, however, is a bit misleading and depends on the notion of distance one considers.
> It is true that some notions of distance lead to a minimax rate that also captures a dependence in the stationary distribution. But for example, by weighing the state-wise discrepancy in terms of the stationary distribution as in Hao et al. [2018, p.2], or considering a distance in terms of the edge-measures of the chains as in Wolfer and Kontorovich [2021, Theorem 3.3], it is possible to obtain rates that are independent on the stationary distribution for the estimation problem."
>
> Response: Thanks for the suggestion. We will clarify in the revision that here consistent estimation is with respect to the usual loss function e.g. squared risk.
>
> About the two references: the minimax rate of Hao et al. [2018, p.2] is obtained for chains with transition probabilities bounded away from 0 and 1 (say by delta), which implicitly restricts stationary distribution $\pi$ away from 0 and 1. The terms involving delta in their minimax rates are suppressed in the theorem statement. On the other hand, the sample complexity produced in Wolfer and Kontorovich [2021, Theorem3.3] also depends on $\pi$ via $||\mu/\pi||_{2,\pi}$.
>
> 2. “Add a few words on algorithmic complexity.”
>
>
> Response: The estimator achieving the minimax rates is given by $\tilde Q$ in equation (25) for m=2, which is average of the add-one estimators. Given any $X^{n-1}=x^{n-1}$, estimating the probability of $X_n=j$ entails calculating add one estimator $\hat{M}^{+1}(j|x_{n-1})$ for the sequences $x^{n-1}_{n-t+1}$, $t=2,\dots,n$, and then taking their average.
>
> Calculating $\hat M^{+1}(j|x_{n-1})$ for sequence $x_1^{n-1}$ takes $O(n)$ time. Given the add one estmator based on
> $x_{n-t+1}^{n-1}$ we need $O(1)$ time to calculate the same based on $x^{n-1}_{n-t+2}$. Summing over all $j$ this implies the total time complexity $O(nk)$. We will add this in the revision.
>
> 3. “On page 1, when taking the infimum, indicate whether estimators must be (proper) Markov kernels.”
>
> Response: Thanks for pointing it out. We will indicate this in the revised version.

---

> > ### Comment · Reviewer_oC8r · 2021-08-17
> > **Score unchanged**
> >
> > Many thanks to the authors for their detailed response.
> > This reviewer keeps the score unchanged: Accept.

---

### Official Review · Reviewer_fh3J · 2021-07-14

**Rating:** 7
**Confidence:** 3

**Summary:**

The paper studies the following problem: given data from a stationary Markov chain with discrete support, can we learn the stationary distribution? It derives the optimal rates for this learning problem when the spectral gap of the Markov chain is arbitrarily slow, and shows that when the spectral gap is bounded below, one can achieve parametric rates.

**Limitations And Societal Impact:**

Yes

**Main Review:**

The paper studies a clear and well motivated problem, and introduces universal compression techniques to the study of Markov chains. The result for Markov chains without spectral gap assumptions is nice.

**Time Spent Reviewing:**

2

---

> ### Author Response · Authors · 2021-08-10
> **Responses to Reviewer 2**
>
> We thank the reviewer for the positive comments.

---

### Official Review · Reviewer_AyiV · 2021-07-17

**Rating:** 7
**Confidence:** 3

**Summary:**

This paper studies the prediction problem for Markov chain with more than $3$ states, and without the spectral gap.

**Limitations And Societal Impact:**

This is a theoretical paper, and I think there is no potential negative societal impact.

**Main Review:**

Strengths:

1. This paper proves the optimal prediction risk with the number of states between 3 and $\sqrt{n}$ without the spectral gap, which requires more difficult proof technique than previous works analyzing markov chain with 2 states or with spectral gap.

2. To achieve optimality, the paper defines an average of add-on estimators to be a new estimator and proves its optimality.

3. The role of spectral gap is characterized clearly in the prediction risk.

Weakness / comments:

1. The most natural estimator for $M(j|i)$ might be the empirical frequency $N_{ij}/N_i$ instead of the add-one estimator in (5). It might be beneficial to discuss the reason why you use the add-one estimator: is it because empirical frequency is sub-optimal in nature, or just a technical issue.

2. The structure of this paper is a little strange. It might be better to move the main theorems and proof techniques (Section 1.1) outside the introduction (Section 1).

**Time Spent Reviewing:**

1.5

---

> ### Author Response · Authors · 2021-08-10
> **Responses to Reviewer 1**
>
> We thank the reviewer for the constructive and positive comments. We will try to implement the comments about wordings that have been pointed out. Specific responses to comments are given below.
>
> “The most natural estimator for $M(j|i)$ might be the empirical frequency $N_{ij}/N_i$ instead of the add-one estimator in (5). It might be beneficial to discuss the reason why you use the add-one estimator: is it because empirical frequency is sub-optimal in nature, or just a technical issue.”
>
> Response: For KL risk, directly using empirical distribution leads to infinite risk when certain symbols have non-zero probability but are not observed. For this reason, smoothed versions  of empirical distribution (add-c estimators) are used. This reasoning is the same for both iid and for Markov data.

---

> > ### Comment · Reviewer_AyiV · 2021-09-03
> > **Thanks for your response**
> >
> > I thank the authors for responding to my questions. I will update my score.

---

### Decision · Program_Chairs · 2021-09-27

**Decision:**

Accept (Poster)

**Comment:**

This paper is interested in predicting the next state, given $n$ observations from a single trajectory of a Markov chain. It is very well-written and offers strong, significant, interesting results.